# Regulation of the luminescence mechanism of two-dimensional tin halide perovskites

Tianju Zhang [1,2], Chaocheng Zhou[3,4], Xuezhen Feng[5], Ningning Dong [1,2], Hong Chen [5], Xianfeng Chen[4,6], Long Zhang[2], Jia Lin [3✉] & Jun Wang [1,2,7✉]

Two-dimensional (2D) Sn-based perovskites are a kind of non-toxic environment-friendly luminescent material. However, the research on the luminescence mechanism of this type of perovskite is still very controversial, which greatly limits the further improvement and application of the luminescence performance. At present, the focus of controversy is defects and phonon scattering rates. In this work, we combine the organic cation control engineering with temperature-dependent transient absorption spectroscopy to systematically study the interband exciton relaxation pathways in layered $A_2SnI_4$ (A = $PEA^+$, $BA^+$, $HA^+$, and $OA^+$) structures. It is revealed that exciton-phonon scattering and exciton-defect scattering have different effects on exciton relaxation. Our study further confirms that the deformation potential scattering by charged defects, not by the non-polar optical phonons, dominates the excitons interband relaxation, which is largely different from the Pb-based perovskites. These results enhance the understanding of the origin of the non-radiative pathway in Sn-based perovskite materials.

[1] Laboratory of Micro-Nano Optoelectronic Materials and Devices, Shanghai Institute of Optics and Fine Mechanics, Chinese Academy of Sciences, Shanghai 201800, China. [2] Center of Materials Science and Optoelectronic Engineering, University of Chinese Academy of Sciences, Beijing 100049, China. [3] Department of Physics, Shanghai Key Laboratory of Materials Protection and Advanced Materials in Electric Power, Shanghai University of Electric Power, Shanghai 200090, China. [4] State Key Laboratory of Advanced Optical Communication Systems and Networks, School of Physics and Astronomy, Shanghai Jiao Tong University, Shanghai 200240, China. [5] State Environmental Protection Key Laboratory of Integrated Surface Water-Groundwater Pollution Control, Guangdong Provincial Key Laboratory of Soil and Groundwater Pollution Control, School of Environmental Science and Engineering, Southern University of Science and Technology, Shenzhen 518055, China. [6] Collaborative Innovation Center of Light Manipulation and Applications, Shandong Normal University, Jinan 250358, China. [7] CAS Center for Excellence in Ultra-intense Laser Science, Shanghai 201800, China. ✉email: jlin@shiep.edu.cn; jwang@siom.ac.cn

Since 2012, organic-inorganic lead halide perovskite materials with different dimensions have been attracting significant attention from researchers in energy, physics, and other disciplines mainly because of their applications in solar cells[1,2], photodetectors[3,4], light communications[5–7], light-emitting diodes (LEDs)[8–10], lasers[11], and so on. However, the issue associated with long-term stability against moisture and the toxicity of lead is still a challenge, hindering the use of Pb-based perovskites for practical applications[12]. Two-dimensional (2D) perovskites with organic cations sandwiching the perovskite slabs which are similar to those of semiconductor quantum wells have recently received more attention not only for improving the stability due to the hydrophobic organic cationic layer which protects the chemical properties of the inorganic layer but also adjusting exciton binding energy by changing the type of organic cations[13]. Considering the negative environmental impacts of $Pb^{2+}$, $Sn^{2+}$ is nontoxic and has the most similar ionic radius compared to $Pb^{2+}$. Accordingly, the lattice parameters obtained by $Sn^{2+}$ substitution will not be seriously changed in principle; thus, Sn-based perovskites are expected to replace Pb-based perovskites in the future. Interestingly, the emission wavelength of single-octahedral-layer Sn-based perovskite is about 630 nm, which is suitable for use in pure red displays[14]. At the same time, the limitation of Sn-based perovskites is that the oxidation potential of $Sn^{2+}/Sn^{4+}$ (−0.15 eV) is considerably lower than that of $Pb^{2+}/Pb^{4+}$ (−1.8 eV)[15]. Sargent et al. used valeric acid to protect $Sn^{2+}$ from undesired oxidation and reduced $Sn^{4+}$ content[14], which enables perovskite LEDs with full width at half maximum (FWHM) of 20 nm and the external quantum efficiency (EQE) of 5%, but that is much lower than the EQE of 9.5% of organic-inorganic hybrid Pb-based perovskites[10]. Tan et al. have shown that strong exciton-phonon coupling can cause the low photoluminescence quantum yield (PLQY) of 2D Sn-based halide perovskites, besides the high $Sn^{4+}$ defect state density[16]. But the effects of key parameters, such as defects and phonon scattering rates, on the luminescence properties, remain ambiguous. Therefore, it is necessary to study the temperature-dependent carrier dynamics to further reveal how these two factors affect the luminescence properties of 2D Sn-based perovskites, to achieve better luminescence performance of 2D Sn-based perovskite than that of 2D Pb-based perovskite.

For the luminescence properties of 2D Pb-based perovskites, through time-resolved and temperature-dependent PL studies, Huang et al. revealed that the scattering of excitons with acoustic phonons and nonpolar optical phonons is the main factor affecting the luminous efficiency of mechanically stripped 2D Pb-based perovskite sheets[17]. Through theoretical calculations, Ghosh et al. revealed that the higher the rigidity of the organic cation, the smaller the fluctuation of the inorganic framework structure, the weaker the exciton–phonon interaction, and the higher the PLQY[18]. Additionally, the electron–phonon scattering phenomenon in polar semiconductors such as perovskites induces self-trapping states, which can also significantly influence their optical properties and energy transport mechanism[19]. We note that the structural composition of the organic cationic layer can influence the defect state density and the exciton–phonon interaction[20], so we can vary the types of organic cationic layers and compare the effects of different cations on the luminescence properties and carrier dynamics of 2D Sn-based perovskites to distinguish how defects and phonon scattering affect the optical properties of materials, and furtherly reveal the difference in the factors affecting the luminescence properties between Sn-based and Pb-based perovskites.

In this work, we select two representative kinds of organic cations, one is phenyl-ethylammonium ($C_6H_5CH_2CH_2NH_3^+$ ($PEA^+$)) cation and the other are the derivatives of alkyl–ammonium chain with different numbers of carbon atoms which are n-butylammonium (($CH_3(CH_2)_3NH_3^+$ ($BA^+$)), hexylammonium ($CH_3(CH_2)_5NH_3^+$ ($HA^+$)), and octylammonium ($CH_3(CH_2)_7NH_3^+$ ($OA^+$)). The $PEA^+$ cations have the $CH–\pi$ stacking characteristics that alkyl chain cations lack, which limit their thermal movement between the inorganic octahedron layers, and different lengths of alkyl chains distort the inorganic octahedron frame in different degrees. So these organic cations not only regulate the exciton–phonon scattering process but also affect the structural characteristics and the density of defect states in the materials. The effects of defect scattering and optical phonon scattering on the relaxation dynamics of excitons were investigated by temperature-dependent PL spectroscopy and transient absorption (TA) spectroscopy. We find it is the deformation potential scattering by charged defects, not by the optical phonons, that dominates the interband exciton relaxation, leading to an increase in the proportion of non-radiative relaxation of photogenerated carriers, for the 2D Sn-based perovskite materials. So it is different from the Pb-based perovskites which the main scattering mechanisms for excitons are deformation potential by acoustic and homopolar optical phonons. Besides, compared with alkyl–ammonium chain cations, $PEA^+$ cation has a stronger ability to prevent $Sn^{2+}$ oxidation, reduce the $Sn^{4+}/Sn^{2+}$ ratio, increase the exciton radiative recombination ratio, and weaken the exciton–phonon scattering intensity. These results enhance the understanding of the origin of the non-radiative pathway and provide physical support for the future synthesis of high-performance luminescent 2D Sn-based perovskite materials.

## Results and discussion

**Sample structure characteristics.** To study the correlation between the structures and optical properties, we used single-crystal X-ray diffraction (SCXRD) to reveal the influence of four organic spacer cations with different chain lengths and geometry on the structures of the materials are shown in Fig. 1. We used the spin-coating method to prepare four kinds of perovskite thin films to explain the physical mechanism that affects the optical properties of 2D Sn-based perovskites. The influence of different cations on the film formation quality was studied through observing the surface morphology of the samples by scanning electron microscopy (SEM), and we found that the $PEA^+$ cation helps to form large size grains with obvious grain boundaries

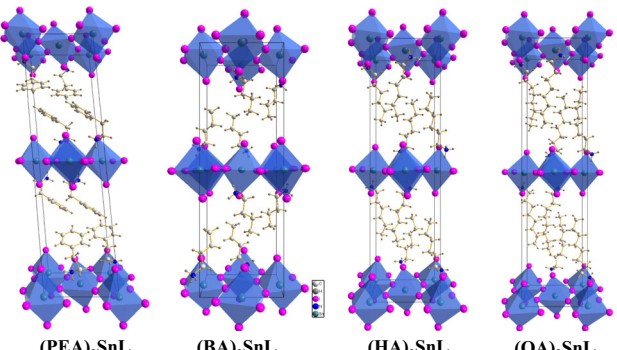

| $(PEA)_2SnI_4$ | $(BA)_2SnI_4$ | $(HA)_2SnI_4$ | $(OA)_2SnI_4$ |

**Fig. 1 Crystal structures of four different types of 2D Sn-based perovskites.** Two-dimensional Sn−I octahedron layers can be seen stacked on top of each other separated by two layers of organic cations ($PEA^+$, $BA^+$, $HA^+$, and $OA^+$). Magenta represents *I* atoms, dark green represents *Sn* atoms, dark blue represents *N* atoms, milky white represents *C* atoms, and gray represents *H* atoms. The detailed crystal structure data are shown in Supplementary Table 1 and Supplementary Note 1.

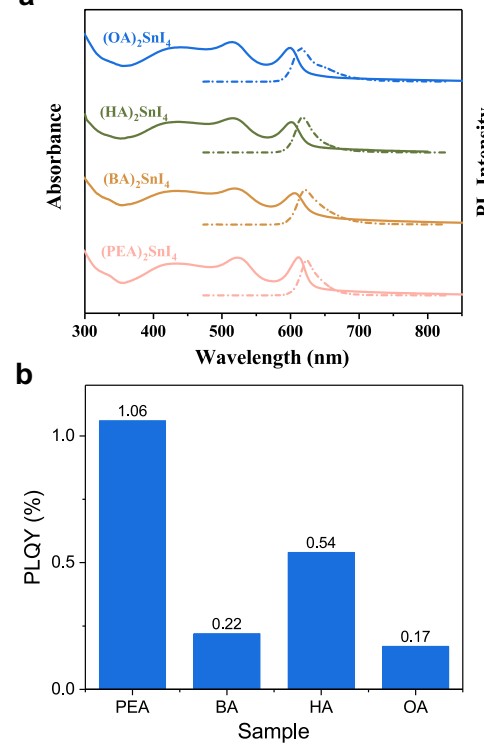

**Fig. 2 The absorption and PLQY of the four perovskites. a** The absorption (solid line) and PL spectra (dot line) of $(PEA)_2SnI_4$ (magenta), $(BA)_2SnI_4$ (tawny), $(HA)_2SnI_4$ (dark green), and $(OA)_2SnI_4$ (blue). **b** The corresponding PLQY of the four perovskites.

without pinholes. For $BA^+$, large and discontinuous perovskite islands were formed. For $HA^+$, small size grains with pinholes were observed. For $OA^+$, the top surface became blurry (Supplementary Fig. 1). Furtherly, we systematically investigated the effect of different organic cationic layers on the density of defect states in the material using X-ray photoelectron spectroscopy (XPS) and ultraviolet photoelectron spectroscopy (UPS) experiments. The oxidation potential of $Sn^{2+}/Sn^{4+}$ ($-0.15$ eV) is considerably lower than that of $Pb^{2+}/Pb^{4+}$ ($-1.8$ eV)[15]. Consequently, Sn-based perovskites have higher $Sn^{4+}$ defect states than the Pb-based perovskites. To prove this point, we applied XPS to study the influence of the organic cations on the chemical and elemental states of Sn. Supplementary Fig. 2 shows the high-resolution XPS spectra of the Sn $3d_{3/2}$ and Sn $3d_{5/2}$ regions of the 2D Sn-based perovskite films with different organic cations ($PEA^+$, $BA^+$, $HA^+$, and $OA^+$), which are fitted by two Gaussian distributions representing the distributions of $Sn^{2+}$ (486.6 and 495.1 eV) and $Sn^{4+}$ (487.4 and 488.9 eV), respectively[16,21]. In this way, we calculate the $Sn^{4+}/Sn^{2+}$ ratio of $(PEA)_2SnI_4$ to be 0.12, which is considerably smaller than those of $(BA)_2SnI_4$ (0.6), $(HA)_2SnI_4$ (0.58), and $(OA)_2SnI_4$ (0.77). These results show that $PEA^+$ cations have a stronger ability to protect $Sn^{2+}$ from oxidation than the organic alkyl chain spacers ($BA^+$, $HA^+$, and $OA^+$), and this protection ability weakens as the alkyl chain length increases. So this illustrates that the $Sn^{4+}$ defect state density in the $(PEA)_2SnI_4$ sample is much lower than that in the alkyl chain samples, which is consistent with the density functional theory (DFT) calculation result[18]. The main reason is that the large molecules of ammonium organic ions are beneficial for the formation of the compact pinhole-free films and block moisture ingress at the boundaries of perovskite nanolayers[20]. $BA^+$ ions increased the defect formation energy of $Sn^{4+}$ by

0.33 eV, while $PEA^+$ could increase the defect formation energy of $Sn^{4+}$ by 0.6 eV[22]. Therefore, compared to $BA^+$, $PEA^+$ effectively hinders the formation of tin vacancies and tin oxidation. The increase in the $Sn^{4+}$ defect concentration can make the Fermi level ($E_F$) of the material approach the valence band maximum (VBM), and this phenomenon was further verified by the UPS experiment (Supplementary Note 2)[21]. Supplementary Fig. 3 presents the derived energy band diagrams of $(PEA)_2SnI_4$, $(BA)_2SnI_4$, $(HA)_2SnI_4$, and $(OA)_2SnI_4$. The energy differences between the VBM and $E_F$ are 1.0, 0.66, 0.77, and 0.53 eV for $(PEA)_2SnI_4$, $(BA)_2SnI_4$, $(HA)_2SnI_4$, and $(OA)_2SnI_4$ samples, respectively, which confirms that the $E_F$ values of the perovskite samples containing organic alkyl chain cations are closer to the VBM compared with that of the $PEA^+$ case and the $E_F$ decreases as the alkyl chain length increases.

**Fundamental optical properties.** The fundamental optical properties of the four perovskite thin films were investigated via UV−Vis absorption and steady-state PL spectroscopy analyses, and the results (Fig. 2a) revealed that the four perovskite poly-crystalline thin films have similar optical characteristics. For $(PEA)_2SnI_4$, three main absorption peaks were observed. The first peak at 420 nm (2.95 eV) was assigned to the high-energy exciton transition energy levels, and the second peak at 520 nm (2.38 eV) was assigned to the intraband transition process in the perovskite layer rather than the charge transfer transition between the organic spacer cations and the inorganic layers[23–25]. The relevant discussions are discussed in Supplementary Note 3. And the sharp peak at 613 nm (2.02 eV) was attributed to the intrinsic band-edge exciton absorption[26]. The PL peak of $(PEA)_2SnI_4$ is located at 624 nm (1.987 eV), with a narrow FWHM of 28 nm (91 meV), which are slightly lower than the values reported in the literature[27] (PL peak position: 628.2 nm, FWHM: 104 meV). The small Stokes shift and narrow bandwidth suggest that the emission almost results from the recombination of the intrinsic excitons (Supplementary Table 2)[28]. On further observation, both the optical bandgap and PL emission peak exhibit a blueshift, indicating the increase in the optical bandgap (Supplementary Table 2 and Supplementary Fig. 6). From a physical viewpoint, 2D perovskite materials possess the characteristics of quantum wells in which the thickness of the inorganic layers and the structure and dielectric coefficient ε of organic cations can change the electronic band structure and affect the exciton binding energy ($E_b$) of the materials. In 2D layered perovskites, different cations can influence the octahedral tilting angle and the length of the Sn-I bond to modulate the bandgap of the material[18,29]. The influence of cations on the bandgap can be explained more comprehensively using theoretical calculations based on density functional theory calculations (DFT). Through DFT calculations, we found that the bandgap of $(PEA)_2SnI_4$ is the smallest and that of $(BA)_2SnI_4$, $(HA)_2SnI_4$, and $(OA)_2SnI_4$ are increasing in order (Supplementary Fig. 7); this is consistent with our results obtained using steady-state spectroscopy and other previously reported results[29,30]. The experimental results of the excitation intensity-dependent integral PL intensity indicate that the power-law dependence $1 < K < 2$ holds in the four materials (Supplementary Fig. 8); this implies that excitons are the nature of optical transitions in the materials[28]. For $(A)_2SnI_4$ (A: $PEA^+$, $BA^+$, $HA^+$, and $OA^+$), $\varepsilon_A$ is smaller than $\varepsilon_w$ (w: $SnI_4$), leading to enhanced the Coulomb interaction between the electron and hole to compose the exciton because of the reduced dielectric screening of the exciton electric field[13,31]. There are many methods to determine $E_b$, such as the absorption spectrum, temperature-dependent PL, and magneto-optical investigation[31–34]. However, the polar nature of perovskites and the associated polaron effects are neglected;

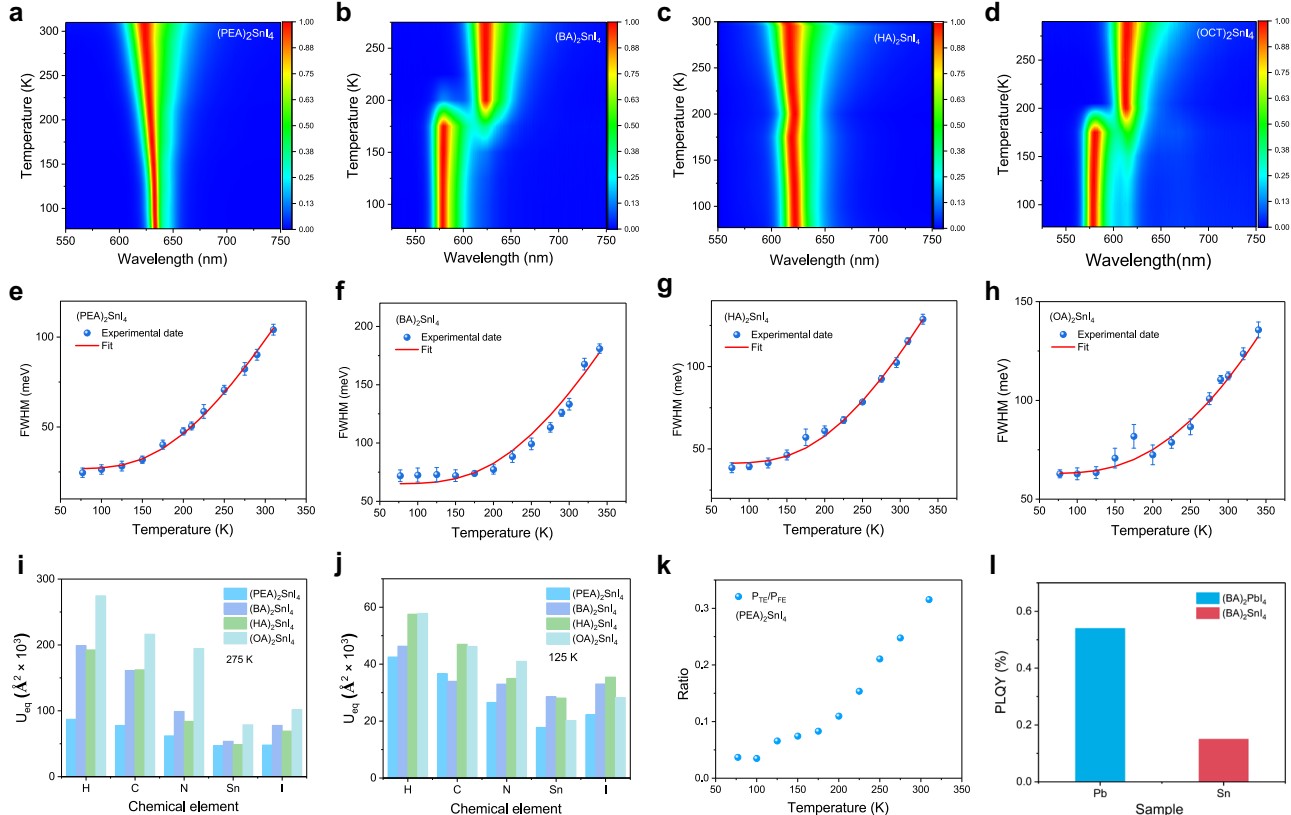

**Fig. 3 Analysis of exciton–phonon coupling effects of the four perovskites.** Contour map of the temperature-dependent normalized PL spectra: **a** (PEA)$_2$SnI$_4$, **b** (BA)$_2$SnI$_4$, **c** (HA)$_2$SnI$_4$, and **d** (OA)$_2$SnI$_4$ films obtained under continuous wave (CW) laser excitation at a wavelength of 473 nm, a power density of 2 µJ cm$^{-2}$, and from 300 to 77 K with 25 K intervals. The color scales represent the intensity of normalized PL. Temperature-dependent FWHM for **e** (PEA)$_2$SnI$_4$, **f** (BA)$_2$SnI$_4$, **g** (HA)$_2$SnI$_4$, and **h** (OA)$_2$SnI$_4$. The blue dot represents the experimental data, the short lines above and below the blue dots represent the error bars, and the red solid lines represent the fitting results of the exciton–phonon coupling model (3) in Supplementary Note 6. Average atomic displacement $U_{eq}$ of chemical elements (*H, C, N, Sn,* and *I*) in (PEA)$_2$SnI$_4$, (BA)$_2$SnI$_4$, (HA)$_2$SnI$_4$, and (OA)$_2$SnI$_4$ at **i** 275 K and **j** 125 K extracted from single-crystal X-ray diffraction (SCXRD) data (Supplementary Information). **k** Characteristics of the ratio (P$_{TE}$/P$_{FE}$) of trapping-state exciton PL intensity to free exciton PL intensity as a function of temperature. **l** the PLQY of (BA)$_2$PbI$_4$ and (BA)$_2$SnI$_4$ obtained under the same experimental conditions.

this makes the $E_b$ values obtained by different methods under different experimental conditions highly discrepant[33]. The accuracy of $E_b$ obtained by fitting the temperature-dependent PL based on the Arrhenius formula is severely affected by other recombination processes, such as shallow defect trapping excitons and Auger recombination[33]. Therefore, $E_b$ was obtained by fitting the steady-state absorption spectrum using a more rigorous Elliott theory (Supplementary Fig. 9 and Supplementary Note 4.)[34], in which the $E_b$ of (PEA)$_2$SnI$_4$ is 213 ± 2 meV smaller than that of (BA)$_2$SnI$_4$ (245 ± 1.6 meV) because $\varepsilon_{BA}$ is smaller than $\varepsilon_{PEA}$. $E_b$ is greater than the thermal energy ($K_B T \approx 25$ meV at 300 K), which further reveals excitons dominate the optical transitions of the 2D layer Sn-based perovskites at room temperature[31]. Although the broadening factor affects the true value of $E_b$ obtained through Elliott's theory at room temperature, it still provides a qualitative comparison of the effect of the dielectric constants of organic cations on the $E_b$.

It can be found that with the PEA$^+$ spacer cations substituted by the alkyl–ammonium chain (such as BA$^+$, HA$^+$, and OA$^+$), PLQY more significantly decreases (Fig. 2b), indicating that (PEA)$_2$SnI$_4$ has the lowest defect states density among these four materials. The non-radiative recombination process in the materials is related not only to the defect density but also to the interaction between excitons and phonons. To further investigate the influence of different organic cations on the

interactions between excitons and phonons in the materials, a temperature-dependent PL experiment was conducted.

**Exciton−phonon coupling.** The results of the temperature-dependent PL experiment are shown in Fig. 3a–d. And the information regarding traps, phase transitions, and bandgap evolution with temperature provided by temperature-dependent PL spectroscopy are analyzed and given in detail in Supplementary Note 5.

The analysis of the temperature-dependent FWHM of PL is the main means to evaluate the mechanisms of electron–phonon coupling in various semiconductors[35]. The variation of the FWHM with temperature involves various scattering physical processes including the scattering process between electrons, optical phonons, and acoustic phonons in the material, which causes the electrons to shift to the thermal equilibrium position, thus affecting the electronic band structure of the material and changing the characteristics of the PL spectra. For most semiconductors, using the first-order perturbation theory, the temperature-dependent characteristics of the PL peak can be simplified to four scattering mechanisms as follows in Supplementary Note 6[35]. Based on the exciton–phonon coupling model, the best-fitting parameters listed in Table 1 are obtained using a simulated annealing algorithm, and Fig. 3e–h plot the fitting results (red lines) of the FWHM data. To improve the credibility

**Table 1 Best-fitting parameters of the $(PEA)_2SnI_4$, $(BA)_2SnI_4$, $(HA)_2SnI_4$, and $(OA)_2SnI_4$ perovskites.**

| Sample | $\Gamma_O$ (meV) | $\Gamma_{LO}$ (meV) | $E_{LO}$ (meV) | $\Gamma_{imp}$ (meV) | $E_{imp}$ (meV) |
|---|---|---|---|---|---|
| $(PEA)_2SnI_4$ | $26.8 \pm 1$ | $199.3 \pm 2$ | $58.5 \pm 1$ | $722.3 \pm 2$ | $69.7 \pm 2$ |
| $(BA)_2SnI_4$ | $63.1 \pm 2$ | $272.8 \pm 4$ | $56.8 \pm 2$ | $1513.3 \pm 2$ | $89.9 \pm 2$ |
| $(HA)_2SnI_4$ | $41.3 \pm 2$ | $254.9 \pm 5$ | $56.8 \pm 1$ | $881.2 \pm 2$ | $82.3 \pm 2$ |
| $(OA)_2SnI_4$ | $66.2 \pm 2$ | $320.2 \pm 7$ | $57.85 \pm 1$ | $1211.7 \pm 2$ | $120.7 \pm 2$ |

of the fitting results, we used the optical phonon energies obtained from the steady-state Raman experiments (Supplementary Fig. 13). A Raman peak is located at 454.3 cm$^{-1}$ (56.3 meV) for $(PEA)_2SnI_4$ and 472.1 cm$^{-1}$ (58.5 meV) for the alkyl chain group samples. The interpretation of these Raman peaks requires strict theoretical calculations, which are beyond the scope of this study. Compared with the alkyl chain samples, the $(PEA)_2SnI_4$ sample not only has a smaller $\Gamma_0$ but also has a relatively smaller Fröhlich coupling intensity ($\Gamma_{LO}$). The results indicate that the $(PEA)_2SnI_4$ sample is more ordered, and the non-radiative energy loss is smaller than that of the alkyl chain samples. The main reason is that the $PEA^+$ cation has the CH–π stacking characteristics that alkyl chain cations lack, which limits their thermal movement between the inorganic layers, and weak dynamic changes are induced in the $SnI_4$ structure[18]. For the samples with an alkyl chain, $(HA)_2SnI_4$ ($254.9 \pm 5$ meV) has the lowest $\Gamma_{LO}$, followed by $(BA)_2SnI_4$ ($272.8 \pm 4$ meV), and finally $(OA)_2SnI_4$ ($320.2 \pm 7$ meV). This implies that the longer alkyl–ammonium chain tends to enhance the intensity of exciton–phonon scattering. Notably, the exciton–phonon Fröhlich interaction of the 2D Sn-based perovskites that was studied is over one order of magnitude larger than that reported for lead perovskites (Supplementary Table 3), which can lead to an increase in the FWHM of PL and the possibility of forming self-trapped exciton (STE) states. The STE further may further couple with defect states to result in the interband PL[36–39]. To furtherly investigate the relationship between the material structure and electron−phonon interactions, we also studied the atomic displacement parameters $U_{eq}$ extracted from SCXRD data of the four materials[9]. The results revealed that the atomic displacements of the different atoms $U_{eq}$ of $A_2SnI_4$ (A: $BA^+$, $HA^+$, and $OA^+$) were distinctly larger than those of $(PEA)_2SnI_4$, as shown in Fig. 3i, j. Thus, $(PEA)_2SnI_4$ has a more rigid structure than that of $(BA)_2SnI_4$, $(HA)_2SnI_4$, and $(OA)_2SnI_4$; this is consistent with the results of the exciton–phonon coupling model and theoretical calculations[18]. Although organic cations cannot directly participate in the band-edge electronic states of the materials in this research, their structural packing and dynamic coupling with the inorganic framework of Sn–I–Sn affect the luminescent properties of the materials[18,30,40]. Thus, in the 2D perovskite system, the more rigid the structure, the more beneficial it is to improve the color purity of the emitted light and reduce the energy loss of non-radiative recombination.

**Exciton relaxation dynamics**. TA experiments are often used to understand the photoexcited carrier dynamics and transport mechanisms in perovskites. Fig. 4 shows the false-color 2D TA mappings of the $(PEA)_2SnI_4$, $(BA)_2SnI_4$, $(HA)_2SnI_4$, and $(OA)_2SnI_4$ thin films. Since the characteristics of the steady-state absorption spectrum and the TA spectrum of the four materials are essentially similar, the $(PEA)_2SnI_4$ sample is applied as the representative. There are two main photobleaching (PB) peaks whose centers are at 615 and 525 nm, corresponding to the absorption peaks in the linear absorption spectra attributed to the band filling effect. A weak and broad (650–730 nm) PB signal

below the bandgap is caused by the defect states with low absorption cross-sections below the optical bandgap filled by the photogenerated carriers[19,41]. Besides the PB signals, positive photoinduced absorption (PIA) signals exist on both sides of the bleaching peak.

The relaxation kinetics of photogenerated carriers have been fitted globally with three relaxation components (Supplementary Fig. 14), in which the lifetime of the first component (I) is of the order of sub picoseconds, that of the second component (II) is approximately 100 ps, while that of the third component (III) is of the order of nanoseconds, exceeding the system test range (1.6 ns); this component is not discussed in detail here for accuracy. The effect of each process on the PL properties is further studied by combining temperature-dependent and pump fluence-dependent TA experiments. For $(BA)_2SnI_4$ and $(OA)_2SnI_4$ samples, the bleaching peak below 200 K is blue-shifted mainly due to the increase in bandgap caused by the phase transition mentioned earlier. (Supplementary Fig. 15), However, for the $(PEA)_2SnI_4$ and $(HA)_2SnI_4$ samples, the TA spectra were characterized by a redshift and a gradual increase in the intensity of the bleaching peak at the band edge with decreasing temperature between 340 and 77 K. Here $(PEA)_2SnI_4$ is still applied as the representative material. The redshift of the bleaching peak of component I with a subpicosecond lifetime (277 fs) usually involves different physical processes, such as bandgap renormalization[42,43], optical Stark effect[44], exciton formation[42,45], and trap states trapping[19,46,47]. In the recent study of TA kinetics, the component I was attributed to the surface defect trapping exciton process (<1 ps) in 2D $(CH_3(CH_2)_8NH_3)_2PbBr_4$ perovskite[47], the surface trap states trapping exciton process (600 fs) for monolayer $MoS_2$ materials[46], and the exciton formation (<1 ps) in monolayer $WS_2$ materials[42,48]. Several reports have shown that the temperature affects the dielectric shielding effect in the material[49,50], i.e., in the low-temperature phase, the exciton binding energy increases, inducing an increase in the proportion and the formation rate of excitons[50]. The fact is that the relaxation rate of component I does not depend on the temperature (Fig. 5a). Therefore, component I is not attributed to the formation of excitons. The photoexcited free carriers may form excitons in the generation process of band-edge bleaching peak, which is difficult to distinguish for fast process systems with time scales of femtosecond magnitude. For the trap states in the 2D perovskites, we need to distinguish the nature of the trap states, that is, whether they are intrinsic STE states, defect trapping states, or extrinsic STE states. In the temperature-dependent PL experiment, using the multi-peak fitting methods (Supplementary Note 5), the $P_{TE}/P_{FE}$ decreases with decreasing temperature (Fig. 3k). This is evidently opposite to the feature of the intrinsic STE states emissions, in which the stronger luminescence from the intrinsic STE states and the band-edge exciton luminescence intensity decreases as the temperature decreases, mainly because the thermal activation of the detrapping process cannot meet the requirements of the detrapping barrier and the STEs cannot return to the band edge at low temperatures[36,39,51,52]. More precisely, this luminescence of trap states below the bandgap may be the emission of the extrinsic STE states; that is, intrinsic STE is influenced by the local heterogeneity of the permanent lattice defects to obtain a different trapping

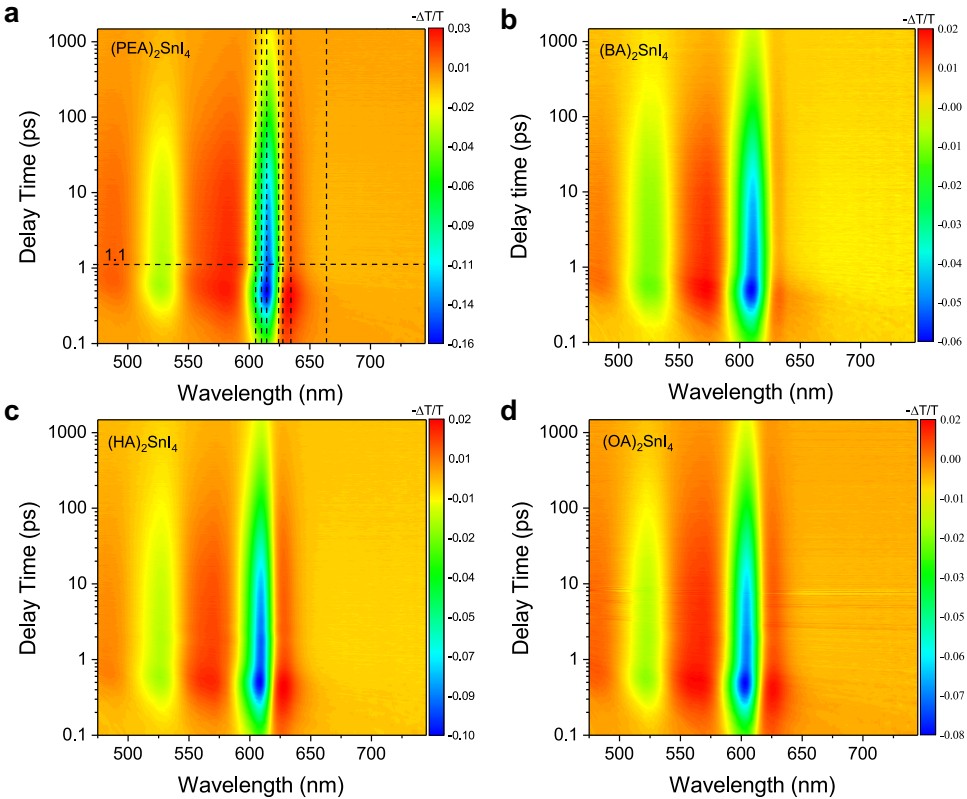

**Fig. 4 False-color 2D TA mapping of the four perovskites. a** $(PEA)_2SnI_4$, **b** $(BA)_2SnI_4$, **c** $(HA)_2SnI_4$, and **d** $(OA)_2SnI_4$ thin films with the excitation at 520 nm (2.38 eV) at a fluence of 2 μJ cm$^{-2}$ at room temperature. The color scale represents the signal intensity of the TA spectrum, that is, the signal intensity of the differential transmittance.

depth[37,39,53]. In the TA spectrum of the four materials (Fig. 4), there are a broad (640–750 nm) and weak PB feature below the optical gap; this is also consistent with the permanent defect states being filled, not the intrinsic STE states featured by a broad PIA at energies below the optical gap because of the formation of transient light-induced trap states[19,36,37]. The relaxation kinetics of different wavelengths show that the process of filling and bleaching of these defect states (663.4 nm), the relaxation process of the PIA signal (624.1 nm of PL center) changing from positive PIA to negative PB, and reaching the maximum are synchronous with the component I of the band-edge exciton (614.8 nm) relaxation (Fig. 5b), indicating that the band-edge excitons are trapped in the in-gap defect states. To further demonstrate that excitons are trapped by chemical defects, we applied stoichiometry engineering of the cations where the PEAI: SnI$_2$ ratio was 2.6:1 in $(PEA)_2SnI_4$ (PEAI-rich)[39,54]. We reduced the defect density in PEAI-rich to improve the PLQY of the PEAI-rich to 2.2% (Fig. 5c), and we also found that the occupancy ratio and relaxation rate of component I of the band-edge exciton relaxation processes in the PEAI-rich sample were significantly lower than those of the $(PEA)_2SnI_4$ sample (Fig. 5d). In addition, at high excitation intensities (10 μJ cm$^{-2}$.), the bleaching signal intensity and the relaxation rate of 676 nm were smaller for PEAI-rich than for $(PEA)_2SnI_4$ (Fig. 5e). In summary, component I contains the process of defect states trapping excitons. The increasingly serious broad-spectrum PL trailing phenomena can be explained by the extrinsic STE effect in $(HA)_2SnI_4$, $(BA)_2SnI_4$, and $(OA)_2SnI_4$, notably $(OA)_2SnI_4$ (Supplementary Fig. 12). Deschler et al. indicated that the broad emission below the optical gap observed at low temperatures in <001> oriented 2D perovskite materials was because of the light-induced formation of localized trap states, associated with interstitial iodide and iodide Frenkel defects that act as color centers in the crystal[53]. In addition, Loi

et al. highlighted the extrinsic in-gap states in the crystal bulk are responsible for the broadband emission[39]. Since $(OA)_2SnI_4$ has the strongest exciton–phonon coupling intensity with the highest density of defects among the four materials, there is a relatively wide PL trailing below the bandgap and this PL trailing is more pronounced at 77 K (Supplementary Fig. 12). Despite the relaxation rate and the proportion of the component I decrease at high pump fluences, as shown in Fig. 5f, this property does not imply that the defect states are filled; rather, it must be determined if the excited exciton density approaches the Mott density, resulting in exciton fission, or whether the material is degenerate as a result of strong light excitation. Assuming an exciton Bohr radius of 1 nm[55], the exciton saturation density is estimated to be ~$10^{14}$ cm$^{-2}$. The saturation of component I only somewhat occurs at a high excitation fluence of more than 40 μJ cm$^{-2}$ ($1.1 \times 10^{14}$ cm$^{-2}$), thus, the excitation intensity exceeds the saturation density causing the saturation of the component I cannot be characterized as a defect states trapping process. In the TA spectrum within 1 ps (Fig. 4), a redshift of the PB peak of the band-edge exciton leading to the PIA center at 625 nm appears; this is attributed to the bandgap renormalization caused by the hot excitons[19]. Besides, the maximum amplitude $-\Delta T/T$ of the pump fluence-dependent PIA satisfies a linear relationship with the excitation intensity $n^{1/2}$ (Fig. 5g)[56–58]. Therefore, this result further confirms the presence of the bandgap renormalization process in component I. To clearly distinguish which of these two processes plays a role in component I, we further investigated the pump fluence-dependent TA spectra of the PEAI-rich and $(PEA)_2SnI_4$. We found that component I was more significantly affected by the defect states at a weak excitation fluence, approximately no more than 10 μJ cm$^{-2}$ (Fig. 5d). When pump fluency is more than 10 μJ cm$^{-2}$, component I appears as the defect state density-independent and the excitation fluence-

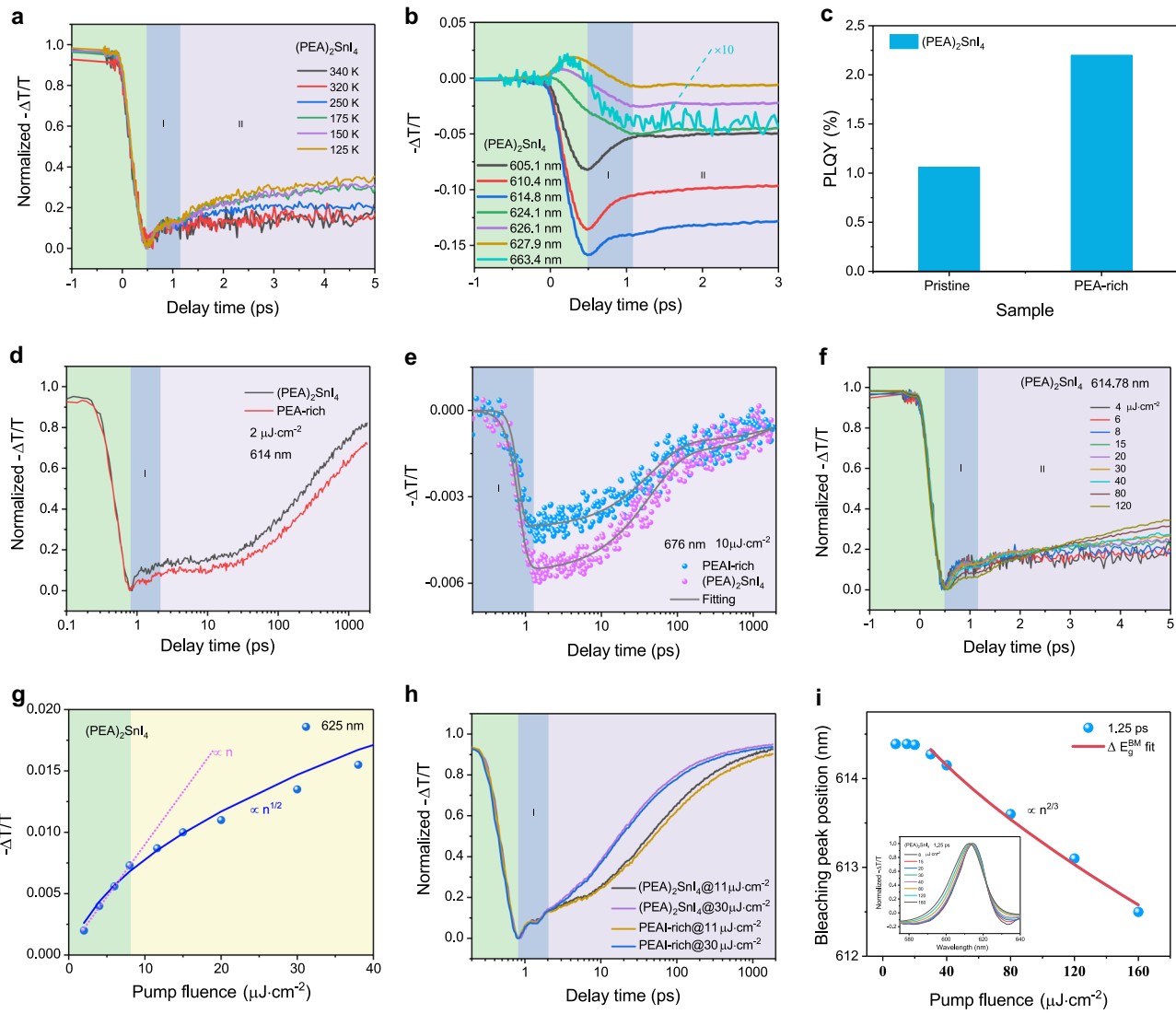

**Fig. 5 The first component (I) of the exciton relaxation process. a** Temperature-dependent normalized band-edge exciton relaxation kinetics of the $(PEA)_2SnI_4$ sample within 5 ps under the pump fluence of $2\,\mu J\,cm^{-2}$. **b** Relaxation kinetics of the different wavelength labeled by the black dotted line in Fig. 4a within 3 ps. **c** PLQY of $(PEA)_2SnI_4$ and PEAI-rich. **d** Normalized band-edge exciton relaxation kinetics of the $(PEA)_2SnI_4$ and PEAI-rich samples under the pump fluence of $2\,\mu J\,cm^{-2}$. **e** Relaxation process of the trapped state excitons of $(PEA)_2SnI_4$ and PEAI-rich under the pump fluence of $10\,\mu J\,cm^{-2}$. **f** Pump fluence-dependent normalized band-edge exciton relaxation kinetics of the $(PEA)_2SnI_4$ sample within 5 ps. **g** Pump fluence-dependent maximum value of the PIA signal at 625 nm of the $(PEA)_2SnI_4$ sample. The pink dotted line represents a linear relationship and the blue solid line represents a one-half power relationship. **h** Pump fluence-dependent normalized band-edge exciton relaxation kinetics of the $(PEA)_2SnI_4$ and PEAI-rich. **i** The PB peak as a function of the pump fluence at 1.25 ps of the $(PEA)_2SnI_4$ sample. The blue data are extracted from the inset, and the red curve is fitting with the band filling theory. The internal illustration is the normalized pump fluence-dependent band-edge bleaching peak with a delay time of 1.25 ps. $n$ is the density of the photoinduced exciton that is proportional to the pump fluence $F$ ($\mu J\,cm^{-2}$). The first component (I) is prominently marked with a light blue background.

independent relaxation characteristics (Fig. 5h) so that the bandgap renormalization process dominates the component I[19,41]. After the process of bandgap renormalization, the band filling effect gradually increases the blueshift of the PB[59], as shown in Supplementary Fig. 16. The change in the position of band-edge PB peak at a delay time of 1.25 ps with the pump fluence is shown in Fig. 5i when the redshift process ends. When the pump fluence was lower than $15\,\mu J\,cm^{-2}$, the position of the PB peak did not change, and the blueshift occurred and was proportional to $n^{2/3}$ as the pump fluence increased due to the band filling effect (Supplementary Note 7)[59]. Additionally, the optical Stark effect can be excluded because the redshift is well beyond the duration of the pump laser pulse[19,44]. So the component I is attributed to the combination of the defect trapping exciton process and the bandgap renormalization process induced by hot excitons, that is, the defect trapping excitons process

plays a leading role at low pump fluence, and the bandgap renormalization process dominates in component I at high pump fluence. The density of defect states is the lowest in $(PEA)_2SnI_4$, so the trapping rate and the proportion of component I in relaxation dynamics are the lowest, which is consistent with our experimental data (Fig. 6a and Supplementary Fig. 14).

The second component (II) is characterized as follows: (a) its lifetime, $\tau_2$, is in the order of hundreds of picoseconds, and (b) its relaxation rate is affected by the temperature, pump fluence, and defect state density (Fig. 6b, c and Supplementary Fig. 18c). The low temperature and high pump fluence can increase the relaxation rate and proportion of component II during the entire relaxation process. In particular, the relaxation rate is noticeably affected by the temperature between 200 and 300 K; however, the relaxation process is independent of temperature in the ranges of

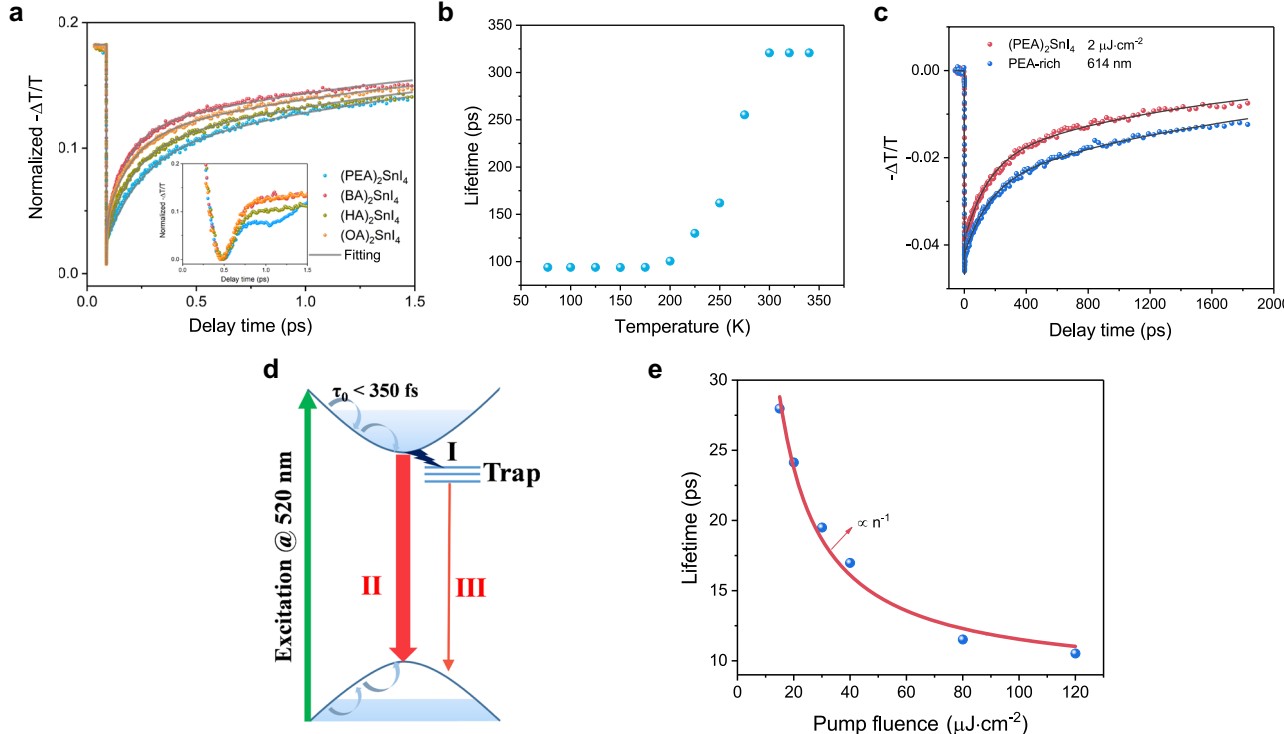

**Fig. 6 The second component (II) of the exciton relaxation kinetics. a** Normalized band-edge GSB relaxation process under the pump fluence of 2 μJ cm$^{-2}$ for the (PEA)$_2$SnI$_4$, (BA)$_2$SnI$_4$, (HA)$_2$SnI$_4$, and (OA)$_2$SnI$_4$ thin films. The internal illustration shows the relaxation process of the four materials within 2 ps. **b** Change in the lifetime of component II in the TA spectra of the (PEA)$_2$SnI$_4$ sample with temperature. **c** Band-edge exciton relaxation process of (PEA)$_2$SnI$_4$ (red represents) and PEAI-rich (blue represents) under the pump fluence of 2 μJ cm$^{-2}$. **d** Schematic diagram of the TA relaxation process with excitation at 520 nm. **e** Relationship between the lifetime (blue represents) of the Auger process and the pump fluence; the red curve is the fitting result of the two-body interaction theory of excitons.

300–340 K and 77–200 K (Fig. 6b). In contrast, the relaxation rate of component III, with a nanosecond lifetime, did not exhibit temperature-dependent characteristics, except that its proportion decreased with the decreasing temperature (Supplementary Fig. 18e). Combined the results of reduced $P_{TE}/P_{FE}$ with those of increased PL intensity of the (PEA)$_2$SnI$_4$ sample as the temperature decreases (Fig. 3k and Supplementary Fig. 17a), we concluded that the proportion of free excitons that undergo band-side radiative recombination increases with decreasing temperature in the exciton relaxation process. Furthermore, compared to (PEA)$_2$SnI$_4$, the occupancy of component I and the relaxation rate of component II of PEAI-rich decreased, which improves PLQY (Fig. 5c, d and Fig. 6c). From the above, it can infer that reduced charged defect scattering can improve the radiative recombination efficiency of free excitons. Despite the interband recombination rate of free carriers increase with decreasing temperature in the 3D perovskites[60], the factors that have a major impact are usually different, and the reason for this may lie in the fact that three-dimensional perovskites have higher ion mobility, high defect tolerance, and very weak interaction between free electrons and holes. Huang et al. revealed that the temperature-dependent time-resolved PL relaxation rate in ultrathin flakes of the mechanically stripped 2D Pb-based perovskite single crystals decreases with temperature. The main reason is that the efficient screening of the Coulomb potential suppresses the scattering of polar optical phonons and charged defects, and the deformation potential scattering by acoustic phonons and nonpolar optical phonons is the dominant factor in exciton relaxation[17]. This result contradicts our time-resolved PL and temperature-dependent TA experimental results obtained under the pump fluence of 2 μJ cm$^{-2}$ to avoid high-order recombination processes

(exciton–exciton annihilation) (Supplementary Figs. 17b, 18a), primarily because the 2D Sn-based perovskites have higher defect states than the 2D Pb-based perovskites, Which confirmed by the PLQY measured under the same experimental conditions (Fig. 3i). Moreover, we studied the polycrystalline thin films, which are more likely to have grain boundaries and defects other than flakes stripped from single crystals. Therefore, component II derived from the band-edge free exciton recombination process is mainly affected by charge defects and this effect can annihilate the radiative recombination of excitons and accelerate the non-radiative relaxation rate[61]. So, the relaxation rate of component II in PEAI-rich with smaller defect state density was slower than that of (PEA)$_2$SnI$_4$, which further confirmed that the effect of deformation potential scattering by charged defects on exciton interband recombination in 2D Sn-based perovskites is weakened by reducing the density of defect states. Besides, component III, with a nanosecond lifetime, is attributed to defect-assisted exciton recombination[32,39,53,62], which involves the radiative recombination induced by relatively shallow extrinsic STE states. Here, the $P_{TE}/P_{FE}$ is approximately 30% at room temperature (Fig. 3k); moreover, the non-radiative recombination processes are induced by deep defects, which make the PLQY extremely low. A schematic diagram of the TA relaxation process is shown in Fig. 6d. By comparing the lifetime and percentage ratio of the components I and II, we can conclude that it is the deformation potential scattering by charged defects, not by the nonpolar optical phonons, that dominates the interband relaxation of excitons and diminishes the PLQY, for the 2D Sn-based perovskite materials. This is in contrast to the observation regarding 2D Pb-based perovskites. Due to the fact that the $\tau_2$ of (PEA)$_2$SnI$_4$ is the longest among those of the four samples at a

low pump fluence at 300 K, (Supplementary Fig. 14 and Fig. 6a), we may conclude that among the four samples, $(PEA)_2SnI_4$ has the lowest defect concentration, which is consistent with the experimental results of SCXRD, UPS, and PLQY.

With the increase in the pump fluence, a proportion of excitons could induce the Auger recombination[48], which is the non-radiative recombination process of the two-body exciton interaction (Supplementary Fig. 18f). By adding the Auger recombination component to exploit the four exponents for global fitting to obtain $\tau_4$, the lifetime of the Auger recombination process decreases with the increase in the pump fluence (Fig. 6e), which corresponds well to the bimolecular recombination model (Supplementary Note 8)[32]. Different organic cations regulate not only the exciton−phonon coupling effect but also the density of defect states in the materials in 2D Sn-based perovskites, more in-depth research work is needed to elucidate the effect of the defect states and exciton−phonon coupling on the AR process. This investigation will help reveal the factor that plays a dominant role in the AR process to improve the PLQY in Sn-based LEDs.

In summary, different cations indicate that the materials have different potential barriers, which affect the optical properties of the materials in the 2D perovskite system. By effectively regulating the kinds of cations ($PEA^+$ vs $BA^+$, $HA^+$, and $OA^+$) and combining the results of temperature-dependent PL spectra and TA spectra, we drew the following conclusions. First, compared with the alkyl chain group samples, reduced structural fluctuations in relatively rigid $(PEA)_2SnI_4$ can improve the surface uniformity, increase crystallization quality, weaken the oxidation of $Sn^{2+}$ to decrease the defect density, prolong the exciton radiative recombination ratio, and increase the PLQY. Moreover, they can reduce the fluctuation of the inorganic layer structure to weaken the scattering effect between excitons and optical phonons, thus reducing the FWHM of the PL and stably maintaining its luminous color in the temperature range of 77–300 K with no structural phase transition lower than 200 K. Second, through the TA experiment, we discovered that it is the deformation potential scattering by charged defects, not by the optical phonons, that dominates the interband excitons relaxation, diminishing the PLQY for the 2D Sn-based perovskite materials. Therefore, it is different from the Pb-based perovskite characterized by fewer defect states, in which the main scattering mechanisms for excitons are the scatterings via deformation potential by acoustic and homopolar optical phonons. The relaxation process of excitons is divided into three components. For the first component (I), with a subpicosecond lifetime, it is attributed to the combination of the defect trapping exciton process and the bandgap renormalization process induced by hot excitons, that is, the defect-trapping exciton process plays a leading role at low pump fluences and a new process appears due to the bandgap renormalization process induced by hot excitons dominates in the process I at the high pump fluences via the pump fluence-dependent, temperature-dependent TA experiments and the stoichiometry engineering of the cations. Meanwhile, we excluded the exciton formation process and optical Stark effect in the first component (I). For the second component (II), with a lifetime of 100 ps, the interband radiation recombination process is mainly affected by the deformation potential scattering by charged defects, not the exciton splitting into free carriers; thus, the relaxation rate increases with the decrease in temperature and defect state density. The third long relaxation component (III), with a nanosecond lifetime, was derived from the defect-assisted exciton recombination process. Because the defect density is the lowest in the $(PEA)_2SnI_4$ samples, the relaxation rate of the second process was the slowest among these samples. Our results reveal the complex contributions of significant materials that improve the luminous efficiency of Sn-based perovskites and provide directional guidance for further improving their luminous properties, that is, a more precise structural design is needed for 2D Sn-based perovskite to

reduce the electron–phonon scattering intensity and reduce $Sn^{2+}$ oxidation.

## Methods

**Synthesis of the four perovskites polycrystalline thin films ($(PEA)_2SnI_4$, $(BA)_2SnI_4$, $(HA)_2SnI_4$, and $(OA)_2SnI_4$).** The glass substrate was cleaned sequentially with detergent, deionized water, ethanol, and isopropanol. Then the substrate was treated with oxygen plasma for 10 min and dried in an argon flow. For the synthesis of $(BA)_2SnI_4$ perovskite film, 0.1 mmol $SnI_2$ and 0.2 mmol BAI were dissolved in 1 ml dimethyl formamidine (DMF): dimethyl sulfoxide (DMSO) (v:v = 4:1) to form the perovskite precursor solution, which was heated and stirred at 70 °C for a few hours before use. Subsequently, the above-mentioned precursor solution was deposited on top of the glass substrate via a spin-coating process at 2500 rpm for 60 s in the argon-filled atmosphere. Then the perovskite film was obtained after thermal annealing at 70 °C for 5 min. The fabrication procedure of $(HA)_2SnI_4$, $(OA)_2SnI_4$, and $(PEA)_2SnI_4$ perovskite thin films is identical to that of $(BA)_2SnI_4$.

**Synthesis of the four tin-based perovskite single crystals.** The single crystals were obtained by the solvent evaporation method. Briefly, for the synthesis of $(BA)_2SnI_4$ single crystals, 0.2 mmol $SnI_2$, and 0.4 mmol BAI were dissolved in 1.2 ml acetonitrile (ACN) to form the precursor solution in a sealed glass bottle. The precursor solution was heated to 75 °C and stirred continuously for about 30 min to achieve complete dissolution. Then, the lid of the glass bottle was removed and the solvent was slowly evaporated at 75 °C for 2.3 h. Finally, the glass bottle was sealed again and the precursor solution was slowly cooled down to room temperature at 0.1 °C min$^{-1}$ to obtain the dark-brown flaky crystals. The fabrication procedure of $(HA)_2SnI_4$, $(OA)_2SnI_4$, and $(PEA)_2SnI_4$ perovskite single crystals is similar to that of $(BA)_2SnI_4$, except that for $(HA)_2SnI_4$, $(OA)_2SnI_4$, and $(PEA)_2SnI_4$, the corresponding precursors were dissolved in 1.1, 1, and 1.6 ml ACN, respectively, and the solvent was evaporated for 2.2, 2, and 2.5 h, respectively.

**Scanning electron microscope (SEM).** The morphologies of the samples were identified by scanning electron microscope (FEI Nova Nano SEM 450).

**Single-crystal X-ray diffraction (SCXRD).** SCXRD was performed using a Bruker D8 Venture diffractometer operating with Mo Kα radiation and equipped with a Triumph monochromator and a Photon100 area detector at 125 and 275 K, respectively. The sample was mounted in a nylon loop using cryo-oil and cooled using a nitrogen flow from an Oxford Cryosystems Cryostream Plus.

Crystallographic data for each of the new/redetermined structures have been deposited with the Cambridge Crystallographic Data Centre. The CCDC Nos. for $(PEA)_2SnI_4$ at 275 K, $(PEA)_2SnI_4$ at 125 K, $(BA)_2SnI_4$ at 275 K, $(BA)_2SnI_4$ at 125 K, $(HA)_2SnI_4$ at 275 K, $(HA)_2SnI_4$ at 125 K, $(OA)_2SnI_4$ at 275 K, and $(OA)_2SnI_4$ at 125 K are 2109461, 2109462, 2109463, 2109464, 2109465, 2109466, 2109467, and 2109468, respectively. These data can be obtained free of charge at www.ccdc.cam.ac.uk/data_request/cif.

**X-ray photoelectron spectroscopy (XPS).** XPS was performed on the polycrystalline thin films prepared on Si substrates using a Thermo Scientific K-Alpha photoelectron spectrometer with Al Kα radiation ($hv = 1486.6$ eV). The peak area and atomic ratio were determined using the XPS data.

**Ultraviolet photoelectron spectroscopy (UPS).** UPS was performed on the polycrystalline thin films prepared on Si substrates by Thermo Scientific Escalab 250Xi system with an $hv = 21.22$ eV under an applied negative bias of 5.0 V and pass energy of 2.0 eV.

**Temperature-dependent photoluminescence (PL) measurement.** For temperature-dependent PL measurement, polycrystalline thin films prepared on quartz substrates were mounted in a cryostat (Janis ST-100) and cooled by liquid nitrogen. The samples were excited by the continuous wave (CW) laser excitation at a wavelength of 473 nm, power density of 2 μJ cm$^{-2}$, and 25 K intervals. Fluorescence is separated by the 150 g/mm grating in the Monochromator SP2500 of the Princeton Instruments. Then, the spectral information was collected by the PIXIS-100BX CCD at −75 °C.

**Photoluminescence quantum yield (PLQY).** PLQY of polycrystalline thin films prepared on quartz substrates was measured using the Edinburgh FLS1000 instrument with an excitation wavelength of 520 nm at 300 K.

**Raman spectra.** Raman spectra of polycrystalline thin films prepared on quartz substrates were obtained with a Raman spectrometer (LHA19120048) using a CW laser (325 nm) as the emission source at room temperature.

**UV-visible (UV-Vis) absorption**. UV-Vis absorption spectra of polycrystalline thin films prepared on quartz substrates were collected by Lambda 950 UV-Vis spectrometer.

**Transient absorption spectrum (TAS)**. Femtosecond transient absorption spectroscopy (fs-TAS) measurements of polycrystalline thin films prepared on quartz substrates were performed using our home-built TAS setup. The frequency-doubled 520 nm output from a Spectra-Physics Spirit laser (350 fs, 1 kHz, 40 µJ/pulse) was used for the pump beam, while a fraction was used for WLC generation using a sapphire crystal. The pump beam was chopped at 500 Hz, and the WLC probe signals were collected using an ultrafast fiber optic spectrometer. The time window of the TAS measurement was 1.6 ns. The samples on quartz substrates were mounted in a cryostat (Janis ST-100) and cooled by liquid nitrogen, 25 K intervals from 340 to 77 K. Schematic diagram of the TA system is shown in Supplementary Fig. 20.

**Time-resolved PL (TRPL)**. TRPL kinetics was detected by HORIBA DeltaFlex ultrafast time-resolved fluorescence spectrometer, where the excitation wavelength was 405 nm at $1 \, \mu J \, cm^{-2}$ and the detection time scale was 40 ns.

**Density functional theory (DFT) calculation**. All calculations in this study were performed with the Vienna ab initio Simulation Package within the frame of DFT[63]. The exchange–correlation interactions of the electron were described via the generalized gradient approximation (GGA) with PBE functional[29], and the projector augmented wave (PAW) method was used to describe the interactions of electron and ion[64]. The Monkhorst–Pack scheme with a $3 \times 3 \times 1$ k-point mesh was used for the integration in the irreducible Brillouin zone. The kinetic energy cut-off of 500 eV was chosen for the plane wave expansion. The lattice parameters and ionic position were fully relaxed, and the total energy was converged within $10^{-5}$ eV per formula unit. The final forces on all ions were less than 0.02/Å.

**Statistics and reproducibility**. For the temperature-dependent PL experiments, temperature-dependent TA experiments, temperature-dependent TRPL experiments, PLQY, and Raman spectra experiments of the four 2D $A_2SnI_4$ ($A = PEA^+$, $BA^+$, $HA^+$, and $OA^+$), we have repeated the experiments more than four times under the same experimental conditions, and each time the results have the same rule with similar experimental results.

## Data availability

The authors declare that the main data supporting the findings of this study are available within the article and its Supplementary Information files. Extra data are available from the corresponding author upon reasonable request.

## Code availability

All custom codes used to generate results in the current study are available from the corresponding author upon reasonable request.

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

## Acknowledgements

The work was supported by the NSFC (Nos. 61975221, 11904375, 12174246, and 61875119), the Strategic Priority Research Program of CAS (Nos. XDB16030700 and XDB43010303), Shanghai Science and Technology International Cooperation Fund (No. 19520710200), the Program for Professor of Special Appointment (Eastern Scholar) at Shanghai Institutions of Higher Learning, Shanghai Rising-Star Program (Grant No. 19QA1404000), Youth Innovation Promotion Association CAS (2016237), the "Chen Guang" project supported by Shanghai Municipal Education Commission and Shanghai Education Development Foundation (Grant No. 18CG63). The author thanks Dr. Chen from Instrumentation and Service Center for Molecular Sciences at Westlake University for supporting the Temperature-dependent steady-state PL spectra measurement and PLQY measurement.

## Author contributions

J.W. and J.L. supervised the execution of the project. T.Z. and J.L. conceived the idea and designed the experiments. T.Z. independently designed and built a TA spectrum testing device. C.Z. prepared perovskite samples. T.Z. carried out the UPS, XPS, SEM, PLQY, SCXRD, Raman spectra, UV/Visible absorption, TRPL, PL spectra, and TA spectra measurement and analysis. X.F. and H.C. contributed to the single crystal testing and structural analysis, J.L. analyzed the SCXRD data. T.Z. wrote and checked the paper. J.L., J.W., N.D., X.C., and L.Z. checked the manuscript. All authors commented on the paper.

## Competing interests

The authors declare no competing interests.
