## [Peer Review File · Nature Communications]

Regulation of the luminescence mechanism of two-dimensional tin halide perovskitesREVIEWER COMMENTS

Reviewer #1 (Remarks to the Author):

In this communication, Zhang et al., explore and decipher the impact of chemical structure of the organic cation on the interband exciton relaxation of layered A_2SnI_4 ($A = \text{PEA}^+$, BA^+ , HA^+ , and OA^+) structures. The authors demonstrate that the exciton-phonon scattering and exciton-defect scattering have different effects on exciton relaxation pathways. The authors also show that the defects can diminish the PL quantum yield for the 2D Sn-based perovskite materials, which is largely different from the Pb-based perovskite characterized by high defect tolerance. Although the authors are dealing with very interesting topic, the current version of this manuscript needs a lot of work before recommending publication in Nature Communications. Here are my major concerns:

- 1) The authors mentioned that "PEA⁺ cation helps the sample to form a better surface morphology without pinholes and clear crystal grains", however, Figure 1 of Supplementary Information clearly shows that HA and OA have better morphology with bigger crystal grains as compared to PEA and BA. The authors should make a clear comment on this.
- 2) The interlayer distance is another key factor in controlling the bandgaps of 2D perovskites. As given in Supplementary Table 1, the interlayer distances are ~16.6 Å for PEA, ~13.8 Å for BA, ~16.4 Å for HA, and ~ 18.8 Å for OA. In other words, BA has the smallest interlayer distance and is expected to have a smaller bandgap. In this case, the detailed density functional theory (DFT) calculations are needed to confirm the trend of bandgaps among these four perovskites. This will also support the descriptions given in lines 258-264.
- 3) The authors assigned second absorption peak at ~520 nm to the charge transfer (CT) transition between the organic spacer and the inorganic layers. It is weird to see the CT state located above the band edge; if it is true, the authors need to further analyze their TA spectra to prove the charge transfer state and clarify whether this CT state becomes nonradiative channel for decreasing the PLQYs of four 2D perovskites. I also think that this feature at 520 nm could be attributed to intraband transitions.
- 4) The UPS data for valence band edges (Supplementary Figure 4b) are quite noisy and it is difficult to obtain accurate values of valence band positions based on the cross lines drawn by hands.
- 5) Although the authors emphasized that PEA cations have a stronger ability to protect Sn²⁺ from oxidation than the organic alkyl chain spacers, the mechanism behind it is completely missing. Moreover, it is not clear whether the oxidation process occurs during the film fabrication or during the measurements exposed to air?
- 6) The low temperature XRD measurements of HA and OA are also needed to confirm the phase transition around 200 K.
- 7) In the TA analysis, the authors divided the TA kinetics into three processes, which are (I) exciton trapped by shallow defects, (II) direct inter-band recombination, and (III) defect-assisted excitons recombination. To justify the processes (I) and (III), I would suggest the authors further treat the 2D perovskite films by using extra organic cations in order to reduce the defect states and then check the TRPL and TA kinetics. Moreover, it might be useful to conduct the PL and TA measurements using slightly below band-edge excitation to understate process (III). Being in this regime, the authors should compare the kinetics at 520- and 610 nm bands. This may help the authors to correctly assign the band at 520 nm (whether it is CT or intraband transition).

Reviewer #2 (Remarks to the Author):

The authors have studied the emission properties of the 2D tin halide perovskites with different organic cations, based on photoluminescence and transient absorption spectroscopy as function of temperature and power excitation. The understanding of the luminescent mechanisms in lead-free 2 hybrid perovskites is important for the applications. Deciphering the influence of the organic cation on the optical properties is particularly important to guide the optimization of the material. The results are interesting and original. However, the clarity of the discussion and the interpretation of these results could be improved. I have the following remarks:

1. Line 161 in the manuscript, the authors claim that the absorption peak at 520 nm has been assigned to charge transfer transition between the organic spacer and the inorganic layers. However, the assumptions on which are based this assignment are not clear. Interestingly, the authors observe a bleaching in transient absorption at this wavelength in addition to the bleaching of the free exciton absorption. I don't think that a similar observation has been made for lead halide 2D perovskites. The authors could provide some comment on that point. Could they resolved the dynamics of the supposed charge transfer transition? Is there any difference between the TA of the charge transfer exciton and the one of the free exciton? It seems surprising that this charge transfer is independent of the choice of the organic cation.

2. Line 177, the authors discuss the influence of the organic cation on the band gap. They highlight the influence of the dielectric constant. However, the influence on the exciton binding energy could be discussed in more details. The dielectric confinement play an important role in the estimation of the exciton binding energy and we can expect large difference between the use of PEA and alkyl-ammonium chains (for example Blancon et al. 10.1038/s41467-018-04659-x).

3. The authors have studied the evolution of the FWHM of the PL emission as function of temperature and extracted values for the coupling with optical phonons (Figure 4, Table 1). The values are given without measurement uncertainty and with a precision which seems really overestimated. In particular, for the compounds based on the cation BA, HA and OA, the authors could not present measurements at temperature below 200 K. For the range of temperature fitted here, the equation 2 in the Supplementary note (which needs a correction, for the Bose-Einstein term, $k_B T$ is missing) gives an almost linear relation as observed on the figure 4b, c and d above 200K. It seems impossible to separate Γ_0 from E_0 properly in this situation. If we compare the figure 4b and 4d, we observe that the data are very similar. However, the results from the fit are very different, with a factor 2 on Γ_0 between OA and BA.

4. Line 408 the authors have extracted the exciton binding energy from an Arrhenius fit of the PL. Some precautions must be taken in the text. This is not a very precise mean to estimate the exciton binding energy. Other mechanisms can explained the thermal quenching of the photoluminescence such as the presence of trap states and the authors have indeed highlighted the presence of trap states in their compounds.

The sentence "Thus, it is a large part of the unseparated excitons that affect the optical properties of the material, despite that only a few excitons split into free carriers, which is consistent with the results of the THz study 51 ." is unclear. What indicates that a few excitons split into free carriers? Additionally, the study 51 refers to lead halide perovskite not tin.

5. The authors assigned the process I of transient absorption measurement to trapping of exciton (line 459) based on its decrease with pump fluence, interpreted as the filling of trap states. However, they also supposed that the trapping is induced by electron-phonon coupling based on the paper of Wu et al. (ref 20) (line 487). The two hypothesis seems conflicting. If the trap states are self-trapped exciton induced by electron-phonon coupling, as supposed by Wu et al., a filling of these traps state is very unlikely. Hence, the trap states observed by the authors is most likely caused by chemical defects in the crystals.

6. The conclusion of the authors is that for tin based 2D perovskite, "it is different from the Pb-based perovskite characterized by high defect tolerance." Line 562
However, the PL quantum yield reported for 2D lead halide perovskite thin films is very low (<1%) (see Yuan et al. Nature Nanotechnology 11 (10) 872—877 (2016) and Duim et al. Advanced Functional Materials, 30 (5) 1907505 (2019) In reality, the PLQY reported in this manuscript for tin based 2D perovskites is much higher than that. The conclusion is not consistent with the results.

7. Line 487 the sentence is unclear : "Moreover, if the component I is caused by the bandgap renormalization process, its maximum value of the PIA should be proportional to $n^{1/2}$."

8. Line 506, the authors write : "From the above, it can be inferred that excitonic contribution to the PL is dominant in the materials and excitons exist stably at room temperature. Thus, the view that excitons split into free carriers can be excluded, although the experimental phenomena are similar". It seems to me that the results are not sufficient to support this conclusion. Why the fact

that the proportion of radiative recombination increases imply that the excitonic contribution to the PL is dominant? The experimental phenomena are similar to what?

9. The third process of the transient absorption is assigned to trapped state exciton radiative recombination. However, the PL quantum yield is relatively low and it seems reasonable to think that the majority of exciton recombine through non-radiative processes, mainly due to trap states.

Response to Reviewer Comments

REVIEWER COMMENTS

Reviewer #1 (Remarks to the Author):

In this communication, Zhang et al., explore and decipher the impact of chemical structure of the organic cation on the interband exciton relaxation of layered A_2SnI_4 ($A = PEA^+$, BA^+ , HA^+ , and OA^+) structures. The authors demonstrate that the exciton-phonon scattering and exciton-defect scattering have different effects on exciton relaxation pathways. The authors also show that the defects can diminish the PL quantum yield for the 2D Sn-based perovskite materials, which is largely different from the Pb-based perovskite characterized by high defect tolerance. Although the authors are dealing with very interesting topic, the current version of this manuscript needs a lot of work before recommending publication in Nature Communications. Here are my major concerns:

We are very grateful to reviewer for spending valuable time making constructive and useful comments. We will address the reviewer's specific points on the following pages. The comments are underlined in black and the content of our response is highlighted in both blue and red, with those highlighted in red representing additions and corrections in the revised manuscript and Supporting Information. The comments are responded to in a point-by-point manner.

1) The authors mentioned that “ PEA^+ cation helps the sample to form a better surface morphology without pinholes and clear crystal grains”, however, Figure 1 of Supplementary Information clearly shows that HA^+ and OA^+ have better morphology with bigger crystal grains as compared to PEA and BA. The authors should make a clear comment on this.

Reply: Many Thanks for the reviewer's comment. We retested SEM and obtained the following results (Supplementary Fig. 1). Liu et al. revealed that when the chain length of the organic spacer cations increases, the grain boundaries become gradually blurry. Especially when the chain length expands to 18 carbon atoms ($CH_3(CH_2)_{17}NH_3^+$), the films present nonuniform morphology and weak light absorption [ACS Energy Lett. 2020, 5, 1422–1429].

The insertion of large molecules of ammonium organic ions, such as alkylammonium chain (BA^+) and phenylethylammonium (PEA^+), into perovskite improves the stability of perovskite under its own hydrophobicity. The existence of large PEA^+ molecules at the boundary of perovskite nanolayers, and the compact pinhole-free films achieved by manipulating the film composition, can block oxygen diffusion into the perovskite lattice [J. Am. Chem. Soc. 2017, 139, 6693–6699].

In the original manuscript,

“PEA⁺ cations help the sample to form a better surface morphology without pinholes and clear crystal grains”

is corrected to (lines 154-157)

“the PEA⁺ cation helps to form large size grains with obvious grain boundaries without pinholes. For BA⁺, large and discontinuous perovskite islands were formed. For HA⁺, small size grains with pinholes were observed. For OA⁺, the top surface became blurry.”

Supplementary Figure 1. The top-view scanning electron microscopy (SEM) images. (a) and (b) $(\text{PEA})_2\text{SnI}_4$, (c) and (d) $(\text{BA})_2\text{SnI}_4$, (e) and (f) $(\text{HA})_2\text{SnI}_4$, and (g) and (h) $(\text{OA})_2\text{SnI}_4$ thin films. (Supporting Information lines 103-107)

2) The interlayer distance is another key factor in controlling the bandgaps of 2D

perovskites. As given in Supplementary Table 1, the interlayer distances are ~16.6 Å for PEA, ~13.8 Å for BA, ~16.4 Å for HA, and ~ 18.8 Å for OA. In other words, BA has the smallest interlayer distance and is expected to have a smaller bandgap. In this case, the detailed density functional theory (DFT) calculations are needed to confirm the trend of bandgaps among these four perovskites. This will also support the descriptions given in lines 258-264.

Reply: For two-dimensional perovskites, the different cations can influence the octahedral tilting angle in the inorganic layer, the length of Sn-I bond, and thus modulate the band gap of the material [J. Phys. Chem. Lett. 2018, 9, 3416–3424, ACS Nano 2018, 12, 3321–3332, and J. Phys. Chem. Lett. 2020, 11, 2955–2964]. It turns out in the series of alkylammonium lead iodide monolayers having from 4 up to 10 (and from 10 up to 18) carbon atoms in the alkyl chain, the increased size of the organic barrier does hardly alter band-edge states and band gaps of the electronic band structure computed for the experimental room temperature structures. The band gap jump between the two sets of chain length can be traced back to a phase transition that induces larger out-of-plane octahedral tilting. The influence of cations on the band gap can be explained more comprehensively by theoretical calculations. Through density functional theory calculations (Supplementary Fig.7) [ACS Nano 2018, 12, 3321–3332 and J. Phys. Chem. Lett. 2020, 11, 2955–2964], we find that the band gap of (PEA)₂SnI₄ is the smallest and (BA)₂SnI₄, (HA)₂SnI₄, (OA)₂SnI₄ are increasing in order, which is consistent with the results we obtained by steady-state spectroscopy. For the alkyl chain group samples, the increase in the length of the alkyl chain increases slightly the electron band gap while the exciton binding energy remains similar. In the meantime, we have revised the relevant contents of the manuscript.

In the original manuscript,

“Hence, the difference of the organic cations in the layered $A_2\text{SnI}_4$ ($A = \text{PEA}^+$, BA^+ , HA^+ , and OA^+) structures is the main factor influencing the composition-dependent bandgap energies. The first is rooted in the difference in the dielectric constant (ϵ) of the organic cationic layer. The relationship between the optical bandgap and the dielectric limiting effect can be described by the following equation where E_0 denotes the energy bandgap without considering the dielectric limitation, whereas ϵ_w and ϵ_b are the dielectric constants of the inorganic layer and the organic layer, respectively. Since the dielectric constant of PEA^+ ($\epsilon_{\text{PEA}} = 3.3$) is smaller than that of BA^+ ($\epsilon_{\text{BA}} = 4.3$), the optical bandgap of $(\text{PEA})_2\text{SnI}_4$ is smaller than that of $(\text{BA})_2\text{SnI}_4$. For the alkyl chain group samples, the increase in the length of the alkyl chain increases slightly the electron band gap while the exciton binding energy remains similar. Based on studies such as the aforementioned, we can determine that the optical bandgap of the four perovskites ($(\text{PEA})_2\text{SnI}_4$, $(\text{BA})_2\text{SnI}_4$, $(\text{HA})_2\text{SnI}_4$, and $(\text{OA})_2\text{SnI}_4$) increases gradually.”

is corrected to

“For two-dimensional perovskites, the different cations can influence the octahedral tilting angle in the inorganic layer, the length of Sn-I bond, and thus modulate the band gap of the material. The influence of cations on the band gap can be explained more comprehensively by theoretical calculations. Through density

functional theory calculations (Supplementary Fig.7) [ACS Nano 2018, 12, 3321–3332 and J. Phys. Chem. Lett. 2020, 11, 2955–2964], we find that the band gap of (PEA)₂SnI₄ is the smallest and (BA)₂SnI₄, (HA)₂SnI₄, (OA)₂SnI₄ are increasing in order, which is consistent with the results we obtained by steady-state spectroscopy. For the alkyl chain group samples, the increase in the length of the alkyl chain increases slightly the electron band gap while the exciton binding energy remains similar.” (lines 213-222)

However, for the A₂SnI₄ structure material, the VBM mainly comprised the 5s orbital of Sn and the 5p orbital of I, and the conduction band minimum mainly comprised the empty 5p orbital of Sn. A decrease in the lattice constant strengthens the interaction between the 5s orbitals of Sn and the 5p orbitals of I, which results in an increase in the VB width and an increase in the VB energy.(lines 295-299)

Supplementary Figure.7 Energy band diagram calculated by density functional theory (DFT) calculations.(a) Band Gap of (PAE)₂SnI₄ is 1.53 eV. (b) Band Gap of (BA)₂SnI₄ is 1.852 eV. (c) Band Gap of (HA)₂SnI₄ is 1.998 eV (Supporting Information lines 147-150)

3) The authors assigned second absorption peak at ~520 nm to the charge transfer (CT) transition between the organic spacer and the inorganic layers. It is weird to see the CT state located above the band edge; if it is true, the authors need to further analyze their TA spectra to prove the charge transfer state and clarify whether this CT state becomes nonradiative channel for decreasing the PLQYs of four 2D perovskites. I also think that this feature at 520 nm could be attributed to intraband transitions.

Reply: Many thanks to the reviewers for their comments, which made us realize that we incorrectly attributed the absorption peak at ~520 nm to the charge transfer (CT) transition between the organic spacer and the inorganic layers, and we attributed the absorption peak at ~520 nm to the intraband transition of SnI₄ inorganic layers by

combining literature study with the results of pump energy-dependent transient absorption (TA) spectroscopy. Since $A_2\text{SnI}_4$ perovskite ($A = \text{PEA}^+, \text{BA}^+, \text{HA}^+, \text{and } \text{OA}^+$) have similar linear absorption and TA spectra characteristics, we selected $(\text{PEA})_2\text{SnI}_4$ as a representative for an explanation. The details are shown below:

In two-dimensional layered halide organic perovskites (LHOPs), the perovskite layer is the dominant component of band-edge absorption. Therefore, a way to demonstrate energy transfer would be the observation of triplet emission from the organic spacer layer. [Phys. Rev. Materials 2018, 2, 105406., J. Am. Chem. Soc. 2018, 140, 7313–7323.] Some LHOPs have demonstrated the ability to induce energy transfer from perovskite layer exciton states to low energy spin-triplet exciton states in the organic layer. [Chem. Phys. Lett. 1999, 303, 157–164., Chem. Phys. Lett. 1999, 307, 373–378., and Phys. Rev. Lett. 2008, 100, 257401.] When the lowest excitation energy (E) of the exciton in the perovskite layer aligns with the first triplet (T_1) excitation energy of the exciton in the organic layer, charge transfer from the perovskite to the organic layer may occur. After transfer, the T_1 excitation energy in the organic layer relaxes to a lower T_1^* energy due to enhanced short-range atomic deformation, thus reaching optimal triplet molecular geometry [Nano Lett. 2019, 19, 8732–8740]. Through theoretical calculations, Neukirch et al. systematically studied organic spacer and perovskite layer pairings for possible transfer of the Wannier excitons from the inorganic perovskite lattice to spin-triplet Frenkel excitons located on the organic cations and successfully identify ten organic spacer candidates for possible pairing with perovskite layers of specific halide composition to achieve triplet light emission across the visible energy range. From their calculations, it is clear that the T_1 energy of PEA^+ remains in a narrow range between 4.43 and 4.46 eV, which is greater than the lowest optical excitation peak energy (2.5 eV) of perovskite layer in the $(\text{PEA})_2\text{PbI}_4$. The absorption spectrum shows that the lowest optical excitation peak energy of SnI_4 in $(\text{PEA})_2\text{SnI}_4$ is smaller than that of PbI_4 in $(\text{PEA})_2\text{PbI}_4$ (2.02 eV vs 2.5 eV), so we can conclude that the exciton in SnI_4 cannot be transferred to organic cation PEA^+ . For the alkylammonium chains, it can not be excited by a photon in the visible region [Solid State Commun. 1989, 69, 933-936], so we can also conclude that the exciton in SnI_4 cannot be transferred to organic cation the alkylammonium chains.

In the pump energy-dependent TA spectroscopy experiments, where we used the wavelength of a pump at 400 nm, the wavelength of the pump at 613 nm with resonant band-edge absorption, and the wavelength of the pump at 630 nm below the band gap, we found the existence of photobleaching peaks at ~523 nm (PB1) and ~613 nm (PB2) in the TA spectra of these three different pump energies (Supplementary Fig. 2.). The relaxation kinetics of $A_2\text{SnI}_4$ ($A = \text{PEA}^+, \text{BA}^+, \text{HA}^+, \text{and } \text{OA}^+$) obtain by low pump fluence have been fitted globally with three components (Figure 6). We find that PB1 has the same relaxation characteristics and lifetime with PB2, and hence it is not consistent with the occurrence of the CT transition. In Supplementary Fig. 2, the two PB peaks are generated almost simultaneously when excited at 400 nm. However, the PB1 peak first reaches the maximum and then decreases, and the PB2 peak reaches a maximum with a delay of about 0.1 ps

compared to the PB1 peak, which is more consistent with the intraband transitions [Science 2013, 342, 344-347].

Combined with the above analysis, the PB1 peak is not the charge transfer transition between the organic spacer and the inorganic layers, but an intraband transitions process in the perovskite layer in the $(\text{PEA})_2\text{SnI}_4$.

Therefore, we revised the original incorrect description in the manuscript as follows:

In the original manuscript,

“the second peak at 520 nm (2.38 eV) was assigned to the charge transfer transition between the organic spacer cations and the inorganic layers”

is corrected to

“The second peak at 520 nm (2.38 eV) was assigned to the intraband transitions process in perovskite layer rather than the charge transfer transition between the organic spacer cations and the inorganic layers in the $(\text{PEA})_2\text{SnI}_4$ [Nano Lett. 2019, 19, 8732–8740, Solid State Commun. 1989, 69, 933-936, and Science 2013, 342, 344-347]. It is mainly due to the fact that the first excitation energies of PEA^+ , BA^+ , HA^+ , and OA^+ cations lie in the UV energy range [Nano Lett. 2019, 19, 8732–8740 and Solid State Commun. 1989, 69, 933-936], which is much higher than the lowest optical excitation peak energy of the perovskite layer and the energy of photobleaching peaks at ~523 nm. As a result, the excitons in the inorganic perovskite lattice can not transfer to spin-triplet Frenkel excitons states located on the organic cation [Nano Lett. 2019, 19, 8732–8740]. In the pump energy-dependent transient absorption spectroscopy experiments (Supplementary Fig. 2), we find that the photobleaching peaks at ~523 nm (PB1) and ~613 nm (PB2) have the same characteristics and lifetime of the relaxation decay process, and no new bleaching peaks appear during the decay relaxation of PB1. Despite the PB2 peak reaches a maximum with a delay of about 0.1 ps compared to the PB1 peak excited at 400 nm in Supplementary Fig. 3, which is more consistent with the intraband transitions while not consistent with the occurrence of the CT transition. [Science 2013, 342, 344-347].” (lines 175-189)

Supplementary Figure 2. Pump energy-dependent TA spectra of $(\text{PEA})_2\text{SnI}_4$. 2D pseudocolor TA spectra were obtained by pumping at a wavelength of (a) 400 nm, (b) 613 nm, and (c) 630 nm. TA spectra at different delay times with pumping at (d) 400 nm, (e) 613 nm, and (f) 630 nm. Relaxation kinetics at different wavelengths with pumping at (g) 400 nm, (h) 613 nm, and (i) 630 nm. (Supporting Information lines 109-113)

Supplementary Figure 3. Pump energy-dependent TA spectra of $(\text{PEA})_2\text{SnI}_4$ within 5 ps. 2D pseudocolor TA spectra obtained by pumping at a wavelength of (a) 400 nm, (b) 613 nm, and (c) 630 nm. Relaxation kinetics at different wavelengths with pumping at (d) 400 nm, (e) 613 nm, and (f) 630 nm. (g) TA spectra at different delay times by pumping at a wavelength of 400 nm. (h) Relaxation kinetics at different wavelengths by pumping at a wavelength of 400 nm. (Supporting Information lines 114-120)

4) The UPS data for valence band edges (Supplementary Figure 4b) are quite noisy and it is difficult to obtain accurate values of valence band positions based on the cross lines drawn by hands.

Reply: Thanks for the reviewer's comment to help us improve the reliability of our UPS experimental results. We optimized the UPS measurement conditions to improve the signal-to-noise ratio of valence band edges and increase the signal intensity to make the UPS data for valence band edges more reliable as shown in Supplementary Figure 8. At the same time, we revised the information in the manuscript as follows: In the original manuscript,

"the energy differences between the top of the VBs and the Fermi levels (EF) are 1.62, 0.9 0.95, and 0.87 eV for $(\text{PEA})_2\text{SnI}_4$, $(\text{BA})_2\text{SnI}_4$, $(\text{HA})_2\text{SnI}_4$, and $(\text{OA})_2\text{SnI}_4$ samples, respectively."

is corrected to

“the energy differences between the top of the VBs and the Fermi levels (EF) are 1.01, 0.66, 0.77, and 0.53 eV for (PEA)₂SnI₄, (BA)₂SnI₄, (HA)₂SnI₄, and (OA)₂SnI₄ samples, respectively.” (lines 258-259)

Supplementary Figure 8. UPS spectra of the different perovskites. (a) The cutoff region, (b) VB edge region, and (c) the derived energy band diagram of the different perovskite films. (Supporting Information lines 153-156)

5) Although the authors emphasized that PEA cations have a stronger ability to protect Sn²⁺ from oxidation than the organic alkyl chain spacers, the mechanism behind it is completely missing. Moreover, it is not clear whether the oxidation process occurs during the film fabrication or during the measurements exposed to air?

Reply: Thanks for the reviewer's comment. Oxygen and water molecules in the environment are the main sources for the decomposition of perovskite crystals. The insertion of large molecules of ammonium organic ions, such as alkyl-ammonium chain (BA⁺) and phenylethylammonium (PEA⁺), into perovskite improves the stability of perovskite by virtue of its own hydrophobicity. The main reason is that the large molecules of ammonium organic ions are beneficial for the formation of the compact pinhole-free films and block moisture ingress at the boundaries of perovskite

nanolayers [J. Am. Chem. Soc. 2017, 139, 6693–6699]. The PEA^+ also has higher intrinsic thermodynamic stability for Sn perovskites with respect to the oxidation disproportionation channel. Angelis et al. investigated the beneficial effects of large cations (BA^+ and PEA^+) on the tin stability at the surface and shown that large cation dipoles of the 2D perovskites modulate tin oxidation potential by hindering the formation of tin vacancies and the degradation of the material. BA^+ ions increased the defect formation energy of Sn^{4+} by 0.33 eV, while PEA^+ could increase the defect formation energy of Sn^{4+} by 0.6 eV [J. Phys. Chem. C 2021, 125, 10901-10908]. Therefore, compared to BA^+ , PEA^+ effectively hinders the formation of tin vacancies and tin oxidation. Although the samples are prepared in a glove box, which is almost a nitrogen environment, there is still a trace amount of oxygen that allows the Sn^{2+} oxidize to Sn^{4+} . Therefore, we have added to the manuscript to further explain that PEA^+ cations have a stronger ability to protect Sn^{2+} from oxidation than the organic alkyl chain spacers, as follows.

“The main reason is that the large molecules of ammonium organic ions are beneficial for the formation of the compact pinhole-free films and block moisture ingress at the boundaries of perovskite nanolayers [J. Am. Chem. Soc. 2017, 139, 6693–6699]. BA^+ ions increased the defect formation energy of Sn^{4+} by 0.33 eV, while PEA^+ could increase the defect formation energy of Sn^{4+} by 0.6 eV [J. Phys. Chem. C 2021, 125, 10901-10908]. Therefore, compared to BA^+ , PEA^+ effectively hinders the formation of tin vacancies and tin oxidation.” (lines 248-253)

6) The low temperature XRD measurements of HA^+ and OA^+ are also needed to confirm the phase transition around 200 K.

Reply: Thanks for the reviewer’s comment. We have added the low temperature XRD measurements of HA^+ and OA^+ , and further discussed the phase transition behaviors. We have determined the crystal structure of BA^- , HA^- , OA^- and PEA^- -perovskites by single-crystal X-ray diffraction at both 275 and 125 K to study the phase transition behaviors (CCDC Nos. 2109461-2109468). From the structural analysis, due to the contraction effects in the cooling process, the unit cell of $(\text{PEA})_2\text{SnI}_4$ shrunk, leading to the decreasing of all a , b and c axis, without the existence of phase transition in the measured temperature range (Supplementary Table 1). The unit cell of $(\text{PEA})_2\text{SnI}_4$ was evolved from a space group of $P-1$, with lattice parameters of $a = 8.6785(12) \text{ \AA}$, $b = 8.6808(12) \text{ \AA}$, $c = 32.800(5) \text{ \AA}$, $\alpha = 84.715(4)$, $\beta = 84.742(4)$ and $\gamma = 89.618(4)$ to space group of $P-1$, with lattice parameters of $a = 8.6344(7) \text{ \AA}$, $b = 8.6468(8) \text{ \AA}$, $c = 32.264(3) \text{ \AA}$, $\alpha = 85.261(3)$, $\beta = 85.123(3)$ and $\gamma = 89.512(3)$. However, for the alkyl-ammonium chain structures, obvious phase transitions were observed. The unit cell of $(\text{BA})_2\text{SnI}_4$ was evolved from a space group of Pbca , with lattice parameters of $a = 8.8378(13) \text{ \AA}$, $b = 8.6456(13) \text{ \AA}$ and $c = 27.587(4) \text{ \AA}$ to space group of Pbca , with lattice parameters of $a = 8.475(3) \text{ \AA}$, $b = 8.895(3) \text{ \AA}$, and $c = 26.135(9) \text{ \AA}$. The orthorhombic–orthorhombic phase transition observed here is a first-order solid–solid phase transition. While for $(\text{HA})_2\text{SnI}_4$ and $(\text{OA})_2\text{SnI}_4$, orthorhombic–monoclinic phase transitions were involved. The unit cell of $(\text{HA})_2\text{SnI}_4$ was evolved from a space

group of Pbc_a, with lattice parameters of $a = 8.8685(4)$ Å, $b = 8.6172(3)$ Å and $c = 32.7079(15)$ Å to space group of P2₁/c, with lattice parameters of $a = 16.1466(13)$ Å, $b = 8.8416(8)$ Å, $c = 8.6125(6)$ Å, and $\beta = 92.126(7)$. The unit cell of (OA)₂SnI₄ was evolved from a space group of P2₁2₁2₁, with lattice parameters of $a = 8.6303(11)$ Å, $b = 37.562(5)$ Å and $c = 8.9275(11)$ Å to space group of P2₁/c, with lattice parameters of $a = 18.729(8)$ Å, $b = 8.877(4)$ Å, $c = 8.397(4)$ Å, and $\beta = 96.707(10)$. (lines 137-152 and lines 322-338)

7) In the TA analysis, the authors divided the TA kinetics into three processes, which are (I) exciton trapped by shallow defects, (II) direct inter-band recombination, and (III) defect-assisted excitons recombination. To justify the processes (I) and (III), I would suggest the authors further treat the 2D perovskite films by using extra organic cations in order to reduce the defect states and then check the TRPL and TA kinetics. Moreover, it might be useful to conduct the PL and TA measurements using slightly below band-edge excitation to understudy process (III). Being in this regime, the authors should compare the kinetics at 520- and 610 nm bands. This may help the authors to correctly assign the band at 520 nm (whether it is CT or intraband transition).

Reply: Many thanks to the reviewer for their critical and high-quality comments. In order to answer the origin of this trap state, we have re-done systematically the temperature-dependent, excitation-fluence-dependent, and excitation-energy-dependent transient absorption experiments and temperature-dependent PL experiments, combined with a more thorough study of the literature. The following conclusions have been obtained as shown: The first fast process I of transient absorption measurement is attributed to the combination of the defects trapping excitons process and the band-gap renormalization process induced by hot excitons, i.e., the defects trapping excitons process play a leading role at low excitation fluence and as the excitation fluence increases, the bandgap renormalization process induced by hot excitons dominates in the process I. The main reasons are as follows.

(1) This trap state in the process I almost is the chemical defect state in the material and not the intrinsic self-trapped exciton (STE) state caused by the exciton-phonon coupling. There are three main types of exciton trapping state, namely the intrinsic STE state, defect trapping state, and the extrinsic STE state. Intrinsic STE state is a transient defect that forms in the excited state where photogenerated charge carriers are stabilized through large lattice distortions driven by strong electron-phonon coupling [Acc. Chem. Res. 2018, 51, 3, 619–627]. Luminescence from the STE state normally gives rise to the broadband and large Stokes-shifted PL. Intrinsic STE state formation is found to have a strong dependence on the dimensionality of the system [Phys. Rev. B: Condens. Matter Mater. Phys. 1993, 47, 6060–6064. and Phys. Rev. Lett. 1976, 36, 323–326.]. There is no potential barrier separating the free exciton and STE in the one-dimensional materials, so the STE states forms promptly within the

subpicosecond time scales from the free exciton [Phys. Rev. Lett. 1998, 81, 417–420, Angew. Chem. Int. Ed. 2019, 58, 2278–2283, and J. Phys.: Condens. Matter 2013, 25, 144204.]. In the three-dimensional systems, intrinsic STE formation usually takes the ns magnitude of a long time to form due to the existence of a large potential barrier for trapping [J. Phys. Soc. Jpn. 1983, 52, 4277–4282]. While the two-dimensional system is a marginal case with low self-trapping activation energy [J. Am. Chem. Soc. 2019, 141, 12619–12623, J. Am. Chem. Soc. 2014, 136, 13154–13157, and J. Phys. Chem. Lett. 2016, 7, 2258–2263]. For the intrinsic STE state, with the temperature decreases, the stronger luminescence from the STE state, the band edge exciton luminescence intensity decreases. This is mainly because at low temperatures, the thermal activation of the detrapping process can not meet the requirements of the detrapping barrier, and the self-trapped excitons can not return to the band edge. Besides, the lattice distortion caused by electron-phonon coupling provides for homogeneous emission broadening [J. Am. Chem. Soc. 2014, 136, 13154–13157]. Defect trapping generally refers to the direct trap process of excitons by permanent defects in the lattice, without taking into account the effect of distortion of the lattice due to the optical excitation. And the emission involving defects may show sublinear behavior if the limited number of defect states become saturated, however, for intrinsic STE states emission, it has a linear relationship. [Phys. Rev. B: Condens. Matter Mater. Phys. 1992, 45, 8989–8994 and J. Am. Chem. Soc. 2014, 136, 13154–13157]. Since the lattice of two-dimensional perovskites is relatively soft and defect state density is relatively high, the intrinsic STE is often influenced by permanent defects, i.e. extrinsic STE. Self-trapping is influenced by the local heterogeneity of permanent defects lattice, in which it will sink to a different trapping depth. Extrinsic STE leads to the inhomogeneous nature of the STE states and their radiative and nonradiative decay compared to the case for intrinsic STE [Acc. Chem. Res. 2018, 51, 3, 619–627, J. Phys. Chem. Lett. 2016, 7, 2258–2263, and Nat. Commun. 2020, 11, 2344.]. Using transient absorption spectroscopy to study the dynamics of the broad emission spectrum below the band gap, it is able to distinguish the intrinsic STE and defect trapping. There is a broad photoinduced absorption at energies below the optical gap for the intrinsic STE states, consistent with the formation of transient, light-induced, trap states [J. Phys. Chem. Lett. 2016, 7, 2258–2263 and Acc. Chem. Res. 2018, 51, 3, 619–627]. In contrast, permanent trap states exhibit below-exciton bleaching features owing to filling in 2D perovskites [J. Am. Chem. Soc. 2015, 137, 2089–2096].

In the previous section of the manuscript, we demonstrated that excitonic contribution to the PL is dominant in the materials by excitation fluence-dependent integral PL in $(\text{PEA})_2\text{SnI}_4$, $(\text{BA})_2\text{SnI}_4$, $(\text{HA})_2\text{SnI}_4$, and $(\text{OA})_2\text{SnI}_4$ materials. Due to the transient and linear absorption spectra of the $(\text{PEA})_2\text{SnI}_4$, $(\text{BA})_2\text{SnI}_4$, $(\text{HA})_2\text{SnI}_4$, and $(\text{OA})_2\text{SnI}_4$ samples have similar characteristics, and $(\text{PEA})_2\text{SnI}_4$ has no phase transition in the temperature range of 300K–77K, the treatment of the relaxation process of $(\text{PEA})_2\text{SnI}_4$ is used as a

representation obtain a more detailed elucidation. In the temperature-dependent PL experiment, using the multi-peak fitting methods mentioned earlier in the manuscript, the ratio of the PL percentage of the free excitons (P_{FE}) to that of the trap state excitons (P_{TE}) below the band gap decrease with decreasing temperature (Supplementary Fig. 16a), which is obviously opposite to the feature of the intrinsic STE states emission. More precisely, it may be the emission of the extrinsic STE states [J. Phys. Chem. Lett. 2016, 7, 2258–2263 and Nat. Commun. 2020, 11, 2344]. The PL of the four materials ((PEA)₂SnI₄, (BA)₂SnI₄, (HA)₂SnI₄, and (OA)₂SnI₄) shows asymmetry at 77 K (Supplementary Fig. 11), where PL peaks under the band gap exist in a wide range of trailing feature. This trailing spectral width is more and more serious in (BA)₂SnI₄, (HA)₂SnI₄, and (OA)₂SnI₄, especially in (HA)₂SnI₄, and (OA)₂SnI₄ at 77 K when the phase transition causes PL blue shift with PL peak still existing at 640-750 nm, which is more obvious in (OA)₂SnI₄. This broad-spectrum PL trailing phenomenon can be explained by the extrinsic STE effect. Deschler et al. indicated that the broad emission below the optical gap seen at low temperatures in <001> oriented 2D-perovskite materials was due to the light-induced formation of localized trap states, associated with interstitial iodide and iodide Frenkel defects that act as color centers in the crystal [J. Am. Chem. Soc. 2017, 139, 18632-18639]. Besides, Loi et al also highlighted that the extrinsic origin of their broad band emission in-gap states in the crystal bulk was responsible for the broad emission [Nat. Commun. 2020, 11, 2344]. There is a broad (640-750 nm) and weak bleach feature below the optical gap, consistent with the defect state being filled in the transient absorption spectrum for the four materials (Fig. 5). The process of filling bleaching of these defect states is the same synchronous step as the first process I of band-edge exciton relaxation (Fig. 6b) so that the band-edge excitons are transferred to the in-gap defect states. The electron-phonon coupling effect with defect state trapping could lead to the extrinsic STE in the four materials. The exciton-phonon coupling of (OA)₂SnI₄ is the strongest among the four materials, so there is a relatively wide PL trailing below the band gap, and this PL trailing is more pronounced at 77 K (Supplementary Fig. 11).

To further demonstrate the fact that excitons are trapped by chemical defects, we applied the stoichiometry engineering of the cations where the PEA:I:SnI₂ ratio is 2.6:1 in (PEA)₂SnI₄ (PEAI-rich) [Nat. Commun. 2020, 11, 2344, and Adv. Funct. Mater. 2020, 30, 1907505]. We reduce the defect density in PEA-rich sample to improve the PLQY of the PEA-rich sample to 2.2% (Supplementary Fig. 11b). We found that the occupancy ratio and relaxation rate of the first relaxation process I of the band-edge exciton relaxation processes in PEA-rich were significantly lower than those of (PEA)₂SnI₄ sample (Fig. 6c). In addition, at high excitation intensities, the bleaching signal intensity at 676 nm is weaker and the relaxation rate is slower for PEA-rich samples compared to (PEA)₂SnI₄. (Supplementary Fig. 16c) These results from the transient absorption spectrum provide more evidence that the exciton capture process of the first process I is the defect trapping process [J. Am. Chem. Soc. 2015, 137, 2089–2096].

Therefore, combining the above results, we attribute the first relaxation process mainly to the defect trapping process.

(2) In the transient absorption spectrum (Fig. 5), the band-edge exciton (~614 nm) shows a photobleaching signal attributed to state filling, i.e., the presence of band gap excitons generated by the pump pulse blocking of the optical transition induced by the probe pulse. Within 1 ps, we observe a redshift of the photobleaching peak of the band-edge exciton leading to photoinduced absorption, which is attributed to the bandgap renormalization caused by the hot excitons, which can also cause process I. Assuming an exciton Bohr radius of 1 nm [Nat. Commun. 2020, 11, 664.], the exciton saturation density is simply estimated to be 10^{14} cm^{-2} . In the previous manuscript (Fig. 6d), the saturation of process I occur at high excitation fluence of $40 \mu\text{J}/\text{cm}^2$ ($1.1 \times 10^{14}/\text{cm}^2$). This excitation fluence makes it possible for the exciton to fission and for the material to degenerate at high excitation intensities. Therefore, the previous conclusion is incorrect. Therefore, we retested the excitation intensity-dependent transient absorption spectrum experiment and found that process I does not be a little saturated until the excitation intensity is near the saturation density. To further clearly distinguish which plays a role in the first relaxation process between the defect states trap exciton process and bandgap renormalization process induced by the hot exciton, we further research the fluence-dependent transient absorption spectra of the PEAI-rich and $(\text{PEA})_2\text{SnI}_4$.

(3) In the excited intensity-dependent transient absorption spectra of the PEAI-rich and $(\text{PEA})_2\text{SnI}_4$,

We find that the first relaxation process is more significantly affected by the defect density of states at weak excitation fluence, approximately no more than $10 \mu\text{J}/\text{cm}^2$ (Fig. 6c). With further increase of the excitation fluence (Supplementary Fig. 16d), the first process appears to be the defect state density-independent, the excitation intensity-independent, temperature-independent relaxation process, so that the bandgap renormalization process dominates the process I.

In summary, the first fast process I of transient absorption measurement is attributed to the combination of the defects trapping excitons process and the band-gap renormalization process induced by hot excitons, i.e., the defects trapping excitons process play a leading role at low excitation fluence and as the excitation fluence increases, the bandgap renormalization process induced by hot excitons dominates in the process I. As a result, we revised the manuscript content as follows.

In the original manuscript:

“When the pump fluence is lower than $15 \mu\text{J cm}^{-2}$, the proportion and relaxation rate of the I process remain basically unchanged. However, with the further increase in the pump fluence, the relaxation rate and the proportion of the I process decrease, as shown in Fig. 7(c). This may occur because the photogenerated excitons fill the shallow trap state in different degrees and reduce the trapping rate. For the trapping process, it should involve the transfer of photogenerated carriers between energy

levels, indicated by the redshift of the bleaching peak in the spectra. To prove this, the relaxation kinetics of different wavelengths in Fig. 7(d) show that the completion of the band edge ground state bleaching (614.8 nm) relaxation process during the delay time range of 0.48–1.14 ps is synchronized with the relaxation process of the PIA signal (624.1 nm), changing from positive PIA to negative bleaching and reaching the maximum. This indicates that the photogenerated excitons undergo a transfer from the band edge energy level to the shallow trap energy level, where the radiative recombination of excitons subsequently occurs, inducing a change in the relaxation kinetics of 624.1 nm from positive to negative absorption. Thus, combining all the above characteristics, component I is attributed to the process of exciton trapping by the shallow trap state, and the trap rate is independent of the temperature but related to the carrier density. However, the bandgap renormalization effect is the result of the competition between the reduction of the electron bandgap, caused by the exchange-correlation potential, which leads to the redshift of the bleaching peak, and the reduction of the exciton binding energy, caused by the dielectric screening effect, which leads to the blueshift of the bleaching peak, of the photogenerated carriers. Hence, this effect should show that the bleaching peak undergoes a redshift–blueshift transition, after which the band filling effect increases the bandgap blueshift gradually. Additionally, the redshift caused by this physical effect should increase as the excitation intensity increases; however, the results are not consistent with this law as they indicate that the redshift initially remains unchanged in the low excitation intensity range below $15 \mu\text{J cm}^{-2}$ and subsequently decreases until disappearing as the excitation intensity further increases to greater than $15 \mu\text{J cm}^{-2}$, as shown in Supplementary Fig. 10. The change in the band edge bleaching peak position at a delay time of 1.25 ps when the redshift process ends with the excitation intensity is observed (Fig. 8(a)). When the excitation intensity is lower than $15 \mu\text{J cm}^{-2}$, the bleaching peak position does not change, and the blueshift occurs and becomes more noticeable as the excitation intensity increases because of the band filling effect (Supplementary Note 3). The blueshift is proportional to $n^{2/3}$, as shown in Fig. 8(a). Moreover, if the component I is caused by the bandgap renormalization process, its maximum value of the PIA should be proportional to $n^{1/2}$. As shown in Fig. 7(e), however, it does not conform to this law. Thus, the bandgap renormalization has little effect on the (I) process, which may play a role in the process of bleaching peak generation. This is mainly because the laser pump light pulse width of 350 fs is compared with the generation time of the bleaching peak, i.e., the processes of photogenerated carrier thermalization and exciton formation. Additionally, the optical Stark effect can be excluded because the redshift mainly increases with the excitation intensity. Since the trap state in 2D perovskite materials is induced by electron-phonon coupling and has weak optical transition intensities, it can be seen from the above that the density of trapped states is the lowest in the $(\text{PEA})_2\text{SnI}_4$ sample, so the trapping rate of excitons and the proportion in relaxation dynamics are the lowest (Fig. 6 and Fig. 7(f)), which is consistent with our experimental data.”

was revised to

“For the trap states in two-dimensional perovskites, we need to define the nature of

the trap states, i.e., whether they are intrinsic self-trapped exciton (STE) state, defect trapping state, or the extrinsic STE state. In the temperature-dependent PL experiment, using the multi-peak fitting methods mentioned in the previous content, the ratio of the PL percentage of the free excitons (P_{FE}) to that of the trap state excitons (P_{TE}) below the band gap decreases with decreasing temperature (Supplementary Fig. 16a), which is obviously opposite to the feature of the intrinsic STE states emission, in which the stronger luminescence from the intrinsic STE state and the band edge exciton luminescence intensity decreases with the temperature decreases, mainly because the thermal activation of the detrapping process can not meet requirements of the detrapping barrier and the self-trapped excitons can not return to the band edge at low temperatures [Nat. Commun. 2020, 11, 2344, Chem. Rev. 2019, 119, 3104–3139, J. Am. Chem. Soc. 2019, 141, 12619–12623, and Acc. Chem. Res. 2018, 51, 619–627]. More precisely, this trap state excitons below the band gap may be the emission of the extrinsic STE states, that is intrinsic Self-trapping is influenced by the local heterogeneity of permanent defects lattice to get the different trapping depth [Nat. Commun. 2020, 11, 2344, J. Phys. Chem. Lett 2016, 7, 2258–2263, and J. Am. Chem. Soc. 2017, 139, 18632–18639]. There is a broad (640–750 nm) and weak bleaching feature below the optical gap in the transient absorption spectrum for four materials (Fig. 5) also consistent with the permanent defect states being filled not the intrinsic STE state featured by a broad PIA at energies below the optical gap consistent with characteristics of the formation of transient light-induced trap states [J. Phys. Chem. Lett. 2016, 7, 2258–2263, J. Am. Chem. Soc. 2015, 137, 2089–2096, and Acc. Chem. Res. 2018, 51, 3, 619–627]. The relaxation kinetics of different wavelengths show that the process of filling bleaching of these defect states (663.4 nm) and the relaxation process of the PIA signal (624.1 nm of PL center) changing from positive PIA to negative bleaching and reaching the maximum are the same synchronous step as the process I of the band-edge exciton (614.8 nm) relaxation (Fig. 6b), revealing that the band-edge excitons are trapped to the in-gap defect states. To further demonstrate the fact that excitons are trapped by chemical defects, we applied the stoichiometry engineering of the cations where the PEA:I:SnI₂ ratio is 2.6:1 in (PEA)₂SnI₄ (PEAI-rich) [Nat. Commun. 2020, 11, 2344, and Adv. Funct. Mater. 2020, 30, 1907505], we reduce the defect density in PEA-rich to improve the PLQY of the PEA-rich to 2.2% (Supplementary Fig. 16b). We found that the occupancy ratio and relaxation rate of the first relaxation process I of the band-edge exciton relaxation processes in PEA-rich were significantly lower than those of (PEA)₂SnI₄ sample (Fig. 6c). In addition, at high excitation intensities, the bleaching signal intensity at 676 nm is weaker and the relaxation rate is slower for PEA-rich compared to (PEA)₂SnI₄ (Supplementary Fig. 16c). In summary, the process I contains the process of defect states trapping exciton. In particular, the trailing spectral width of PL is more and more serious in (BA)₂SnI₄, (HA)₂SnI₄, and (OA)₂SnI₄, especially in (HA)₂SnI₄, and (OA)₂SnI₄ at 77 K when the phase transition causes blueshift of PL with existing PL trailing at 640–700 nm, and (OA)₂SnI₄ is more obvious (Supplementary Fig. 11). This broad-spectrum PL trailing phenomenon could be explained by the extrinsic STE effect. Deschler et al. indicated that the broad emission below the optical gap seen at

low temperatures in <001> oriented 2D-perovskite materials was due to the light-induced formation of localized trap states, associated with interstitial iodide and iodide Frenkel defects that act as color centers in the crystal [J. Am. Chem. Soc. 2017, 139, 18632-18639]. Besides, Loi et al also highlighted the extrinsic origin of their broad band emission in-gap states in the crystal bulk are responsible for the broad emission [Nat. Commun. 2020, 11, 2344]. The electron-phonon coupling effect with defect states trapping in the four materials may give rise to the extrinsic STE. The exciton-phonon coupling of (OA)₂SnI₄ is the strongest among the four materials, so there is a relatively wide PL trailing below the band gap, and the PL trailing is more pronounced at 77 K. Despite the relaxation rate and the proportion of the I process decrease at the high pump fluence as shown in Fig. 6d, this feature does not mean that the defect state is filled with the process, and it needs to be considered whether the excited exciton density reaches Mott density and thus allows exciton fission or the material is degenerate by strong light excitation. Assuming an exciton Bohr radius of 1 nm [Nat. Commun. 2020, 11, 664], the exciton saturation density is simply estimated to be 10¹⁴ cm⁻². The saturation of process I only a little occurs at high excitation fluence more than 40 μJ/cm² (1.1×10¹⁴ cm⁻²), so the excitation intensity exceeding the saturation density causing the saturation of process I cannot be characterized as a defect state trapping process. In the TA spectrum within 1 ps (Fig. 5), the redshift of the photobleaching peak of the band-edge exciton leading to the PIA center at 626 nm appears, which is attributed to the bandgap renormalization caused by the hot excitons [J. Am. Chem. Soc. 2015, 137, 2089–2096], where the maximum amplitude $-\Delta T/T$ of the PIA extracted at different excitation intensities n satisfies a linear relationship with the excitation intensity $n^{1/2}$ (Fig. 6e) [Science 2017, 356, 59–62, Nature Photon. 2016, 10, 53–59, and J. Appl. Phys. 2007, 101, 083705]. So this result also further confirms the presence of the bandgap recombination process in the process I. To further clearly distinguish which plays a role in the first relaxation process between the defect states trap exciton process and bandgap renormalization process induced by the hot exciton, we further research the fluence-dependent transient absorption spectra of the PEAI-rich and (PEA)₂SnI₄. We find that the first relaxation process is more significantly affected by the defect density of states at weak excitation fluence, approximately no more than 10 μJ/cm² (Fig. 6c). With further increase of the excitation fluence (Supplementary Fig. 16d), the first process appears the defect state density-independent, the excitation intensity-independent, temperature-independent relaxation process, so that the bandgap renormalization process dominates process I [J. Am. Chem. Soc. 2015, 137, 2089–2096 and J. Phys. Chem. C 2015, 119, 14714–14721] So the first fast process I of transient absorption measurement is attributed to the combination of the defects trapping excitons process and the band-gap renormalization process induced by hot excitons, i.e., the defects trapping excitons process play a leading role at low excitation fluence and as the excitation fluence increases, the bandgap renormalization process induced by hot excitons dominates in the process I. After the process of the bandgap renormalization caused by the hot excitons, the band filling effect gradually increases the blueshift of the bandgap [Nat Photonics, 2014, 8, 737-743], as shown in Supplementary Fig. 12.

The change of the band edge bleaching peak position at a delay time of 1.25 ps with the excitation intensity is observed, when the redshift process ends (Fig. 7a). As the excitation intensity is lower than $15 \mu\text{J cm}^{-2}$, the bleaching peak position does not change, and the blueshift occurs and becomes more noticeable as the excitation intensity increases because of the band filling effect (Supplementary Note 3) and the blueshift is proportional to $n^{2/3}$ [Nat Photonics, 2014, 8, 737-743], as shown in Fig. 7a. Additionally, the optical Stark effect can be excluded because we observe the shift well beyond the time duration of the pump laser pulse [J. Am. Chem. Soc. 2015, 137, 2089–2096 and J. Phys. Chem. C 2015, 119, 14714–14721]. Since it can be seen from the above that the density of defect states is the lowest in the $(\text{PEA})_2\text{SnI}_4$ sample, so the trapping rate of excitons and the proportion of process I in relaxation dynamics are the lowest, which is consistent with our experimental data (Supplementary Fig. 14 and Fig. 6f).” (lines 489-577)

In addition, we add content discussion to further confirm the validity of our conclusions.

In addition, the relaxation rate of the second component (II) in PEA-rich with smaller defect density states is slower than that of $(\text{PEA})_2\text{SnI}_4$ and PEA-rich has higher PLQY (Supplementary Fig. S16b), indicating that the effect of deformation potential scattering by charged defects on exciton interband recombination can be weakened by reducing the density of defect states, and improve the PLQY. (lines 639-644)

Fig. 6 | Ground state bleaching (GSB) relaxation process. (a) Temperature-dependent normalized GSB relaxation kinetics of the $(\text{PEA})_2\text{SnI}_4$ sample within 5 ps under the pump fluence of $3 \mu\text{J cm}^{-2}$. (b) Relaxation kinetics of the TA of each wavelength labeled by the black dotted line in Fig. 5(a) within 3 ps. (c) the TA spectra of the $(\text{PEA})_2\text{SnI}_4$ and PEA-rich sample under the pump fluence of $2 \mu\text{J cm}^{-2}$. (d) Pump fluence-dependent normalized GSB relaxation kinetics of the $(\text{PEA})_2\text{SnI}_4$ sample within 5 ps. (e) Pump fluence-dependent maximum value of the PIA signal at 624 nm of the $(\text{PEA})_2\text{SnI}_4$ sample. When the pump fluence is less than $15 \mu\text{J cm}^{-2}$, it is a linear excitation range, and when it exceeds $15 \mu\text{J cm}^{-2}$, it is a nonlinear excitation range. n is the density of the photoinduced exciton that is proportional to the pump fluence F . The solid black line represents a linear relationship and the solid blue line represents a one-half power relationship. (f) Normalized band edge GSB relaxation process under the pump fluence of $12 \mu\text{J cm}^{-2}$ for the $(\text{PEA})_2\text{SnI}_4$, $(\text{BA})_2\text{SnI}_4$, $(\text{HA})_2\text{SnI}_4$, and $(\text{OA})_2\text{SnI}_4$ thin films. The internal illustration shows the

relaxation process of the four materials within 2 ps. (lines 405-418)

Supplementary Figure 16. The influence of the cations stoichiometry engineering on the ground state bleaching (GSB) relaxation process of $(PEA)_2SnI_4$. (a) Characteristics of the ratio (P_{TE}/P_{FE}) of trap state exciton PL intensity to free exciton PL intensity as a function of temperature. (b) PLQY of $(PEA)_2SnI_4$ and PEAI-rich. (c) Relaxation process of the captured state of $(PEA)_2SnI_4$ and PEAI-rich. (d) Pump fluence-dependent normalized GSB relaxation kinetics of the $(PEA)_2SnI_4$ and PEAI-rich. (e) ground-state bleaching (GSB) relaxation process of $(PEA)_2SnI_4$ and PEAI-rich under the pump fluence of $2 \mu\text{J cm}^{-2}$. (f) The PLQY of $(BA)_2PbI_4$ and $(BA)_2SnI_4$ were obtained under the same experimental conditions. (Supporting Information lines 209-217)

The third component (III) is attributed to defect-assisted excitons recombination, which involves the radiative recombination induced by relatively shallow defect states, where the ratio of the PL the defect states (center at 660 nm) to that of the free exciton(center at 614nm) is about 30% at room temperature (Supplementary Fig. 16a), and the non-radiative recombination processes induced by deep defects which make the PLQY very low. Besides, the emission range of relatively shallow defect states increases with the enhancement of exciton-phonon scattering interaction among all the four materials (Fig.3, Fig.4, and Supplementary Fig. 11), which indicates the extrinsic STE state may exist. Deschler et al. indicated that the broad emission below the optical gap seen at low temperatures in <001> oriented 2D-perovskite materials was due to the light-induced formation of localized trap states, associated with interstitial iodide and iodide Frenkel defects that act as color centers in the crystal [J. Am. Chem. Soc. 2017, 139, 18632-18639]. Besides, Loi et al also highlighted the extrinsic origin of their broad band emission in-gap states in the crystal bulk are responsible for the broad emission [Nat. Commun. 2020, 11, 2344]. We applied the stoichiometry engineering of the cations where the PEA:I:SnI₂ ratio is 2.6:1 in (PEA)₂SnI₄ (PEAI-rich) [Nat. Commun. 2020, 11, 2344, and Adv. Funct. Mater. 2020, 30, 1907505], we reduce the defect density in PEAI-rich to improve the PLQY of the PEAI-rich to 2.2% (Supplementary Fig. 16b), which makes the integrated area of the bleaching peak relaxation process of trap states center at 676 nm of PEA-rich after 1.2 ps smaller than that of PEA. (Supplementary Fig. 16c).

In the original manuscript:

“Conversely, the third long relaxation component (III) with a nanosecond lifetime is derived from the trapped state exciton radiative recombination process”

was revised to

“Conversely, the third long relaxation component (III) with a nanosecond lifetime is attributed to defect-assisted excitons recombination, which involves the radiative recombination induced by relatively shallow defect states, where the ratio of the PL the defect states (center at 660 nm) to that of the free exciton (center at 614nm) is about 30% at room temperature (Supplementary Fig. 16a), and the non-radiative recombination processes induced by deep defects which make the PLQY very low. Besides, the emission range of relatively shallow defect states increases with the enhancement of exciton-phonon scattering interaction among all the four materials (Fig.3, Fig.4, and Supplementary Fig. 11), which indicates the extrinsic STE state may exist.[J. Am. Chem. Soc. 2017, 139, 18632-18639 and Nat. Commun. 2020, 11, 2344]” (lines 625-634)

In addition, the answer to the question of whether the bleaching peak at 520 nm is a CT or an intraband transition is given in detail in Question 1.

We thank you very much again for your valuable review comments and sincerely hope that our responses will be approved by you.

Reviewer #2 (Remarks to the Author):

The authors have studied the emission properties of the 2D tin halide perovskites with different organic cations, based on photoluminescence and transient absorption spectroscopy as function of temperature and power excitation. The understanding of the luminescent mechanisms in lead-free 2 hybrid perovskites is important for the applications. Deciphering the influence of the organic cation on the optical properties is particularly important to guide the optimization of the material. The results are interesting and original. However, the clarity of the discussion and the interpretation of these results could be improved. I have the following remarks:

We are very grateful to reviewer for spending valuable time making constructive and useful comments. We will address the reviewer s specific points on the following pages. The comments are underlined in black and the content of our response is highlighted in both blue and red, with those highlighted in red representing additions and corrections in the revised manuscript and Supporting Information. The comments are responded to in a point-by-point manner.

1. Line 161 in the manuscript, the authors claims that the absorption peak at 520 nm has been assigned to charge transfer transition between the organic spacer and the inorganic layers. However, the assumptions on which are based this assignation are not clear. Interestingly, the authors observe a bleaching in transient absorption at this wavelength in addition to the bleaching of the free exciton absorption. I don't think that a similar observation has been made for lead halide 2D perovskites. The authors could provide some comment on that point. Could they resolved the dynamics of the supposed charge transfer transition? Is there any difference between the TA of the charge transfer exciton and the one of the free exciton? It seems surprising that this charge transfer is independent of the choice of the organic cation.

Reply: Many thanks to the reviewers for their comments, which made us realize that we incorrectly attributed the absorption peak at ~520 nm to the charge transfer (CT) transition between the organic spacer and the inorganic layers, and we attributed the absorption peak at ~520 nm to the intraband transition of SnI₄ inorganic layers by combining literature study with the results of pump energy-dependent transient absorption (TA) spectroscopy. Since A₂SnI₄ perovskite (A = PEA⁺, BA⁺, HA⁺, and OA⁺) have similar linear absorption and TA spectra characteristics, we selected (PEA)₂SnI₄ as a representative for an explanation. The details are shown below:

In two-dimensional layered halide organic perovskites (LHOPs), the perovskite layer is the dominant component of band-edge absorption. Therefore, a way to demonstrate energy transfer would be the observation of triplet emission from the organic spacer layer. [Phys. Rev. Materials 2018, 2, 105406., J. Am. Chem. Soc. 2018, 140, 7313–7323.] Some LHOPs have demonstrated the ability to induce energy transfer from perovskite layer exciton states to low energy spin-triplet exciton states in the organic layer. [Chem. Phys. Lett. 1999, 303, 157–164., Chem. Phys. Lett. 1999,

307, 373–378., and Phys. Rev. Lett. 2008, 100, 257401.] When the lowest excitation energy (E) of the exciton in the perovskite layer aligns with the first triplet (T_1) excitation energy of the exciton in the organic layer, charge transfer from the perovskite to the organic layer may occur. After transfer, the T_1 excitation energy in the organic layer relaxes to a lower T_1^* energy due to enhanced short-range atomic deformation, thus reaching optimal triplet molecular geometry [Nano Lett. 2019, 19, 8732–8740]. Through theoretical calculations, Neukirch et al. systematically studied organic spacer and perovskite layer pairings for possible transfer of the Wannier excitons from the inorganic perovskite lattice to spin-triplet Frenkel excitons located on the organic cations and successfully identify ten organic spacer candidates for possible pairing with perovskite layers of specific halide composition to achieve triplet light emission across the visible energy range. From their calculations, it is clear that the T_1 energy of PEA^+ remains in a narrow range between 4.43 and 4.46 eV, which is greater than the lowest optical excitation peak energy (2.5 eV) of perovskite layer in the $(\text{PEA})_2\text{PbI}_4$. The absorption spectrum shows that the lowest optical excitation peak energy of SnI_4 in $(\text{PEA})_2\text{SnI}_4$ is smaller than that of PbI_4 in $(\text{PEA})_2\text{PbI}_4$ (2.02 eV vs 2.5 eV), so we can conclude that the exciton in SnI_4 cannot be transferred to organic cation PEA^+ . For the alkylammonium chains, it can not be excited by a photon in the visible region [Solid State Commun. 1989, 69, 933-936], so we can also conclude that the exciton in SnI_4 cannot be transferred to organic cation the alkylammonium chains.

In the pump energy-dependent TA spectroscopy experiments, where we used the wavelength of the pump at 400 nm, the wavelength of the pump at 613 nm with resonant band-edge absorption, and the wavelength of the pump at 630 nm below the band gap, we found the existence of photobleaching peaks at ~ 523 nm (PB1) and ~ 613 nm (PB2) in the TA spectra of these three different pump energies (Supplementary Fig. 2.). The relaxation kinetics of A_2SnI_4 ($\text{A} = \text{PEA}^+$, BA^+ , HA^+ , and OA^+) obtain by low pump fluence have been fitted globally with three components (Supplementary Fig. 14). We find that PB1 has the same relaxation characteristics and lifetime with PB2, and hence it is not consistent with the occurrence of the CT transition. In Supplementary Fig. 2, the two PB peaks are generated almost simultaneously when excited at 400 nm. However, the PB1 peak first reaches the maximum and then decreases, and the PB2 peak reaches a maximum with a delay of about 0.1 ps compared to the PB1 peak, which is more consistent with the intraband transitions [Science 2013, 342, 344-347.].

Combined with the above analysis, the PB1 peak is not the charge transfer transition between the organic spacer and the inorganic layers, but an intraband transitions process in the perovskite layer in the $(\text{PEA})_2\text{SnI}_4$.

Therefore, we revised the original incorrect description in the manuscript as follows:

In the original manuscript,

“the second peak at 520 nm (2.38 eV) was assigned to the charge transfer transition between the organic spacer cations and the inorganic layers”

is corrected to

“the second peak at 520 nm (2.38 eV) was assigned to the intraband transitions process in perovskite layer rather than the charge transfer transition between the organic spacer cations and the inorganic layers in the (PEA)₂SnI₄ [Nano Lett. 2019, 19, 8732–8740, Solid State Commun. 1989, 69, 933-936, and Science 2013, 342, 344-347]. It is mainly due to the fact that the first excitation energies of PEA⁺, BA⁺, HA⁺, and OA⁺ cations lie in the UV energy range [Nano Lett. 2019, 19, 8732–8740 and Solid State Commun. 1989, 69, 933-936], which is much higher than the lowest optical excitation peak energy of the perovskite layer and the energy of photobleaching peaks at ~523 nm. As a result, the excitons in the inorganic perovskite lattice can not transfer to spin-triplet Frenkel excitons states located on the organic cation [Nano Lett. 2019, 19, 8732–8740]. In the pump energy-dependent transient absorption spectroscopy experiments (Supplementary Fig. 2), we find that the photobleaching peaks at ~523 nm (PB1) and ~613 nm (PB1) have the same characteristics and lifetime of the relaxation decay process, and no new bleaching peaks appear during the decay relaxation of PB1. Despite the PB2 peak reaches a maximum with a delay of about 0.1 ps compared to the PB1 peak excited at 400 nm in Supplementary Fig. 3, which is more consistent with the intraband transitions while not consistent with the occurrence of the CT transition.[Science 2013, 342, 344-347].”
(a lines 175-189)

Supplementary Figure 2. Pump energy-dependent TA spectra of (PEA)₂SnI₄. 2D pseudocolor TA spectra obtained by pumping at a wavelength of (a) 400 nm, (b) 613

nm, and (c) 630 nm. TA spectra at different delay times with pumping at (d) 400 nm, (e) 613 nm, and (f) 630 nm. Relaxation kinetics at different wavelengths with pumping at (g) 400 nm, (h) 613 nm, and (i) 630 nm. (Supporting Information lines 109-113)

Supplementary Figure 3. Pump energy-dependent TA spectra of $(\text{PEA})_2\text{SnI}_4$ within 5ps. 2D pseudocolor TA spectra obtained by pumping at a wavelength of (a) 400 nm, (b) 613 nm, and (c) 630 nm. Relaxation kinetics at different wavelengths with pumping at (d) 400 nm, (e) 613 nm, and (f) 630 nm. (g) TA spectra at different delay times by pumping at a wavelength of 400 nm. (h) Relaxation kinetics at different wavelengths by pumping at a wavelength of 400 nm. (Supporting Information lines 114-120)

2. Line 177, the authors discuss the influence of the organic cation on the band gap. They highlight the influence of the dielectric constant. However, the influence on the exciton binding energy could be discussed in more details. The dielectric confinement play an important role in the estimation of the exciton binding energy and we can expect large difference between the use of PEA and alkyl-ammonium chains (for example Blancon et al. 10.1038/s41467-018-04659-x).

Reply: Many thanks to the reviewer for the comment. For the 2D perovskites, the

surrounding organic layer with a low dielectric constant is less polarizable and hence decreases the screening of the hole-electron Coulomb interaction. It results in an increase of the exciton binding energy. The discrepancy of dielectric constants ϵ between the inorganic framework (semiconductor) and the organic layers (surrounding) should be anticipated to give rise to the dielectric confinement on the exciton, which can be modulated by changing the composition of the organic cations [Nat. Commun. 2018, 9, 2254]. The decrease in ϵ_b and/or the increase in ϵ_w would lead to an increase in the 2D exciton binding energy. The PLQY of a layered tin perovskite may thus be further improved by enlarging the dielectric contrast between the tin halide layer and the intercalating ammonium cation. For the $(A)_2SnI_4$ (A : PEA^+ , BA^+ , HA^+ , and OA^+), ϵ_A is larger than ϵ_w (w : SnI_4), which leads to an enhancement of the Coulomb interaction between the electron and hole pair composing each exciton, which is a consequence of the reduced dielectric screening of the exciton electric field partially located outside the quantum well [Nat Commun. 2018, 9, 2254 and J. Am. Chem. Soc. 2019, 141, 10324–10330]. And $\epsilon_{BA}/\epsilon_{PEA}$ is larger than 1, which makes the exciton binding energy of $(BA)_2SnI_4$ is larger than that of $(PEA)_2SnI_4$. In the meantime, we obtained the exciton binding energy by fitting the steady-state absorption spectrum using a more rigorous Elliott theory, in which exciton binding energy of $(PEA)_2SnI_4$ is 213 ± 2 meV smaller than that of $(PEA)_2SnI_4$ (245 meV) (Supplementary Figure 6)). Plochocka et al. revealed that one of the biggest challenges in perovskites is in determining accurately the exciton binding energy in the material by experiments. The main contradiction lies in the fact that the polar nature of perovskites and the associated polariton effect are neglected [Adv. Energy Mater. 2020, 10, 1903659]. However, such a simple comparison of the dielectric constants of organic cations allows qualitative conclusions to be drawn. The fluorescence quantum yield is not only related to the exciton binding energy, but also the defect density caused by the difference of organic cations. At the same time, we add the relevant parts of the manuscript, as follows.

For the $(A)_2SnI_4$ (A : PEA^+ , BA^+ , HA^+ , and OA^+), ϵ_A is larger than ϵ_w (w : SnI_4), which leads to an enhancement of the Coulomb interaction between the electron and hole pair composing the exciton. It is a consequence of the reduced dielectric screening of the exciton electric field [Nat Commun. 2018, 9, 2254 and J. Am. Chem. Soc. 2019, 141, 10324–10330]. And ϵ_{BA} (2.2)/ ϵ_{PEA} (3.3) is larger than 1, which makes the exciton binding energy of $(BA)_2SnI_4$ is larger than that of $(PEA)_2SnI_4$. In the meantime, we obtained the exciton binding energy by fitting the steady-state absorption spectrum using a more rigorous Elliott theory, in which exciton binding energy of $(PEA)_2SnI_4$ is 213 ± 2 meV smaller than that of $(PEA)_2SnI_4$ (245 meV) (Supplementary Figure 6)). However, such a simple comparison of the dielectric constants of organic cations can provide qualitative conclusions due to that one of the biggest challenges in perovskites is in determining accurately the exciton binding energy in the material by experiments [Adv. Energy Mater. 2020, 10, 1903659]. (lines 202-222)

Supplementary Figure 6. Elliott theory of excitonic absorption of the four perovskites: (a) $(\text{PEA})_2\text{SnI}_4$, (b) $(\text{BA})_2\text{SnI}_4$, (c) $(\text{HA})_2\text{SnI}_4$, and (d) $(\text{OA})_2\text{SnI}_4$. yielding Exciton binding energy $E_b = 213 \pm 2 \text{ meV}$, line broadening with $\Gamma = 36 \pm 1.5 \text{ meV}$, band Gap $E_g = 2:275 \pm 0.001 \text{ eV}$ for $(\text{PEA})_2\text{SnI}_4$. Exciton binding energy $E_b = 245 \pm 1.6 \text{ meV}$, line broadening with $\Gamma = 70 \pm 1.5 \text{ meV}$, band Gap $E_g = 2:289 \pm 0.002 \text{ eV}$ for $(\text{BA})_2\text{SnI}_4$. Exciton binding energy $E_b = 248 \pm 1.5 \text{ meV}$, line broadening with $\Gamma = 60 \pm 1.5 \text{ meV}$, band Gap $E_g = 2:31 \pm 0.001 \text{ eV}$ for $(\text{HA})_2\text{SnI}_4$. Exciton binding energy $E_b = 236 \pm 2 \text{ meV}$, line broadening with $\Gamma = 60 \pm 1.5 \text{ meV}$, band Gap $E_g = 2:304 \pm 0.001 \text{ eV}$ for $(\text{OA})_2\text{SnI}_4$. (Supporting Information lines 134-142)

3. The authors have studied the evolution of the FWHM of the PL emission as function of temperature and extracted values for the coupling with optical phonons (Figure 4, Table 1). The values are given without measurement uncertainty and with a precision which seems really overestimated. In particular, for the compounds based on the cation BA, HA and OA, the authors could not present measurements at temperature below 200 K. For the range of temperature fitted here, the equation 2 in the Supplementary note (which needs a correction, for the Bose-Einstein term, kbT is missing) gives an almost linear relation as observed on the figure 4b, c and d above 200K. It seems impossible to separate ΓL_0 from $E L_0$ properly in this situation. If we compare the figure 4b and 4d, we observe that the data are very similar. However, the results from the fit are very different, with a factor 2 on ΓL_0 between OA and BA.

Reply: Many thanks to the reviewer for the comments. In order to improve the reliability and accuracy of the experimental data, we have performed several

temperature-dependent PL experiments under the same experimental conditions, from high temperature down to low temperature, in steps of 25 K. And each temperature was kept for 20 minutes to ensure the temperature stabilization. Also, in order to improve the realism and credibility of the fitting results, we refer to the optical phonon energies obtained from the steady-state Raman experiments (Supplementary Figure 12). And we did not discard the last term of the model representing the inhomogeneous broadening caused by ionized impurities to improve the reasonableness of the exciton-phonon coupling model (Supplementary Note 2) fitting experimental data for the two-dimensional Sn-based properties system we studied. The results of the model fit are shown in Figure 4 in the manuscript, and the fitted parameters were shown in Table 1. Compared with the alkyl chain samples, the (PEA)₂SnI₄ sample has a relatively smaller Fröhlich coupling intensity (Γ_{LO}). The results indicate that the (PEA)₂SnI₄ sample is more ordered, and the non-radiative energy loss of PL is smaller than those of the alkyl chain samples. The main reason is that PEA⁺ cations having the CH- π stacking characteristics that alkyl chain cations lack limit their thermal movement between the inorganic layers and induce weak dynamic changes in the SnI₄ structure. For the samples with an alkyl chain, (HA)₂SnI₄ (254.9±5 meV) has the lowest Γ_{LO} , followed by (BA)₂SnI₄ (272.8±4 meV), and finally (OA)₂SnI₄ (320.2±7 meV), which means that the longer alkyl-ammonium chain tends to enhance the intensity of exciton-phonon scattering. To investigate the relationship between the material structure and electron-phonon interactions, we also used atomic displacement parameters U_{eq} extracted from single-crystal X-ray diffraction (SCXRD) data corresponding to the four materials (Supplementary Information). The results revealed that the atomic displacements of the different atoms of A₂SnI₄ (A:BA⁺, HA⁺, and OA⁺) were distinctly larger than those of (PEA)₂SnI₄, as shown in Figure 4e and f. Thus, (PEA)₂SnI₄ has a more rigid structure compared to (BA)₂SnI₄, (HA)₂SnI₄, and (OA)₂SnI₄, which is consistent with the results of the exciton-phonon coupling model and theoretical calculations [J. Phys. Chem. Lett. 2020, 11, 2955–2964]. At the same time, we modified the relevant parts of the manuscript, as follows.

In the original manuscript:

“Compared with the alkyl chain group samples, the (PEA)₂SnI₄ sample not only has a smaller Γ_0 but also has a relatively smaller Fröhlich coupling intensity (Γ_{LO}), although they have similar optical phonon energies (E_{LO}). The results indicate that the (PEA)₂SnI₄ sample is more ordered, and the FWHM and non-radiative energy loss of PL caused by exciton-optical phonon scattering are smaller than those of the alkyl chain samples. The main reason is that PEA⁺ cations having the CH- π stacking characteristics that alkyl chain cations lack limit their thermal movement between the inorganic layers and induce weak dynamic changes in the SnI₄ structure. For the samples with an alkyl chain, (HA)₂SnI₄ (232.6 meV) has the lowest Γ_{LO} , followed by (BA)₂SnI₄ (257.94 meV), and finally (OA)₂SnI₄ (500.58 meV), which means that the longer alkyl-ammonium chain tends to enhance the intensity of exciton-phonon scattering. Notably, the exciton-phonon Fröhlich interaction is over one order of

magnitude larger than that reported for lead perovskites (Supplementary Table 4). The giant carrier phonon coupling can lead to an increase in the FWHM of the PL. To verify the rationality of the parameters obtained by our fitting, the four samples were subjected to steady-state Raman spectroscopy experiments (Fig. 4(e)). A Raman peak is located at 454.3 cm^{-1} (56.3 meV) for $(\text{PEA})_2\text{SnI}_4$ and 472.1 cm^{-1} (58.5 meV) for the alkyl chain group samples, respectively, which is close to those of the optical phonons obtained by the fitting. The interpretation of these Raman peaks requires strict theoretical calculations, which is beyond the scope of this study.”

was revised to

“In order to improve the realism and credibility of the fitting results, we refer to the optical phonon energies obtained from the steady-state Raman experiments (Supplementary Figure 12). A Raman peak is located at 454.3 cm^{-1} (56.3 meV) for $(\text{PEA})_2\text{SnI}_4$ and 472.1 cm^{-1} (58.5 meV) for the alkyl chain group samples, respectively. The interpretation of these Raman peaks requires strict theoretical calculations, which is beyond the scope of this study.” Compared with the alkyl chain samples, the $(\text{PEA})_2\text{SnI}_4$ sample has a relatively smaller Fröhlich coupling intensity (F_{LO}). The results indicate that the $(\text{PEA})_2\text{SnI}_4$ sample is more ordered, and the non-radiative energy loss of PL is smaller than those of the alkyl chain samples. The main reason is that PEA^+ cations having the CH- π stacking characteristics that alkyl chain cations lack limit their thermal movement between the inorganic layers and induce weak dynamic changes in the SnI_4 structure [J. Phys. Chem. Lett. 2020, 11, 2955–2964]. For the samples with an alkyl chain, $(\text{HA})_2\text{SnI}_4$ (254.9 \pm 5 meV) has the lowest F_{LO} , followed by $(\text{BA})_2\text{SnI}_4$ (272.8 \pm 4 meV), and finally $(\text{OA})_2\text{SnI}_4$ (320.2 \pm 7 meV), which means that the longer alkyl-ammonium chain tends to enhance the intensity of exciton-phonon scattering. Notably, the exciton-phonon Fröhlich interaction of the 2D Sn-based properties system we studied. is over one order of magnitude larger than that reported for lead perovskites (Supplementary Table 4). The giant exciton - phonon coupling can lead to an increase in the FWHM of the PL and the possibility form self-trapped exciton (STE) states, which further might couple with defective states [Acc. Chem. Res. 2018, 51, 3, 619–627, J. Phys. Chem. Lett. 2016, 7, 2258-2263, and Nat. Commun. 2020, 11, 2344.]. To investigate the relationship between the material structure and electron-phonon interactions, We also studied the atomic displacement parameters U_{eq} extracted from single-crystal X-ray diffraction (SCXRD) data corresponding to the four materials (Supplementary Information). The results revealed that the atomic displacements of the different atoms of A_2SnI_4 (A: BA^+ , HA^+ , and OA^+) were distinctly larger than those of $(\text{PEA})_2\text{SnI}_4$, as shown in Figure 4e and f. Thus, $(\text{PEA})_2\text{SnI}_4$ has a more rigid structure compared to $(\text{BA})_2\text{SnI}_4$, $(\text{HA})_2\text{SnI}_4$, and $(\text{OA})_2\text{SnI}_4$, which is consistent with the results of the exciton-phonon coupling model and theoretical calculations [J. Phys. Chem. Lett. 2020, 11, 2955–2964].” (lines 358-385)

Table 1 Best-fitting parameters of the $(\text{PEA})_2\text{SnI}_4$, $(\text{BA})_2\text{SnI}_4$, $(\text{HA})_2\text{SnI}_4$, and

(OA)₂SnI₄ perovskites.

Sample	Γ_0 (meV)	Γ_{LO} (meV)	E_{LO} (meV)	Γ_{imp} (meV)	E_{imp} (meV)
(PEA) ₂ SnI ₄	26.8±1	199.3±2	58.5±1	722.3±2	69.7±2
(BA) ₂ SnI ₄	63.1±2	272.8±4	56.8±2	1513.3±2	89.9±2
(HA) ₂ SnI ₄	41.3±2	254.9±5	56.8±1	881.2±2	82.3±2
(OA) ₂ SnI ₄	66.2±2	320.2±7	57.85±1	1211.7±2	120.7±2

(lines 392-393)

Fig. 4 | Exciton-phonon coupling. Temperature-dependent FWHM for (a) (PEA)₂SnI₄, (b) (BA)₂SnI₄, (c) (HA)₂SnI₄, and (d) (OA)₂SnI₄. The red fitting line of the data (blue

point) is obtained by model (2) fitting in Supplementary Note 2. Average atomic displacement U_{eq} of chemical elements (H, C, N, Sn, and I) in $(PEA)_2SnI_4$, $(BA)_2SnI_4$, $(HA)_2SnI_4$, and $(OA)_2SnI_4$ at (e) 275K and (f) 125K extracted from single-crystal X-ray diffraction (SCXRD) data (Supplementary Information). (lines 394-400)

Supplementary Figure 12. Steady-state Raman spectra of the four perovskites $(PEA)_2SnI_4$, $(BA)_2SnI_4$, $(HA)_2SnI_4$, and $(OA)_2SnI_4$. (Supporting Information lines 178-181)

4. Line 408 the authors have extracted the exciton binding energy from an Arrhenius fit of the PL. Some precautions must be taken in the text. This is not a very precise means to estimate the exciton binding energy. Other mechanisms can explained the

thermal quenching of the photoluminescence such as the presence of trap states and the authors have indeed highlighted the presence of trap states in their compounds. The sentence “Thus, it is a large part of the unseparated excitons that affect the optical properties of the material, despite that only a few excitons split into free carriers, which is consistent with the results of the THz study 51 .” is unclear. What indicates that a few excitons split into free carriers? Additionally, the study 51 refers to lead halide perovskite not tin.

Reply: Many thanks to the reviewer for this comment. Plochocka et al. revealed that one of the biggest challenges in perovskites is in determining accurately the exciton binding energy in the material by experiments. The main contradiction lies in the fact that the polar nature of perovskites and the associated polariton effect are neglected [Adv. Energy Mater. 2020, 10, 1903659]. There are many methods to determine the exciton binding energy, such as absorption spectrum methods, temperature-dependent PL methods, Magneto-Optical Investigation, etc. However, these methods have insurmountable drawbacks of their own that hinder the experimental simplicity and convenience of indeed being able to excite the true binding energy. In the present work, our accurate determination of the exciton binding energy is not the focus of our study; we need the nature of the optical transition in the material at room temperature to be predominantly the exciton. This is because our intensity-dependent PL experimental results indicate that the nature of the optical transitions in the material is a single-particle radiative recombination feature. Also, we determine the exciton binding energy higher than $K_B T$ at room temperature by temperature-dependent PL, such that the result generally concludes that excitons can be stable at room temperature. Determination of exciton binding energy by temperature-dependent PL is mainly using the quenching of the integrated photoluminescence (PL) intensity with the temperature with assuming that the rate of nonradiative recombination is related solely to the thermally activated exciton dissociation [Energy Environ. Sci. 2014, 7, 399 and Nano Lett. 2013, 13, 4505]. The PL intensity was fitted using an Arrhenius formula. However, there are some shallow trap states affecting the fitting accuracy of the fitting method of the Arrhenius formula. In order to reduce the influence of this factor, we performed a split-peak fitting of the PL at each temperature and extracted the sub-PL peaks of the radiation recombination from edge-free excitons. And this sub-PL peak was fitted using the Arrhenius formula and the binding energy of the free exciton was obtained in $(\text{PEA})_2\text{SnI}_4$ as 43.86 ± 4 meV (Supplementary Figure 10.). Second, although we use continuous light excitation fluorescence with low excitation intensity, which excludes higher-order intermittent compound processes, it is also difficult to exclude the influence of other compound processes. In addition, we obtained exciton binding energies of about 213 meV for $(\text{PEA})_2\text{SnI}_4$ using a linear steady-state absorption spectroscopy method in combination with the Tau plot [Nat. Commun. 2018, 9, 2254] (Supplementary Figure 9), which is bigger than 43.86 ± 4 meV obtained by the Arrhenius formula. The exciton binding energy obtained by absorption spectroscopy is more trustworthy, mainly because of the distinct exciton absorption peak in the steady-state absorption spectrum and the more pronounced separation from the band edge of the continuum. Although the difference between

these two results is relatively large, it still illustrates the optical leap characteristics in the exciton dominant material at room temperature for n=1 2D perovskites [Nat. Commun. 2018, 9, 2254]. So we made the corresponding changes in the manuscript.

In the original manuscript:

“Furthermore, the exciton binding energy of (PEA)₂SnI₄, obtained by fitting of A peak using Arrhenius relation, is 30.6 meV (Supplementary Fig. 6(g)), which is greater than the thermal energy ($k_B T = 25$ meV at 300 K) and the same order of magnitude as that for (PEA)₂SnI₄ (44.9 meV) reported in the literature. Thus, it is a large part of the unseparated excitons that affect the optical properties of the material, despite that only a few excitons split into free carriers, which is consistent with the results of the THz study.”

was revised to

“There are many methods to determine the exciton binding energy, such as absorption spectrum methods, temperature-dependent PL methods, Magneto-Optical Investigation, etc [Nat Commun. 2018, 9, 2254, Adv. Energy Mater. 2020, 10, 1903659]. But the polar nature of perovskites and the associated polariton effect are neglected, which makes the exciton binding energy obtained by different methods under different experimental conditions highly discrepant. [Adv. Energy Mater. 2020, 10, 1903659]. The accuracy of the exciton binding energy obtained by fitting the temperature-dependent PL intensity using the Arrhenius formula is severely affected by recombination processes such as shallow captured-state emission and Auger recombination, so we obtained the exciton binding energy by fitting the steady-state absorption spectrum using a more rigorous Elliott theory [Nat. Commun. 2020 11, 850]. (Supplementary Figure 6.) The exciton binding energy of (PEA)₂SnI₄ is 213±2 meV greater than the thermal energy ($k_B T = 25$ meV at 300 K), which reveals excitons dominate the nature of optical transitions at room temperature [Nat. Commun. 2018, 9, 2254].” (lines 448-460)

The femtosecond optical pumping THz transient absorption spectra can prove the exciton generation and decay in materials on a time scale. Our technology to study THz is not mature yet, but we have been working to optimize the technology of the THz research platform, and we will focus on using THz to study the photogenerated carrier dynamics of two-dimensional tin-based perovskites in our next work.

Supplementary Figure 6. Elliott theory of excitonic absorption of the four perovskites: (a) $(\text{PEA})_2\text{SnI}_4$, (b) $(\text{BA})_2\text{SnI}_4$, (c) $(\text{HA})_2\text{SnI}_4$, and (d) $(\text{OA})_2\text{SnI}_4$, yielding Exciton binding energy $E_b = 213 \pm 2$ meV, line broadening with $\Gamma = 36 \pm 1.5$ meV, band Gap $E_g = 2:275 \pm 0.001$ eV for $(\text{PEA})_2\text{SnI}_4$. Exciton binding energy $E_b = 245 \pm 1.6$ meV, line broadening with $\Gamma = 70 \pm 1.5$ meV, band Gap $E_g = 2:289 \pm 0.002$ eV for $(\text{BA})_2\text{SnI}_4$. Exciton binding energy $E_b = 248 \pm 1.5$ meV, line broadening with $\Gamma = 60 \pm 1.5$ meV, band Gap $E_g = 2:31 \pm 0.001$ eV for $(\text{HA})_2\text{SnI}_4$. Exciton binding energy $E_b = 236 \pm 2$ meV, line broadening with $\Gamma = 60 \pm 1.5$ meV, band Gap $E_g = 2:304 \pm 0.001$ eV for $(\text{OA})_2\text{SnI}_4$. (Supporting Information lines 134-142)

5. The authors assigned the process I of transient absorption measurement to trapping of exciton (line 459) based on its decrease with pump fluence, interpreted as the filling of trap states. However, they also supposed that the trapping is induced by electron-phonon coupling based on the paper of Wu et al. (ref 20) (line 487). The two hypothesis seems conflicting. If the trap states are self-trapped exciton induced by electron-phonon coupling, as supposed by Wu et al., a filling of these traps state is very unlikely. Hence, the trap states observed by the authors is most likely caused by chemical defects in the crystals.

Reply: Many thanks to the reviewer for the critical and high quality comments. In order to answer the origin of this trap state, we have re-done systematically the temperature-dependent, excitation-fluence-dependent, and excitation-energy-dependent transient absorption experiments and temperature-dependent PL experiments, combined with a more thorough study of the

literature. The following conclusions have been obtained as shown: The first fast process I of transient absorption measurement is attributed to the combination of the defects trapping excitons process and the band-gap renormalization process induced by hot excitons, i.e., the defects trapping excitons process play a leading role at low excitation fluence and as the excitation fluence increases, the bandgap renormalization process induced by hot excitons dominates in process I. The main reasons are as follows.

(4) This trap state in the process I almost is the chemical defect state in the material and not the intrinsic self-trapped exciton (STE) state caused by the exciton-phonon coupling. There are three main types of exciton trapping state, namely the intrinsic STE state, defect trapping state, and the extrinsic STE state. Intrinsic STE state is a transient defect that forms in the excited state where photogenerated charge carriers are stabilized through large lattice distortions driven by strong electron-phonon coupling [Acc. Chem. Res. 2018, 51, 3, 619–627]. Luminescence from the STE state normally gives rise to the broadband and large Stokes-shifted PL. Intrinsic STE state formation is found to have a strong dependence on the dimensionality of the system [Phys. Rev. B: Condens. Matter Mater. Phys. 1993, 47, 6060–6064. and Phys. Rev. Lett. 1976, 36, 323–326.]. There is no potential barrier separating the free exciton and STE in the one-dimensional materials, so the STE states form promptly within the subpicosecond time scales from the free exciton [Phys. Rev. Lett. 1998, 81, 417–420, Angew. Chem. Int. Ed. 2019, 58, 2278–2283, and J. Phys.: Condens. Matter 2013, 25, 144204.]. In the three-dimensional systems, intrinsic STE formation usually takes the ns magnitude of a long time to form due to the existence of a large potential barrier for trapping [J. Phys. Soc. Jpn. 1983, 52, 4277–4282]. While the two-dimensional system is a marginal case with low self-trapping activation energy [J. Am. Chem. Soc. 2019, 141, 12619–12623, J. Am. Chem. Soc. 2014, 136, 13154–13157, and J. Phys. Chem. Lett. 2016, 7, 2258–2263]. For the intrinsic STE state, with the temperature decreases, the stronger luminescence from the STE state, the band edge exciton luminescence intensity decreases, mainly because at low temperatures, the thermal activation of the detrapping process can not meet requirements of the detrapping barrier, the self-trapped excitons can not return to the band edge. Besides, the lattice distortion caused by electron-phonon coupling provides for homogeneous emission broadening [J. Am. Chem. Soc. 2014, 136, 13154–13157]. Defect trapping generally refers to the direct trap process of excitons by permanent defects in the lattice, without taking into account the effect of distortion of the lattice due to the optical excitation. And the emission involving defects may show sublinear behavior if the limited number of defect states become saturated, however, for intrinsic STE states emission, it has a linear relationship. [Phys. Rev. B: Condens. Matter Mater. Phys. 1992, 45, 8989–8994 and J. Am. Chem. Soc. 2014, 136, 13154–13157]. Since the lattice of two-dimensional perovskites is relatively soft and defect state density is relatively high, the intrinsic STE is often influenced by permanent defects, i.e. extrinsic STE. Self-trapping is influenced by the local

heterogeneity of permanent defects lattice, in which it will sink to a different trapping depth. Extrinsic STE leads to the inhomogeneous nature of the STE states and their radiative and nonradiative decay compared to the case for intrinsic STE [Acc. Chem. Res. 2018, 51, 3, 619–627, J. Phys. Chem. Lett. 2016, 7, 2258–2263, and Nat. Commun. 2020, 11, 2344.]. Using transient absorption spectroscopy to study the dynamics of the broad emission spectrum below the bandgap, it is the ability to distinguish the intrinsic STE and defect trapping. There is a broad photoinduced absorption at energies below the optical gap for the intrinsic STE states consistent with the formation of transient, light-induced, trap states [J. Phys. Chem. Lett. 2016, 7, 2258–2263 and Acc. Chem. Res. 2018, 51, 3, 619–627]. In contrast, permanent trap states exhibit below-exciton bleaching features owing to filling in 2D perovskites [J. Am. Chem. Soc. 2015, 137, 2089–2096].

In the previous section of the manuscript, we demonstrated that excitonic contribution to the PL is dominant in the materials by excitation fluence-dependent integral PL in (PEA)₂SnI₄, (BA)₂SnI₄, (HA)₂SnI₄, and (OA)₂SnI₄ materials. Due to the transient and linear absorption spectra of the (PEA)₂SnI₄, (BA)₂SnI₄, (HA)₂SnI₄, and (OA)₂SnI₄ samples have similar characteristics, and (PEA)₂SnI₄ has no phase transition in the temperature range of 300K–77K, the treatment of the relaxation process of (PEA)₂SnI₄ is used as a representation obtain a more detailed elucidation. In the temperature-dependent PL experiment, using the multi-peak fitting methods mentioned earlier in the manuscript, the ratio of the PL percentage of the free excitons (P_{FE}) to that of the trap state excitons (P_{TE}) below the bandgap decrease with decreasing temperature (Supplementary Fig. 16a), which is opposite to the feature of the intrinsic STE states emission. More precisely, it may be the emission of the extrinsic STE states [J. Phys. Chem. Lett. 2016, 7, 2258–2263 and Nat. Commun. 2020, 11, 2344]. The PL of the four materials((PEA)₂SnI₄, (BA)₂SnI₄, (HA)₂SnI₄, and (OA)₂SnI₄) shows asymmetry at 77 K (Supplementary Fig.11), where PL peaks under the bandgap exist in a wide range of trailing feature. This trailing spectral width is more and more serious in (BA)₂SnI₄, (HA)₂SnI₄, and (OA)₂SnI₄, especially in (HA)₂SnI₄, and (OA)₂SnI₄ at 77 K when the phase transition causes PL blue shift with PL peak still existing at 640–750 nm, which is more obvious in (OA)₂SnI₄. This broad-spectrum PL trailing phenomenon can be explained by the extrinsic STE effect. Deschler et al. indicated that the broad emission below the optical gap seen at low temperatures in <001> oriented 2D-perovskite materials was due to the light-induced formation of localized trap states, associated with interstitial iodide and iodide Frenkel defects that act as color centers in the crystal [J. Am. Chem. Soc. 2017, 139, 18632–18639]. Besides Loi et al also highlighted the extrinsic origin of their broadband emission in-gap states in the crystal bulk are responsible for the broad emission [Nat. Commun. 2020, 11, 2344]. There is a broad (640–750 nm) and weak bleach feature below the optical gap consistent with the defect state being filled in the transient absorption spectrum for the four materials (Fig. 5). The process of filling bleaching of these defect states is the

same synchronous step as the first process I of band-edge exciton relaxation (Fig. 6b) so that the band-edge excitons are transferred to the in-gap defect states. The electron-phonon coupling effect with defect state trapping could lead to the extrinsic STE in the four materials. The exciton-phonon coupling of (OA)₂SnI₄ is the strongest among the four materials, so there is a relatively wide PL trailing below the bandgap, and this PL trailing is more pronounced at 77 K (Supplementary Fig.11).

To further demonstrate the fact that excitons are trapped by chemical defects, we applied the stoichiometry engineering of the cations where the PEAI:SnI₂ ratio is 2.6:1 in (PEA)₂SnI₄ (PEAI-rich) [Nat. Commun. 2020, 11, 2344, and Adv. Funct. Mater. 2020, 30, 1907505], we reduce the defect density in PEAI-rich to improve the PLQY of the PEAI-rich to 2.2% (Supplementary Fig. 11b). We found that the occupancy ratio and relaxation rate of the first relaxation process I of the band-edge exciton relaxation processes in PEAI-rich were significantly lower than those of (PEA)₂SnI₄ sample (Fig. 6c). In addition, at high excitation intensities, the bleaching signal intensity at 676 nm is weaker and the relaxation rate is slower for PEA-rich compared to (PEA)₂SnI₄. (Supplementary Fig.16c) These results from the transient absorption spectrum provide more evidence that the exciton capture process of the first process I is the defect trapping process [J. Am. Chem. Soc. 2015, 137, 2089–2096].

Therefore, combining the above results, we attribute the first relaxation process mainly to the defect trapping process.

- (5) In the transient absorption spectrum (Figure 5), the band-edge exciton (~614 nm) shows a photobleaching signal attributed to state filling, i.e., the presence of bandgap excitons generated by the pump pulse blocking of the optical transition induced by the probe pulse. Within 1 ps, we observe a redshift of the photobleaching peak of the band-edge exciton leading to photoinduced absorption, which is attributed to the bandgap renormalization caused by the hot excitons, which can also cause process I. Assuming an exciton Bohr radius of 1 nm [Nat. Commun. 2020, 11, 664.], the exciton saturation density is simply estimated to be 10¹⁴ cm⁻². In the previous manuscript (Fig. 6d), the saturation of process I occur at high excitation fluence of 40 μJ/cm² (1.1×10¹⁴/cm²). This excitation fluence makes it possible for the exciton to fission and for the material to degenerate at high excitation intensities. Therefore, the previous conclusion is incorrect. Therefore, we retested the excitation intensity-dependent transient absorption spectrum experiment and found that process I do not be a little saturated until the excitation intensity is near the saturation density. To further clearly distinguish which plays a role in the first relaxation process between the defect states trap exciton process and bandgap renormalization process induced by the hot exciton, we further research the fluence-dependent transient absorption spectra of the PEAI-rich and (PEA)₂SnI₄.
- (6) In the excited intensity-dependent transient absorption spectra of the PEAI-rich and (PEA)₂SnI₄,

We find that the first relaxation process is more significantly affected by the defect density of states at weak excitation fluence, approximately no more than $10 \mu\text{J}/\text{cm}^2$ (Figure 6c). With further increase of the excitation fluence (Supplementary Fig.16d), the first process appears the defect state density-independent, the excitation intensity-independent, temperature-independent relaxation process, so that the bandgap renormalization process dominates the process I.

In summary, the first fast process I of transient absorption measurement is attributed to the combination of the defects trapping excitons process and the band-gap renormalization process induced by hot excitons, i.e., the defects trapping excitons process play a leading role at low excitation fluence and as the excitation fluence increases, the bandgap renormalization process induced by hot excitons dominates in the process I. As a result, we revised the manuscript content as follows.

In the original manuscript:

“When the pump fluence is lower than $15 \mu\text{J cm}^{-2}$, the proportion and relaxation rate of the I process remain basically unchanged. However, with the further increase in the pump fluence, the relaxation rate and the proportion of the I process decrease, as shown in Fig. 7(c). This may occur because the photogenerated excitons fill the shallow trap state in different degrees and reduce the trapping rate. For the trapping process, it should involve the transfer of photogenerated carriers between energy levels, indicated by the redshift of the bleaching peak in the spectra. To prove this, the relaxation kinetics of different wavelengths in Fig. 7(d) show that the completion of the band edge ground state bleaching (614.8 nm) relaxation process during the delay time range of $0.48\text{--}1.14 \text{ ps}$ is synchronized with the relaxation process of the PIA signal (624.1 nm), changing from positive PIA to negative bleaching and reaching the maximum. This indicates that the photogenerated excitons undergo a transfer from the band edge energy level to the shallow trap energy level, where the radiative recombination of excitons subsequently occurs, inducing a change in the relaxation kinetics of 624.1 nm from positive to negative absorption. Thus, combining all the above characteristics, component I is attributed to the process of exciton trapping by the shallow trap state, and the trap rate is independent of the temperature but related to the carrier density. However, the bandgap renormalization effect is the result of the competition between the reduction of the electron bandgap, caused by the exchange-correlation potential, which leads to the redshift of the bleaching peak, and the reduction of the exciton binding energy, caused by the dielectric screening effect, which leads to the blueshift of the bleaching peak, of the photogenerated carriers. Hence, this effect should show that the bleaching peak undergoes a redshift–blueshift transition, after which the band filling effect increases the bandgap blueshift gradually. Additionally, the redshift caused by this physical effect should increase as the excitation intensity increases; however, the results are not consistent with this law as they indicate that the redshift initially remains unchanged in the low excitation intensity range below $15 \mu\text{J cm}^{-2}$ and subsequently decreases until disappearing as the excitation intensity further increases to greater than $15 \mu\text{J cm}^{-2}$, as shown in Supplementary Fig. 10. The change in the band edge bleaching peak position at a

delay time of 1.25 ps when the redshift process ends with the excitation intensity is observed (Fig. 8(a)). When the excitation intensity is lower than $15 \mu\text{J cm}^{-2}$, the bleaching peak position does not change, and the blueshift occurs and becomes more noticeable as the excitation intensity increases because of the band filling effect (Supplementary Note 3). The blueshift is proportional to $n^{2/3}$, as shown in Fig. 8(a). Moreover, if the component I is caused by the bandgap renormalization process, its maximum value of the PIA should be proportional to $n^{1/2}$. As shown in Fig. 7(e), however, it does not conform to this law. Thus, the bandgap renormalization has little effect on the (I) process, which may play a role in the process of bleaching peak generation. This is mainly because the laser pump light pulse width of 350 fs is compared with the generation time of the bleaching peak, i.e., the processes of photogenerated carrier thermalization and exciton formation. Additionally, the optical Stark effect can be excluded because the redshift mainly increases with the excitation intensity. Since the trap state in 2D perovskite materials is induced by electron-phonon coupling and has weak optical transition intensities, it can be seen from the above that the density of trapped states is the lowest in the $(\text{PEA})_2\text{SnI}_4$ sample, so the trapping rate of excitons and the proportion in relaxation dynamics are the lowest (Fig. 6 and Fig. 7(f)), which is consistent with our experimental data.”

was revised to

“For the trap states in two-dimensional perovskites, we need to define the nature of the trap states, i.e., whether they are intrinsic self-trapped exciton (STE) state, defect trapping state, or the extrinsic STE state. In the temperature-dependent PL experiment, using the multi-peak fitting methods mentioned in the previous content, the ratio of the PL percentage of the free excitons (P_{FE}) to that of the trap state excitons (P_{TE}) below the bandgap decreases with decreasing temperature (Supplementary Fig. 16a), which is obviously opposite to the feature of the intrinsic STE states emission, in which the stronger luminescence from the intrinsic STE state and the band edge exciton luminescence intensity decreases with the temperature decreases, mainly because the thermal activation of the detrapping process can not meet requirements of the detrapping barrier and the self-trapped excitons can not return to the band edge at low temperatures [Nat. Commun. 2020, 11, 2344, Chem. Rev. 2019, 119, 3104–3139, J. Am. Chem. Soc. 2019, 141, 12619–12623, and Acc. Chem. Res. 2018, 51, 619–627]. More precisely, this trap state excitons below the bandgap may be the emission of the extrinsic STE states, that is intrinsic Self-trapping is influenced by the local heterogeneity of permanent defects lattice to get the different trapping depth [Nat. Commun. 2020, 11, 2344, J. Phys. Chem. Lett 2016, 7, 2258–2263, and J. Am. Chem. Soc. 2017, 139, 18632–18639]. There is a broad (640–750 nm) and weak bleaching feature below the optical gap in the transient absorption spectrum for four materials (Fig. 5) also consistent with the permanent defect states being filled not the intrinsic STE state featured by a broad PIA at energies below the optical gap consistent with characteristics of the formation of transient light-induced trap states [J. Phys. Chem. Lett. 2016, 7, 2258–2263, J. Am. Chem. Soc. 2015, 137, 2089–2096, and Acc. Chem. Res. 2018, 51, 3, 619–627] The relaxation kinetics of different wavelengths show that the process of filling bleaching of these defect states (663.4 nm) and the relaxation

process of the PIA signal (624.1 nm of PL center) changing from positive PIA to negative bleaching and reaching the maximum are the same synchronous step as the process I of the band-edge exciton (614.8 nm) relaxation (Fig. 6b), revealing that the band-edge excitons are trapped to the in-gap defect states. To further demonstrate the fact that excitons are trapped by chemical defects, we applied the stoichiometry engineering of the cations where the PEAI:SnI₂ ratio is 2.6:1 in (PEA)₂SnI₄ (PEAI-rich) [Nat. Commun. 2020, 11, 2344, and Adv. Funct. Mater. 2020, 30, 1907505], we reduce the defect density in PEA-rich to improve the PLQY of the PEA-rich to 2.2% (Supplementary Fig. 16b). We found that the occupancy ratio and relaxation rate of the first relaxation process I of the band-edge exciton relaxation processes in PEA-rich were significantly lower than those of (PEA)₂SnI₄ sample (Fig. 6c). In addition, at high excitation intensities, the bleaching signal intensity at 676 nm is weaker and the relaxation rate is slower for PEA-rich compared to (PEA)₂SnI₄ (Supplementary Fig. 16c). In summary, process I contain the process of defect states trapping exciton. In particular, the trailing spectral width of PL is more and more serious in (BA)₂SnI₄, (HA)₂SnI₄, and (OA)₂SnI₄, especially in (HA)₂SnI₄, and (OA)₂SnI₄ at 77 K when the phase transition causes blueshift of PL with existing PL trailing at 640-700 nm, and (OA)₂SnI₄ is more obvious (Supplementary Fig. 11). This broad-spectrum PL trailing phenomenon could be explained by the extrinsic STE effect. Deschler et al. indicated that the broad emission below the optical gap seen at low temperatures in <001> oriented 2D-perovskite materials was due to the light-induced formation of localized trap states, associated with interstitial iodide and iodide Frenkel defects that act as color centers in the crystal [J. Am. Chem. Soc. 2017, 139, 18632-18639]. Besides, Loi et al also highlighted the extrinsic origin of their broadband emission in-gap states in the crystal bulk are responsible for the broad emission [Nat. Commun. 2020, 11, 2344]. The electron-phonon coupling effect with defect states trapping in the four materials may give rise to the extrinsic STE. The exciton-phonon coupling of (OA)₂SnI₄ is the strongest among the four materials, so there is a relatively wide PL trailing below the bandgap, and the PL trailing is more pronounced at 77 K. Despite the relaxation rate and the proportion of the I process decrease at the high pump fluence as shown in Fig. 6d, this feature does not mean that the defect state is filled with the process, and it needs to be considered whether the excited exciton density reaches Mott density and thus allows exciton fission or the material is degenerate by strong light excitation. Assuming an exciton Bohr radius of 1 nm [Nat. Commun. 2020, 11, 664], the exciton saturation density is simply estimated to be 10¹⁴ cm⁻². The saturation of process I only a little occurs at high excitation fluence more than 40 μJ/cm² (1.1×10¹⁴ cm⁻²), so the excitation intensity exceeding the saturation density causing the saturation of process I cannot be characterized as a defect state trapping process. In the TA spectrum within 1 ps (Fig. 5), the redshift of the photobleaching peak of the band-edge exciton leading to the PIA center at 626 nm appears, which is attributed to the bandgap renormalization caused by the hot excitons [J. Am. Chem. Soc. 2015, 137, 2089–2096], where the maximum amplitude $-\Delta T/T$ of the PIA extracted at different excitation intensities n satisfies a linear relationship with the excitation intensity $n^{1/2}$ (Fig. 6e) [Science 2017,

356, 59–62, Nature Photon. 2016, 10, 53–59, and J. Appl. Phys. 2007, 101, 083705]. So this result also further confirms the presence of the bandgap recombination process in the process I. To further clearly distinguish which plays a role in the first relaxation process between the defect states trap exciton process and bandgap renormalization process induced by the hot exciton, we further research the fluence-dependent transient absorption spectra of the PEAI-rich and $(\text{PEA})_2\text{SnI}_4$. We find that the first relaxation process is more significantly affected by the defect density of states at weak excitation fluence, approximately no more than $10 \mu\text{J}/\text{cm}^2$ (Fig. 6c). With further increase of the excitation fluence (Supplementary Fig. 16d), the first process appears the defect state density-independent, the excitation intensity-independent, temperature-independent relaxation process, so that the bandgap renormalization process dominates process I [J. Am. Chem. Soc. 2015, 137, 2089–2096 and J. Phys. Chem. C 2015, 119, 14714–14721] So the first fast process I of transient absorption measurement is attributed to the combination of the defects trapping excitons process and the band-gap renormalization process induced by hot excitons, i.e., the defects trapping excitons process play a leading role at low excitation fluence and as the excitation fluence increases, the bandgap renormalization process induced by hot excitons dominates in the process I. After the process of the bandgap renormalization caused by the hot excitons, the band filling effect gradually increases the blueshift of the bandgap [Nat Photonics, 2014, 8, 737-743], as shown in Supplementary Fig. 12. The change of the band edge bleaching peak position at a delay time of 1.25 ps with the excitation intensity is observed, when the redshift process ends (Fig. 7a). As the excitation intensity is lower than $15 \mu\text{J cm}^{-2}$, the bleaching peak position does not change, and the blueshift occurs and becomes more noticeable as the excitation intensity increases because of the band filling effect (Supplementary Note 3) and the blueshift is proportional to $n^{2/3}$ [Nat Photonics, 2014, 8, 737-743], as shown in Fig. 7a. Additionally, the optical Stark effect can be excluded because we observe the shift well beyond the time duration of the pump laser pulse [J. Am. Chem. Soc. 2015, 137, 2089–2096 and J. Phys. Chem. C 2015, 119, 14714–14721]. Since it can be seen from the above that the density of defect states is the lowest in the $(\text{PEA})_2\text{SnI}_4$ sample, so the trapping rate of excitons and the proportion of process I in relaxation dynamics are the lowest, which is consistent with our experimental data (Supplementary Fig. 14 and Fig. 6f).” (lines 489-577)

In addition, we add content discussion to further confirm the validity of our conclusions.

In addition, the relaxation rate of the second component (II) in PEAI-rich with smaller defect density states is slower than that of $(\text{PEA})_2\text{SnI}_4$ and PEAI-rich has higher PLQY (Supplementary Fig. S16), indicating that the effect of deformation potential scattering by charged defects on exciton interband recombination can be weakened by reducing the density of defect states, and improve the PLQY. (lines 639-644)

Fig. 6 | Ground state bleaching (GSB) relaxation process. (a) Temperature-dependent normalized GSB relaxation kinetics of the (PEA)₂SnI₄ sample within 5 ps under the pump fluence of 3 $\mu\text{J cm}^{-2}$. (b) Relaxation kinetics of the TA of each wavelength labeled by the black dotted line in Fig. 5(a) within 3 ps. (c) the TA spectra of the (PEA)₂SnI₄ and PEA-rich sample under the pump fluence of 2 $\mu\text{J cm}^{-2}$. (d) Pump fluence-dependent normalized GSB relaxation kinetics of the (PEA)₂SnI₄ sample within 5 ps. (e) Pump fluence-dependent maximum value of the PIA signal at 624 nm of the (PEA)₂SnI₄ sample. When the pump fluence is less than 15 $\mu\text{J cm}^{-2}$, it is a linear excitation range, and when it exceeds 15 $\mu\text{J cm}^{-2}$, it is a nonlinear excitation range. n is the density of the photoinduced exciton that is proportional to the pump fluence F . The solid black line represents a linear relationship and the solid blue line represents a one-half power relationship. (f) Normalized band edge GSB relaxation process under the pump fluence of 12 $\mu\text{J cm}^{-2}$ for the (PEA)₂SnI₄, (BA)₂SnI₄, (HA)₂SnI₄, and (OA)₂SnI₄ thin films. The internal illustration shows the

relaxation process of the four materials within 2 ps. (lines 405-418)

Supplementary Figure 16. The influence of the cations stoichiometry engineering on the ground state bleaching (GSB) relaxation process of (PEA)₂SnI₄. (a) Characteristics of the ratio (P_{TE}/P_{FE}) of trap state exciton PL intensity to free exciton PL intensity as a function of temperature. (b) PLQY of (PEA)₂SnI₄ and PEAI-rich. (c) Relaxation process of the captured state of (PEA)₂SnI₄ and PEAI-rich. (d) Pump fluence-dependent normalized GSB relaxation kinetics of the (PEA)₂SnI₄ and PEAI-rich. (e) Ground state bleaching (GSB) relaxation process of (PEA)₂SnI₄ and PEAI-rich under the pump fluence of $2 \mu\text{J cm}^{-2}$. (f) The PLQY of (BA)₂PbI₄ and (BA)₂SnI₄ obtained under the same experimental conditions. (Supporting Information lines 210-218)

6. The conclusion of the authors is that for tin based 2D perovskite, “it is different from the Pb-based perovskite characterized by high defect tolerance.” Line 562 However, the PL quantum yield reported for 2D lead halide perovskite thin films is very low (<1%) (see Yuan et al. Nature Nanotechnology 11 (10) 872—877 (2016) and Duim et al. Advanced Functional Materials, 30 (5) 1907505 (2019) In reality, the PLQY reported in this manuscript for tin based 2D perovskites is much higher than that. The conclusion is not consistent with the results.

Reply: We are grateful to the reviewers for their comments, which helped us to correct the wrong expressions and perceptions. For the defect tolerance, it is often for three-dimensional Pb-based perovskites, mainly due to the easy migration of ions in the inner of three-dimensional perovskites, which can reduce the non-radiative composite centers [J. Phys. Chem. Lett. 2017, 8, 2, 489–493]. For two-dimensional (2D) Pb-based perovskites (n=1), the defect density of states is relatively high, which greatly increases the probability of non-radiative recombination [Nature Nanotech. 11, 2016, 872–877 and Phys. Rev. Appl. 2014, 2, 034007.]. However, compared to 2D Pb-based perovskites, the oxidation potential of $\text{Sn}^{2+}/\text{Sn}^{4+}$ (–0.15 eV) is considerably lower than that of $\text{Pb}^{2+}/\text{Pb}^{4+}$ (–1.8 eV) [ACS Energy Lett., 2017, 2, 1089-1098], 2D Sn-based perovskites have higher Sn^{4+} defect states than the 2D Pb-based perovskites, which can decrease the PLQY. In addition, the magnitude of PLQY depends on the sample fabrication method and experimental conditions (excitation intensity, temperature), so the PLQY in different literatures comparing with each other is generally not rigorous enough, but to some extent indicates the general characteristics of the problem. In order to rigorously compare the PLQY magnitudes of 2D Pb-based perovskites and 2D Sn-based perovskites, we used the same sample fabrication method and the same experimental conditions, and we obtained that the PLQY of $(\text{BA})_2\text{PbI}_4$ (0.54%) is higher than that of $(\text{BA})_2\text{SnI}_4$ (0.15%) (Supplementary Fig. 16f). This result is consistent with the theoretical results, in which 2D Sn-based perovskites have higher defect states than the 2D Pb-based perovskites decreasing the PLQY. In the previous manuscript, we tested the fluorescence quantum yield of two-dimensional tin-based chalcogenides inaccurately, mainly because the unsuitable test conditions reduced the signal-to-noise ratio of the experimental data and obtained incorrect data. We optimized the experimental conditions (Figure R1) to improve the signal-to-noise ratio of the experimental data and obtained reasonably correct data in Fig. 1b.

We corrected the experimental data in the original manuscript (Fig. 1b) and added relevant content discussion in the corresponding position in the manuscript (Supplementary Fig. 16f), as shown:

In the original manuscript:

“For the 2D Sn-based perovskites, it has higher defect states than 2D Pb-based perovskites.”

was revised to

“For the 2D Sn-based perovskites, it has higher defect states than 2D Pb-based

perovskites, obtained from the PLQY measured under the same experimental conditions (Supplementary Fig. 16f).” (lines 618-621)

In the original manuscript:

“So it is different from the Pb-based perovskite characterized by high defect tolerance.”

was revised to

“So it is different from the Pb-based perovskite characterized by fewer defect states, in which the main scattering mechanisms for excitons in the scatterings via deformation potential by acoustic and homopolar optical phonons [ACS Nano 2016, 10, 9992–9998].” (lines 679-681)

Optimization of experimental conditions

Figure R1. Optimization of experimental conditions for PLQY of $(\text{BA})_2\text{SnI}_4$

Figure R2. Correction of PLQY experimental data in Figure 1b of the manuscript.

Supplementary Figure 16. The influence of the cations stoichiometry engineering on the ground state bleaching (GSB) relaxation process of $(\text{PEA})_2\text{SnI}_4$. (a) Characteristics of the ratio ($P_{\text{TE}}/P_{\text{FE}}$) of trap state exciton PL intensity to free exciton PL intensity as a function of temperature. (b) PLQY of $(\text{PEA})_2\text{SnI}_4$ and PEAI-rich. (c) Relaxation process of the captured state of $(\text{PEA})_2\text{SnI}_4$ and PEAI-rich. (d) Pump fluence-dependent normalized GSB relaxation kinetics of the $(\text{PEA})_2\text{SnI}_4$ and PEAI-rich. (e) Ground state bleaching (GSB) relaxation process of $(\text{PEA})_2\text{SnI}_4$ and PEAI-rich under the pump fluence of $2 \mu\text{J cm}^{-2}$. (f) The PLQY of $(\text{BA})_2\text{PbI}_4$ and $(\text{BA})_2\text{SnI}_4$ obtained under the same experimental conditions. (Supporting Information lines 209-218)

7. Line 487 the sentence is unclear : “Moreover, if the component I is caused by the

bandgap renormalization process, its maximum value of the PIA should be proportional to $n^{1/2}$.”

Reply: Fig. 5 shows the false-color 2D TA mappings of the (PEA)₂SnI₄, (BA)₂SnI₄, (HA)₂SnI₄, and (OA)₂SnI₄ thin films. Since the characteristics of the steady-state absorption spectrum and the transient absorption spectrum of the four materials are essentially similar, the (PEA)₂SnI₄ sample is applied as the representative. There are two main bleaching peaks whose centers are at 615 and 525 nm, corresponding to the absorption peaks in the linear absorption spectra attributed to the band filling effect. A weak and broad (650–730 nm) bleaching signal below the bandgap is caused by the photogenerated carriers trapped by the defect state with low absorption cross-sections below the optical bandgap [J. Am. Chem. Soc. 2015, 137, 2089–2096 and J. Phys. Chem. C 2015, 119, 14714–14721]. Besides the bleaching signals, positive photoinduced absorption (PIA) signals exist on both sides of the bleaching peak. Especially for within 1 ps, the PIA center at 627 nm resulted from the bandgap renormalization, the redshift of the photobleaching peak of the band-edge exciton (ΔE_{BGN}), is caused by hot excitons. And once hot carriers cooled down and band filling effect settled in, the PIA band at 627 nm decayed (Fig. 6b) [J. Am. Chem. Soc. 2015, 137, 2089–2096, Science 2017, 356, 59–62, and Nature Photon. 2016, 10, 53–59]. Beard et al. revealed that the $-\Delta T/T$ of the PIA band empirically scaled linearly with ΔE_{BGN} [Nature Photon. 2016, 10, 53–59]. In two-dimensional perovskites with strong carrier-carrier interactions, the shift magnitude of bandgap renormalization (ΔE_{BGN}) is proportional to the carrier density $n^{1/2}$ [Science 2017, 356, 59–62 and J. Appl. Phys. 2007, 101, 083705]. So the $-\Delta T/T$ of the PIA band was empirically scaled linearly with $n^{1/2}$. We find that the amplitude $-\Delta T/T$ of the PIA extracted at different excitation intensities satisfies a linear relationship with the excitation intensity $n^{1/2}$, so this result also further confirms the presence of the bandgap renormalization process in the process I. In summary, we have made the following revisions to the contents of the manuscript.

In the original manuscript:

“Moreover, if the component I is caused by the bandgap renormalization process, its maximum value of the PIA should be proportional to $n^{1/2}$.”

was revised to

“In the TA spectrum within 1 ps (Fig. 5), the redshift of the photobleaching peak of the band-edge exciton leading to photoinduced absorption is observed, which is attributed to the bandgap renormalization caused by the hot excitons [J. Am. Chem. Soc. 2015, 137, 2089–2096], where the maximum amplitude $-\Delta T/T$ of the PIA extracted at different excitation intensities satisfies a linear relationship with the excitation intensity $n^{1/2}$ (Fig. 6e) [Science 2017, 356, 59–62, Nature Photon. 2016, 10, 53–59, and J. Appl. Phys. 2007, 101, 083705]. So this result also further confirms the presence of the bandgap recombination process in process I.” (lines 544-550)

8. Line 506, the authors write: “From the above, it can be inferred that excitonic contribution to the PL is dominant in the materials and excitons exist stably at room

temperature. Thus, the view that excitons split into free carriers can be excluded, although the experimental phenomena are similar". It seems to me that the results are not sufficient to support this conclusion. Why the fact that the proportion of radiative recombination increases imply that the excitonic contribution to the PL is dominant? The experimental phenomena are similar to what?

Reply: We are grateful to the reviewer for the comments. In the manuscript, we have demonstrated that the PL of four materials is generated by exciton radiative recombination by excitation intensity-dependent PL experiments (Supplementary Fig.6 and Supplementary Fig.13). And we obtained exciton binding energy at room temperature greater than thermal energy ($K_B T = 25$ meV at 300 K) without exciton fission into free electrons and holes. Moreover, we attribute process II to the interband radiative recombination process of excitons that are mainly affected by charge defects, which can annihilate the radiative recombination of excitons and accelerate the relaxation rate. The steady-state PL intensity increases with decreasing temperature, which indicates that the proportion of exciton radiative recombination process in the whole relaxation process of excitons increases, which in turn increases the PLQY. Besides, we can learn that the proportion (P_{TE}/P_{FE}) of PL of trapped state excitons to that of free excitons decreases with temperature by the previous detailed discussion of the fifth question, thus indicating that the proportion of free excitons that occur band-side radiative recombination increases with decreasing temperature in the exciton relaxation process. What's more, we used the stoichiometry engineering of the cations to reduce the density of defect states in $(PEA)_2SnI_4$ and increase the PLQY. This optimization reduced the occupancy of process I, decreased the relaxation rate of process II, and improves the percentage of process II in the exciton relaxation process (Fig.6c and Supplementary Fig. 16e). From the above, it can be inferred that reduced charged defects scattering helps to improve the radiative recombination efficiency of free excitons.

In three-dimensional perovskites, the carriers generated by photoexcitation exist mainly in the form of free electrons and holes. As the temperature increases to the room temperature, a decrease in bimolecular band-to-band radiative recombination with increasing temperatures because of more thermal spreading of electrons and holes across the conduction and valence bands [Adv. Funct. Mater. 2020, 30, 2004312, Adv. Funct. Mater. 2015, 25, 6218, and Nat. Commun. 2018, 9, 293]. In the three-dimensional and two-dimensional perovskite systems, the interband recombination rate of free carriers both increase with decreasing temperature. However, the factors that play a major impact are different, and the reason involved may lie in the fact that three-dimensional perovskites have higher ion mobility, high defect tolerance, and very weak interaction between free electrons and holes.

In summary, we have made the following revisions to the contents of the manuscript.

In the original manuscript:

“The integral steady-state PL intensity of the $(PEA)_2SnI_4$ sample increases gradually as the temperature decreases (Supplementary Fig. 3a), the implication being that the proportion of the radiative recombination component increases in the

whole relaxation process. From the above, it can be inferred that excitonic contribution to the PL is dominant in the materials and excitons exist stably at room temperature. Thus, the view that excitons split into free carriers can be excluded, although the experimental phenomena are similar.”

was revised to

“Combined with the results of the proportion (P_{TE}/P_{FE}) of PL of trapped state excitons to that of free excitons decreases as the temperature decreases, the results of the integral steady-state PL intensity of the $(PEA)_2SnI_4$ sample gradually increasing as the temperature decreases (Supplementary Fig. 4a and Supplementary Fig. 16a) indicates that the proportion of free excitons that occur band-side radiative recombination increases with decreasing temperature in the exciton relaxation process. What’s more, we used the stoichiometry engineering of the cations to reduce the density of defect states in $(PEA)_2SnI_4$ and increase the PLQY. This optimization reduced the occupancy of process I, decreased the relaxation rate of process II, and improves the percentage of process II in the exciton relaxation process (Fig.6c, Supplementary Fig. 16 e). From the above, it can be inferred that reduced charged defects scattering can improve the radiative recombination efficiency of free excitons. In the three-dimensional and two-dimensional perovskite systems, the interband recombination rate of free carriers both increase with decreasing temperature [Adv. Funct. Mater. 2020, 30, 2004312, Adv. Funct. Mater. 2015, 25, 6218, and Nat. Commun. 2018, 9, 293]. However, the factors that play a major impact are usually different, and the reason involved may lie in the fact that three-dimensional perovskites have higher ion mobility, high defect tolerance, and very weak interaction between free electrons and holes.” (lines 592-5608)

9. The third process of the transient absorption is assigned to trapped state exciton radiative recombination. However, the PL quantum yield is relatively low and it seems reasonable to think that the majority of exciton recombine through non-radiative processes, mainly due to trap states.

Reply: Many thanks to the reviewer for this critical and high quality comment. we have re-done systematically the temperature-dependent, excitation-fluence-dependent, and excitation-energy-dependent transient absorption experiments and temperature-dependent PL experiments, combined with a more thorough study of the literature. From the answers to all the above questions, the first fast process I of transient absorption measurement is attributed to the combination of the defects trapping excitons process and the band-gap renormalization process induced by hot excitons, i.e., the defects trapping excitons process play a leading role at low excitation fluence and as the excitation fluence increases, the bandgap renormalization process induced by hot excitons dominates in the process I. The second component (II) is derived from the band edge free exciton recombination process that is mainly affected by charge defects, which can annihilate the radiative recombination of excitons and accelerate the non-radiative relaxation rate. The third component (III) is attributed to defect-assisted excitons recombination, which

involves the radiative recombination induced by relatively shallow defect states, where the ratio of the PL the defect states (center at 660 nm) to that of the free exciton (center at 614nm) is about 30% at room temperature (Supplementary Fig. 16a), and the non-radiative recombination processes induced by deep defects which make the PLQY very low. Besides, the emission range of relatively shallow defect states increases with the enhancement of exciton-phonon scattering interaction among all the four materials (Fig.3, Fig.4, and Supplementary Fig. 11), which indicates the extrinsic STE state may exist. Deschler et al. indicated that the broad emission below the optical gap seen at low temperatures in <001> oriented 2D-perovskite materials was due to the light-induced formation of localized trap states, associated with interstitial iodide and iodide Frenkel defects that act as color centers in the crystal [J. Am. Chem. Soc. 2017, 139, 18632-18639]. Besides, Loi et al also highlighted the extrinsic origin of their broadband emission in-gap states in the crystal bulk are responsible for the broad emission [Nat. Commun. 2020, 11, 2344]. We applied the stoichiometry engineering of the cations where the PEAI:SnI₂ ratio is 2.6:1 in (PEA)₂SnI₄ (PEAI-rich) [Nat. Commun. 2020, 11, 2344, and Adv. Funct. Mater. 2020, 30, 1907505], we reduce the defect density in PEAI-rich to improve the PLQY of the PEAI-rich to 2.2% (Supplementary Fig. 11b), which makes the integrated area of the bleaching peak relaxation process of trap states center at 676 nm of PEA-rich after 1.2 ps smaller than that of PEA. (Supplementary Fig. 16c).

In the original manuscript:

“Conversely, the third long relaxation component (III) with a nanosecond lifetime is derived from the trapped state exciton radiative recombination process”
was revised to

“Conversely, the third long relaxation component (III) with a nanosecond lifetime is attributed to defect-assisted excitons recombination, which involves the radiative recombination induced by relatively shallow defect states, where the ratio of the PL the defect states (center at 660 nm) to that of the free exciton (center at 614nm) is about 30% at room temperature (Supplementary Fig. 16a), and the non-radiative recombination processes induced by deep defects which make the PLQY very low. Besides, the emission range of relatively shallow defect states increases with the enhancement of exciton-phonon scattering interaction among all the four materials (Fig.3, Fig.4, and Supplementary Fig. 11), which indicates the extrinsic STE state may exists.[J. Am. Chem. Soc. 2017, 139, 18632-18639 and Nat. Commun. 2020, 11, 2344]” (lines 625-634)

Finally, we would like to thank the reviewer once again for your valuable comments. We hope that our revisions will be to your satisfaction.

REVIEWERS' COMMENTS

Reviewer #1 (Remarks to the Author):

The authors have addressed most of my major concerns. Therefore, I do recommend the revised version of the manuscript for publication in Nature Communications.

Reviewer #2 (Remarks to the Author):

The authors have provided very long and detailed answers to the questions and also additional data. They present in the manuscript a large amount of interesting results. The main problem is that the text is made more cumbersome. The clarity of the text should be improved. Some part of the discussion could likely be moved in supplementary. The authors should get more to the point and avoid very long sentences of more than 50 words.

Examples of long sentences with poor readability:

"In the temperature-dependent PL experiment, using the multi-peak fitting methods mentioned earlier in the manuscript, the ratio of the PL percentage of the free excitons (PFE) to that of the trap state excitons (PTE) below the band gap decreases with decreasing temperature (Supplementary Fig. 16a), which is obviously opposite to the feature of the intrinsic STE states emission, in which the stronger luminescence from the intrinsic STE state and the band edge exciton luminescence intensity decreases as the temperature decreases, mainly because the thermal activation of the detrapping process cannot meet requirements of the detrapping barrier and the self-trapped excitons cannot return to the band edge at low temperatures."

"Combined with the results of the proportion (PTE / PFE) of PL of trapped state excitons to that of free excitons decreases as the temperature decreases, the results of the integral steady-state PL intensity of the (PEA)₂SnI₄ sample gradually increasing as the temperature decreases (Supplementary Fig. 4a and Supplementary Fig. 16a) indicates that the proportion of free excitons that occur band-side radiative recombination increases with decreasing temperature in the exciton relaxation process."

"For the first component (I) with the subpicosecond lifetime, it is attributed to the combination of the defects trapping excitons process and the band-gap renormalization process induced by hot excitons, i.e., the defects trapping excitons process play a leading role at low excitation fluence and as the excitation fluence increases, the bandgap renormalization process induced by hot excitons dominates in the process I. excluding the exciton formation process and optical Stark effect via the pump fluence-dependent, temperature-dependent TA experiments and stoichiometry engineering of the cations."

I have the following additional remarks:

1. Regarding the discussion on the nature of transition at 520nm, intraband or CT: the attribution to intraband transition seems quite reasonable. The justification of the authors to rule out the CT hypothesis is quite long and could be moved in supplementary.
2. I agree that the exciton binding energy (Eb) could be difficult to determine experimentally, in part as claimed by the author due to the polar nature of hybrid perovskite. However, the associated effect is the formation of polaron, not polariton.
3. The difficulty of measuring Eb is mentioned twice (line 215 and line 457). Repetition should be avoided.
4. The authors evaluate the exciton binding energy based on Elliott theory. Can the authors precise the formulation used to fit the absorption. I expect it to be a generalization of the Elliott theory for 2D systems?

5. The authors described the “process I” as due to two different physical processes: recombination from defects, and band gap renormalization “dominates the process I” at high fluence. This is not clear and a rather confusing way to present the results. It would be better to speak only of the “component I” or the fast component and to separate the low fluence and high fluence regime. At low fluence, the fastest component of the dynamic correspond to defect recombination. At high excitation, a new process (no more the process I) appears due to band gap renormalization...

Response to Reviewer Comments

REVIEWERS' COMMENTS

Reviewer #2 (Remarks to the Author):

The authors have provided very long and detailed answers to the questions and also additional data. They present in the manuscript a large amount of interesting results. The main problem is that the text is made more cumbersome. The clarity of the text should be improved. Some part of the discussion could likely be moved in supplementary. The authors should get more to the point and avoid very long sentences of more than 50 words.

Examples of long sentences with poor readability:

“In the temperature-dependent PL experiment, using the multi-peak fitting methods mentioned earlier in the manuscript, the ratio of the PL percentage of the free excitons (P_{FE}) to that of the trap state excitons (P_{TE}) below the band gap decreases with decreasing temperature (Supplementary Fig. 16a), which is obviously opposite to the feature of the intrinsic STE states emission, in which the stronger luminescence from the intrinsic STE state and the band edge exciton luminescence intensity decreases as the temperature decreases, mainly because the thermal activation of the detrapping process cannot meet requirements of the detrapping barrier and the self-trapped excitons cannot return to the band edge at low temperatures.”

“Combined with the results of the proportion (P_{TE}/P_{FE}) of PL of trapped state excitons to that of free excitons decreases as the temperature decreases, the results of the integral steady-state PL intensity of the $(\text{PEA})_2\text{SnI}_4$ sample gradually increasing as the temperature decreases (Supplementary Fig. 4a and Supplementary Fig. 16a) indicates that the proportion of free excitons that occur band-side radiative recombination increases with decreasing temperature in the exciton relaxation process.”

“For the first component (I) with the subpicosecond lifetime, it is attributed to the combination of the defects trapping excitons process and the band-gap renormalization process induced by hot excitons, i.e., the defects trapping excitons process play a leading role at low excitation fluence and as the excitation fluence increases, the bandgap renormalization process induced by hot excitons dominates in the process I. excluding the exciton formation process and optical Stark effect via the

pump fluence-dependent, temperature-dependent TA experiments and stoichiometry engineering of the cations.”

I have the following additional remarks:

1. Regarding the discussion on the nature of transition at 520nm, intraband or CT: the attribution to intraband transition seems quite reasonable. The justification of the authors to rule out the CT hypothesis is quite long and could be moved in supplementary.
2. I agree that the exciton binding energy (E_b) could be difficult to determine experimentally, in part as claimed by the author due to the polar nature of hybrid perovskite. However, the associated effect is the formation of polaron, not polariton.
3. The difficulty of measuring E_b is mentioned twice (line 215 and line 457). Repetition should be avoided.
4. The authors evaluate the exciton binding energy based on Elliott theory. Can the authors precise the formulation used to fit the absorption. I expect it to be a generalization of the Elliott theory for 2D systems?
5. The authors described the “process I” as due to two different physical processes: recombination from defects, and band gap renormalization “dominates the process I” at high fluence. This not clear and a rather confusing way to present the results. It would be better to speak only of the “component I” or the fast component and to separate the low fluence and high fluence regime. At low fluence, the fastest component of the dynamic correspond to defect recombination. At high excitation, a new process (no more the process I) appear due to band gap renormalization...

We are very grateful to the reviewer for taking the valuable time to review our responses, and we appreciate your approval of our responses. It is because of your review comments that we have been able to promote more depth in our findings; furthermore, with the help of your critical review comments, we have further clarified the central points of our research manuscript and enhanced the revised manuscript to be more logical and readable through special English language touch-ups.

1. Regarding the discussion on the nature of transition at 520nm, intraband or CT: the attribution to intraband transition seems quite reasonable. The justification of the authors to rule out the CT hypothesis is quite long and could be moved in

supplementary.

Reply: We have moved the entire discussion on the nature of the transition at 520 nm attributed to the intraband transition to the supporting supplementary Note 3 of the manuscript.

2. I agree that the exciton binding energy (E_b) could be difficult to determine experimentally, in part as claimed by the author due to the polar nature of hybrid perovskite. However, the associated effect is the formation of polaron, not polariton.

Reply: Many thanks to the reviewers for their critical comments, which made us realize that we had misspelled polaron as polariton. We have corrected this error in the manuscript as shown below:

“There are many methods to determine E_b , such as the absorption spectrum, temperature-dependent PL, and magneto-optical investigation. However, the polar nature of perovskites and the associated polaron effects are neglected; this makes the E_b values obtained by different methods under different experimental conditions highly discrepant.” (lines 205-209)

3. The difficulty of measuring E_b is mentioned twice (line 215 and line 457). Repetition should be avoided.

Reply: We have reorganized the paper and avoided the problem of repeating the exciton binding energy-related content twice. We have corrected this error in the manuscript as shown below:

“There are many methods to determine E_b , such as the absorption spectrum, temperature-dependent PL, and magneto-optical investigation. However, the polar nature of perovskites and the associated polaron effects are neglected; this makes the E_b values obtained by different methods under different experimental conditions highly discrepant. The accuracy of E_b obtained by fitting the temperature-dependent PL based on the Arrhenius formula is severely affected by other recombination processes, such as shallow defect trapping excitons and Auger recombination. Therefore, E_b was obtained by fitting the steady-state absorption spectrum using a more rigorous Elliott theory (Supplementary Fig. 9 and Supplementary Note 4.), in which the E_b of $(\text{PEA})_2\text{SnI}_4$ is 213 ± 2 meV smaller than that of $(\text{BA})_2\text{SnI}_4$ (248 ± 1.5 meV) because ε_{BA} is smaller than ε_{PEA} . E_b is greater than the thermal energy ($K_B T = 25$ meV at 300 K), which further reveals excitons dominate the optical transitions of the 2D layer Sn-based perovskites at room temperature.” (lines 205-218)

4. The authors evaluate the exciton binding energy based on Elliott theory. Can the authors precise the formulation used to fit the absorption. I expect it to be a generalization of the Elliott theory for 2D systems?

Reply: Many thanks to the reviewer for this comment. We use the Elliot equation applied to the 2D system for 2D Sn-based perovskites, and the precise formulation used was given in detail in the supporting supplementary Note 4 as shown below:

“For the estimation of the binding energy (E_b) of Wannier exciton contained in the direct bandgap semiconductor, an effective method is to fit the band-edge absorption spectrum using the Elliott formula [Physical Review 108, 1384-1389 (1957)]. According to Elliott theory of two-dimensional system [Physical Review B 52, 1978-1983 (1995), Solid State Communications 98, 65-68 (1996), and Physical Review Materials 2, 064605 (2018)], the absorption coefficient can be expressed as

$$\begin{aligned} \alpha(\omega) &= \alpha_{\text{exc}} + \alpha_{\text{cont}} \\ &= \alpha_0 \left[\sum_{n=1}^{\infty} \frac{4E_0}{\left(n - \frac{1}{2}\right)^3} \operatorname{sech}\left(\frac{\hbar\omega - E_g + \frac{E_0}{(n - 1/2)^2}}{\Gamma_{ex}}\right) \right. \\ &\quad \left. + \int_{E_g}^{\infty} \operatorname{sech}\left(\frac{\hbar\omega - x}{\Gamma_c}\right) \frac{2}{1 + \exp\left(-2\pi \sqrt{\frac{E_0}{\hbar\omega - E_g}}\right)} \right. \\ &\quad \left. \times \frac{1}{1 - (8\alpha m^*/\hbar^4)(x - E_g)} dx \right] \end{aligned}$$

Here, the first term on the right side of the equation is the excitonic absorption component below the band edge, and the second term is the free-carrier continuum absorption component. A term to correct for the non-parabolic band dispersion is introduced. E_g is the single-particle bandgap, m^* is the exciton reduced mass, a hyperbolic secant function to account for a phenomenological broadening Γ_{ex} and Γ_c represent exciton and free carrier transitions, respectively.” (Supplementary Note 4)

5. The authors described the “process I” as due to two different physical processes: recombination from defects, and band gap renormalization “dominates the process I” at high fluence. This not clear and a rather confusing way to present the results. It would be better to speak only of the “component I” or the fast component and to separate the low fluence and high fluence regime. At low fluence, the fastest component of the dynamic correspond to defect recombination. At high excitation, a new process (no more the process I) appear due to band gap renormalization...

Reply: We are very grateful to the reviewer for such a good suggestion, and we have further optimized the language and article structure in our manuscript to further improve the readability and comprehensibility of the article. We used “component I” for the first component (I) consistently throughout the text to avoid confusion, and optimized the relevant description of the first component (I) as follows:

“So the component I is attributed to the combination of the defect trapping exciton process and the band-gap renormalization process induced by hot excitons, that is, the defect trapping excitons process plays a leading role at low pump fluence, and the bandgap renormalization process dominates in component I at high pump fluence.” (lines 205-218)

Finally, we would like to thank the reviewer once again for your valuable comments. We hope that our revisions will be to your satisfaction.